# The Curious Price of Distributional Robustness in Reinforcement Learning with a Generative Model

**Laixi Shi**[*]  **Gen Li**[†]  **Yuting Wei**[‡]  **Yuxin Chen**[‡]  **Matthieu Geist**[§]  **Yuejie Chi**[¶]
Caltech         CUHK          UPenn          UPenn          Google          CMU

## Abstract

This paper investigates model robustness in reinforcement learning (RL) via the framework of distributionally robust Markov decision processes (RMDPs). Despite recent efforts, the sample complexity of RMDPs is much less understood regardless of the uncertainty set in use; in particular, there exist large gaps between existing upper and lower bounds, and it is unclear if distributional robustness bears any statistical implications when benchmarked against standard RL. In this paper, assuming access to a generative model, we derive the sample complexity of RMDPs — when the uncertainty set is measured via either total variation or $\chi^2$ divergence over the full range of uncertainty levels — using a model-based algorithm called distributionally robust value iteration, and develop minimax lower bounds to benchmark its tightness. Our results not only strengthen the prior art in both directions of upper and lower bounds, but also deliver surprising messages that learning RMDPs is not necessarily easier or more difficult than standard MDPs. In the case of total variation, we establish the minimax-optimal sample complexity of RMDPs which is always smaller than that of standard MDPs. In the case of $\chi^2$ divergence, we establish the sample complexity of RMDPs that is tight up to polynomial factors of the effective horizon, and grows linearly with respect to the uncertainty level when it approaches infinity.

## 1 Introduction

Reinforcement learning (RL) deals with the problem of learning to make sequential decisions based on trial-and-error interactions with some unknown environment. As a fast-growing subfield of artificial intelligence, it has achieved significant success in a variety of domains such as large language model alignment (OpenAI, 2023; Ziegler et al., 2019), healthcare (Liu et al., 2019; Fatemi et al., 2021), robotics and control (Kober et al., 2013; Mnih et al., 2013). Due to the high dimensionality of the state-action space, achieving sample efficiency lies at the core of modern RL practice, especially in various sample-starved applications. As a result, a large portion of efforts in RL has been put in designing sample-efficient algorithms and understanding the fundamental statistical difficulty for diverse RL problems (Azar et al., 2013; Li et al., 2020).

While standard RL has been heavily invested recently, its use can be significantly hampered in practice due to the sim-to-real gap, where an agent trained in an ideal, nominal environment might be extremely sensitive and fail catastrophically when the deployed environment is subject to small changes in task objectives or unexpected perturbations (Zhang et al., 2020a; Klopp et al., 2017; Mahmood et al., 2018). Consequently, in addition to maximizing the long-term cumulative reward,

---

[*]Department of Computing Mathematical Sciences, California Institute of Technology, CA, USA.

[†]Department of Statistics, The Chinese University of Hong Kong, Hong Kong, China.

[‡]Department of Statistics and Data Science, Wharton School, University of Pennsylvania, PA, USA.

[§]Google Research, Brain Team, Paris, France.

[¶]Department of Electrical and Computer Engineering, Carnegie Mellon University, PA, USA.

37th Conference on Neural Information Processing Systems (NeurIPS 2023).

robustness becomes another critical goal for an RL agent, especially in high-stake applications such as robotics, autonomous driving, clinical trials, financial investments, and so on. To address this, distributionally robust RL (Iyengar, 2005; Nilim and El Ghaoui, 2005), which leverages insights from distributionally robust optimization and supervised learning (Rahimian and Mehrotra, 2019; Gao, 2020; Bertsimas et al., 2018; Duchi and Namkoong, 2018; Blanchet and Murthy, 2019), becomes a natural and versatile framework with the goal of learning a policy that performs well even when the deployed environment deviates from the nominal one in the face of environment uncertainty.

In this paper, we are particularly interested in understanding whether, and how, the choice of distributional robustness bears statistical implications in learning the desired policy, by studying the sample complexity in the widely-used generative model (Kearns and Singh, 1999). Suppose that one has access to a generative model which draws samples from a Markov decision processes (MDP) with a nominal transition kernel. Standard RL aims to learn the optimal policy tailored for the nominal kernel based on this set of samples, where the sample complexity has been well understood with matching upper and lower bounds developed recently (Azar et al., 2013; Li et al., 2020). In contrast, distributionally robust RL — leveraging the same set of samples — aims to learn the optimal *robust* policy whose worst-case performance is maximized when the transition kernel is from some *prescribed* uncertainty set around the nominal kernel, a setting that is referred to as the robust MDP (RMDPs).[6] Clearly, this ensures that the performance of the learned policy is robust and does not fail catastrophically as long as the sim-to-real gap is not too large. It is then natural to wonder how the robustness consideration impacts the RL performance: should we always prefer to learn a robust policy for a given set of samples? Is there a statistical premium when asking for additional robustness?

Compared with standard MDPs, RMDPs is a richer class of models since one additionally needs to prescribe the shape and size of the uncertainty set, which is usually hand-picked as a small ball around the nominal kernel measured with respect to some distance measure $\rho$ and uncertainty level $\sigma$. To ensure the tractability of solving RMDPs, the uncertainty set is usually assumed to obey certain structures, where the uncertainty set can be decomposed as a product of independent uncertainty subsets over each state or state-action pair (Zhou et al., 2021; Wiesemann et al., 2013), denoted as the $s$- and $(s, a)$-rectangular rectangularity respectively; in particular, our paper adopts the second choice by assuming the uncertainty set satisfies the $(s, a)$-rectangularity. An additional challenge with RMDPs arises from distribution shift, where the transition kernel drawn from the uncertainty set can be different from the nominal kernel, leading to complicated nonlinearity and nested optimization in the problem structure not present in standard MDPs.

## 1.1 Prior art and open questions

In this paper, we focus on understanding the sample complexity of learning the optimal robust policy of RMDPs in the infinite-horizon setting assuming access to a generative model, when the uncertainty set is measured using one of the $f$-divergence: total variation (TV) distance and $\chi^2$ divergence. These two choices are motivated by their practical appeal: easy to implement, and already adopted by empirical RL (Lee et al., 2021; Pan et al., 2023).

A popular learning approach is model-based, which first estimates the nominal transition kernel using a plug-in estimator based on the collected samples, and then runs a planning algorithm such as a robust variant of value iteration on top of the estimated RMDP. Despite the surge of recent activities, however, existing statistical guarantees for the above paradigm remain highly inadequate, as we shall elaborate momentarily (see Table 1 and Table 2 respectively for a summary). For concreteness, let $S$ be the size of the state space, $A$ be the size of the action space, $\gamma$ be the discount factor (and the effective horizon $\frac{1}{1-\gamma}$), and $\sigma$ be the uncertainty level. We are interested in the sample complexity — the number of samples needed for an algorithm to output a policy whose robust value function (the worst-case value over all the transition kernels in the uncertainty set) is at most $\varepsilon$ away from the optimal robust one — with respect to all salient problem parameters.

• *Large gaps between existing upper and lower bounds.* There remained large gaps between the sample complexity upper and lower bounds established in prior literature, regardless of the divergence metric in use. Specifically, considering the cases using either TV distance or $\chi^2$ divergence, the state-of-the-art upper bounds (Panaganti and Kalathil, 2022) scales quadratically with the size $S$ of the

---

[6]While it is straightforward to incorporate additional uncertainty of the reward in our framework, we do not consider it here for simplicity, since the key challenge is to deal with the uncertainty of the transition kernel.

| Result type | Reference | Sample complexity | |
|---|---|---|---|
| | | $0 < \sigma \lesssim 1 - \gamma$ | $1 - \gamma \lesssim \sigma < 1$ |
| Upper bound | Yang et al. (2022) | $\frac{S^2 A (2+\sigma)^2}{\sigma^2 (1-\gamma)^4 \varepsilon^2}$ | |
| | Panaganti and Kalathil (2022) | $\frac{S^2 A}{(1-\gamma)^4 \varepsilon^2}$ | |
| | **Ours** | $\frac{SA}{(1-\gamma)^3 \varepsilon^2}$ | $\frac{SA}{(1-\gamma)^2 \sigma \varepsilon^2}$ |
| Lower bound | Yang et al. (2022) | $\frac{SA}{(1-\gamma)^3 \varepsilon^2}$ | $\frac{SA(1-\gamma)}{\sigma^4 \varepsilon^2}$ |
| | **Ours** | $\frac{SA}{(1-\gamma)^3 \varepsilon^2}$ | $\frac{SA}{(1-\gamma)^2 \sigma \varepsilon^2}$ |

Table 1: Comparisons between our results and prior arts for finding an $\varepsilon$-optimal robust policy in the infinite-horizon RMDPs with an uncertainty set measured with respect to the TV distance, where we ignore logarithmic factors in the sample complexities. Here, $S$, $A$, $\gamma$, and $\sigma \in (0, 1)$ are the state space size, the action space size, the discount factor, and the uncertainty level, respectively.

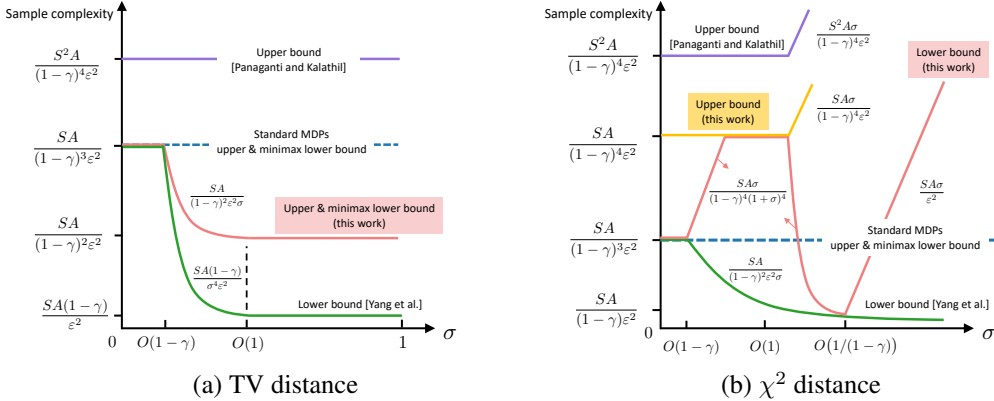

(a) TV distance        (b) $\chi^2$ distance

Figure 1: Illustrations of the obtained sample complexity upper and lower bounds for learning RMDPs with comparisons to state-of-the-art and the sample complexity of standard MDPs, where the uncertainty set is specified using the TV distance (a) and the $\chi^2$ distance (b).

state space, while the lower bound (Yang et al., 2022) exhibits only linear scaling with $S$. Moreover, in the $\chi^2$ divergence case, the state-of-the-art upper bound grows linearly with the uncertainty level $\sigma$ when $\sigma \gtrsim 1$,[7] while the lower bound (Yang et al., 2022) is inversely proportional to $\sigma$. These lead to unbounded gaps between the upper and lower bounds as $\sigma$ grows. *Can we hope to close these gaps for RMDPs?*

● *Benchmarking with standard MDPs.* Perhaps a more pressing issue is that, past works failed to provide an affirmative answer regarding how to benchmark the sample complexity of RMDPs with that of standard MDPs regardless of the chosen shape (determined by $\rho$) or size (determined by $\sigma$) of the uncertainty set, given the large unresolved gaps mentioned above. Specifically, existing sample complexity upper (resp. lower) bounds are all larger (resp. smaller) than the sample size requirement for standard MDPs. As a consequence, it remains mostly unclear *whether learning RMDPs is harder or easier than learning standard MDPs.*

## 1.2 Main contributions

To address these questions, we have developed new upper bounds on learning RMDPs with TV distance and $\chi^2$ distance in the infinite-horizon setting using the model-based approach called distributionally robust value iteration (DRVI), as well as minimax lower bounds to help gauge their

---

[7]Let $\mathcal{X} := \left(S, A, \frac{1}{1-\gamma}, \sigma, \frac{1}{\varepsilon}, \frac{1}{\delta}\right)$. The notation $f(\mathcal{X}) = O(g(\mathcal{X}))$ or $f(\mathcal{X}) \lesssim g(\mathcal{X})$ indicates that there exists a universal constant $C_1 > 0$ such that $f \leq C_1 g$, the notation $f(\mathcal{X}) \gtrsim g(\mathcal{X})$ indicates that $g(\mathcal{X}) = O(f(\mathcal{X}))$, and the notation $f(\mathcal{X}) \asymp g(\mathcal{X})$ indicates that $f(\mathcal{X}) \lesssim g(\mathcal{X})$ and $f(\mathcal{X}) \gtrsim g(\mathcal{X})$ hold simultaneously. Additionally, the notation $\widetilde{O}(\cdot)$ is defined in the same way as $O(\cdot)$ except that it hides logarithmic factors.

| Result type | Reference | Sample complexity | | |
|---|---|---|---|---|
| | | $0 < \sigma \lesssim 1-\gamma$ | $1-\gamma \lesssim \sigma \lesssim \frac{1}{1-\gamma}$ | $\sigma \gtrsim \frac{1}{1-\gamma}$ |
| Upper bound | Panaganti and Kalathil (2022) | $\frac{S^2 A(1+\sigma)}{(1-\gamma)^4 \varepsilon^2}$ | | |
| | Yang et al. (2022) | $\frac{S^2 A(1+\sigma)^2}{(\sqrt{1+\sigma}-1)^2(1-\gamma)^4 \varepsilon^2}$ | | |
| | **Ours** | $\frac{SA(1+\sigma)}{(1-\gamma)^4 \varepsilon^2}$ | | |
| Lower bound | Yang et al. (2022) | $\frac{SA}{(1-\gamma)^3 \varepsilon^2}$ | $\frac{SA}{(1-\gamma)^2 \sigma \varepsilon^2}$ | |
| | **Ours** | $\frac{SA}{(1-\gamma)^3 \varepsilon^2}$ | $\frac{SA\sigma}{(1-\gamma)^4(1+\sigma)^4 \varepsilon^2}$ | $\frac{SA\sigma}{\varepsilon^2}$ |

Table 2: Comparisons between our results and prior arts for finding an $\varepsilon$-optimal robust policy in the infinite-horizon RMDPs with an uncertainty set measured with respect to the $\chi^2$ distance, where we ignore logarithmic factors in the sample complexities. Here, $S$, $A$, $\gamma$, and $\sigma \in (0, \infty)$ are the state space size, the action space size, the discount factor, and the uncertainty level, respectively.

tightness and provide benchmarking with standard MDPs. As shall be outlined, these new analyses lead to new insights on the interplay between the geometry of uncertainty sets and the statistical sample complexity.

**Sample complexity of RMDPs under the TV distance.** We summarize the results and comparisons to prior works in Table 1; see Figure 1(a) for an illustration.

• *Minimax-optimal sample complexity.* We prove that DRVI reaches $\varepsilon$ accuracy as soon as the sample complexity is on the order of

$$\widetilde{O}\left(\frac{SA}{(1-\gamma)^2\varepsilon^2} \min\left\{\frac{1}{1-\gamma}, \frac{1}{\sigma}\right\}\right)$$

for all $\sigma \in (0, 1)$, assuming that $\varepsilon$ is small enough. In addition, a matching minimax lower bound (modulo some logarithmic factor) is established to guarantee the tightness of the upper bound. To the best of our knowledge, this is the *first* minimax-optimal sample complexity for RMDPs, which was previously unavailable regardless of the divergence metric and uncertainty level in use and is over the full range of the uncertainty level.

• *RMDPs are easier than standard MDPs under the TV distance.* Given the sample complexity $O\left(\frac{SA}{(1-\gamma)^3\varepsilon^2}\right)$ of standard MDPs, it can be seen that RMDPs under the TV distance is never harder than standard MDPs, where it matches that of standard MDPs when $\sigma \lesssim 1-\gamma$, and becomes simpler by a factor of $\sigma/(1-\gamma)$ when $1-\gamma \lesssim \sigma < 1$. Therefore, in this case, the robustness comes almost for free since we do not need to collect more samples to reach the same accuracy.

**Sample complexity of RMDPs under the $\chi^2$ distance.** We summarize the results and comparisons to prior works in Table 2; see Figure 1(b) for an illustration.

• *Near-optimal sample complexity.* It is established that DRVI reaches $\varepsilon$ accuracy as soon as the sample complexity is on the order of $\widetilde{O}\left(\frac{SA(1+\sigma)}{(1-\gamma)^4\varepsilon^2}\right)$ for all $\sigma \in (0, \infty)$, which is the first sample complexity that scales linearly with respect to the size of the state space $S$ in the infinite-horizon setting, breaking the quadratic scaling bottleneck presented in prior works (Panaganti and Kalathil, 2022; Yang et al., 2022). We have also developed a strengthened lower bound that is optimized by leveraging the geometry of the uncertainty set under different ranges of $\sigma$. Comparing the two bounds, they match at $\widetilde{O}\left(\frac{SA}{(1-\gamma)^4\varepsilon^2}\right)$ when $\sigma \asymp 1$, and have a bounded gap no larger than a polynomial factor of the effective horizon $1/(1-\gamma)$ over the entire range of the uncertainty level, again significantly improving prior art that exhibits an unbounded gap.

• *RMDPs can be harder than standard MDPs under the $\chi^2$ distance.* Somewhat surprisingly, the new lower bound suggests that RMDPs in this case can be much harder than standard MDPs, at least for certain ranges of the uncertainty level. We single out two regimes of particular interest. First, when $\sigma \asymp 1$, the sample size requirement of RMDPs is $\widetilde{O}\left(\frac{SA}{(1-\gamma)^4\varepsilon^2}\right)$, which is provably harder than

standard MDPs by a factor of the effective horizon $\frac{1}{1-\gamma}$. Second, when $\sigma \gtrsim \frac{1}{(1-\gamma)^3}$, the lower bound exceeds the sample complexity of standard MDPs and keeps growing linearly with the uncertainty level $\sigma$.

In sum, our sample complexity bounds not only strengthen the prior art in both directions of upper and lower bounds, but also reveal new insights on how the additional consideration of distributional robustness fundamentally changes the sample complexity of RL in a surprising manner. It turns out that RMDPs are not necessarily harder nor easier than standard MDPs, but the answer is far more nuanced and highly dependent on both the size and shape of the uncertainty set, which are not elucidated in prior analyses.

## 2 Problem formulation

In this section, we introduce the model of distributionally robust Markov decision processes (RMDPs) focusing on the discounted infinite-horizon setting, the sampling mechanism, as well as our goal.

**Standard MDPs.** To begin, we first introduce the standard Markov decision process (MDP), which facilitates the understanding of RMDPs. A discounted infinite-horizon MDP is represented by $\mathcal{M} = (\mathcal{S}, \mathcal{A}, P, \gamma, r)$, where $\mathcal{S} = \{1, \cdots, S\}$ and $\mathcal{A} = \{1, \cdots, A\}$ are the finite state and action spaces, respectively, $\gamma \in [0, 1)$ is the discount factor, $P : \mathcal{S} \times \mathcal{A} \to \Delta(\mathcal{S})$ denotes the probability transition kernel, and $r : \mathcal{S} \times \mathcal{A} \to [0, 1]$ is the immediate reward function which is assumed to be deterministic. A policy is denoted as $\pi : \mathcal{S} \to \Delta(\mathcal{A})$, which specifies the action selection probability over the action space in any state. When the policy is deterministic, we overload the notation and refer to $\pi(s)$ as the action selected by policy $\pi$ in state $s$. To characterize the long term cumulative reward, the value function $V^{\pi,P}$ for any policy $\pi$ under the transition kernel $P$ is defined by

$$\forall s \in \mathcal{S}: \qquad V^{\pi,P}(s) := \mathbb{E}_{\pi,P}\left[\sum_{t=0}^{\infty} \gamma^t r(s_t, a_t) \,\Big|\, s_0 = s\right], \tag{1}$$

where the expectation is taken over the randomness of the trajectory $\{s_t, a_t, r_t\}_{t=0}^{\infty}$ generated by executing policy $\pi$ under the transition kernel $P$, namely, $a_t \sim \pi(s_t)$, and $s_{t+1} \sim P(\cdot \,|\, s_t, a_t)$. Similarly, the Q-function $Q^{\pi,P}$ associated with any policy $\pi$ under the transition kernel $P$ is defined as

$$\forall (s, a) \in \mathcal{S} \times \mathcal{A}: \qquad Q^{\pi,P}(s, a) := \mathbb{E}_{\pi,P}\left[\sum_{t=0}^{\infty} \gamma^t r(s_t, a_t) \,\Big|\, s_0 = s, a_0 = a\right], \tag{2}$$

where the expectation is again taken over the randomness of the trajectory.

**Distributionally robust MDPs.** In this work, we focus on the discounted infinite-horizon distributionally robust MDP (RMDP), denoted as $\mathcal{M}_{\mathsf{rob}} = \{\mathcal{S}, \mathcal{A}, \mathcal{U}_\rho^\sigma(P^0), \gamma, r\}$, where $\mathcal{S}, \mathcal{A}, \gamma, r$ are defined the same as those in the above standard MDP. A key distinction from the standard MDP, is that rather than assuming a fixed transition kernel $P$, it postulates the transition kernel lies in an uncertainty set $\mathcal{U}_\rho^\sigma(P^0)$ centered around a *nominal* kernel $P^0 : \mathcal{S} \times \mathcal{A} \to \Delta(\mathcal{S})$, where the uncertainty set is specified using some distance metric $\rho$ of radius $\sigma > 0$. In particular, given the nominal transition kernel $P^0$ and some uncertainty level $\sigma$, the uncertainty set—with divergence $\rho : \Delta(\mathcal{S}) \times \Delta(\mathcal{S}) \to \mathbb{R}^+$—is specified as

$$\mathcal{U}_\rho^\sigma(P^0) := \otimes\, \mathcal{U}_\rho^\sigma(P_{s,a}^0), \qquad \mathcal{U}_\rho^\sigma(P^0) := \left\{P_{s,a} \in \Delta(\mathcal{S}) : \rho\left(P_{s,a}, P_{s,a}^0\right) \leq \sigma\right\}, \tag{3}$$

where we denote a vector of the transition kernel $P$ or $P^0$ at state-action pair $(s, a)$ respectively as

$$P_{s,a} := P(\cdot \,|\, s, a) \in \mathbb{R}^{1 \times S}, \qquad P_{s,a}^0 := P^0(\cdot \,|\, s, a) \in \mathbb{R}^{1 \times S}. \tag{4}$$

In other words, the uncertainty is imposed in a separate manner for each state-action pair, obeying the so-called $(s, a)$-rectangularity (Zhou et al., 2021; Wiesemann et al., 2013). In RMDPs, we are interested in the worst-case performance of a policy $\pi$ over all the possible transition kernels in the uncertainty set. This is measured by the *robust value function* $V^{\pi,\sigma}$ and the *robust Q-function* $Q^{\pi,\sigma}$ in $\mathcal{M}_{\mathsf{rob}}$, defined respectively as

$$\forall (s, a) \in \mathcal{S} \times \mathcal{A}: \quad V^{\pi,\sigma}(s) := \inf_{P \in \mathcal{U}_\rho^\sigma(P^0)} V^{\pi,P}(s), \qquad Q^{\pi,\sigma}(s, a) := \inf_{P \in \mathcal{U}_\rho^\sigma(P^0)} Q^{\pi,P}(s, a).$$

**Optimal robust policy and robust Bellman operator.** Generalizing standard MDPs, it is well-known that there exists at least one deterministic policy that maximizes the robust value function and

the robust Q-function simultaneously (Iyengar, 2005; Nilim and El Ghaoui, 2005). Therefore, we denote the *optimal robust value function* (resp. *optimal robust Q-function*) as $V^{\star,\sigma}$ (resp. $Q^{\star,\sigma}$), and the optimal robust policy as $\pi^\star$, which satisfies

$$\forall s \in \mathcal{S}: \quad V^{\star,\sigma}(s) := V^{\pi^\star,\sigma}(s) = \max_\pi V^{\pi,\sigma}(s), \tag{5a}$$

$$\forall (s,a) \in \mathcal{S} \times \mathcal{A}: \quad Q^{\star,\sigma}(s,a) := Q^{\pi^\star,\sigma}(s,a) = \max_\pi Q^{\pi,\sigma}(s,a). \tag{5b}$$

The robust Bellman operator (Iyengar, 2005; Nilim and El Ghaoui, 2005) is denoted as $\mathcal{T}^\sigma(\cdot)$ : $\mathbb{R}^{SA} \to \mathbb{R}^{SA}$, which is defined as follows: for all $(s,a) \in \mathcal{S} \times \mathcal{A}$,

$$\mathcal{T}^\sigma(Q)(s,a) := r(s,a) + \gamma \inf_{\mathcal{P} \in \mathcal{U}_\rho^\sigma(P_{s,a}^0)} \mathcal{P}V, \quad \text{with} \quad V(s) := \max_a Q(s,a). \tag{6}$$

Given that $Q^{\star,\sigma}$ is the unique fixed point of $\mathcal{T}^\sigma$, one can recover the optimal robust value function and Q-function using a procedure termed *distributionally robust value iteration*—which generalizes the standard value iteration—by recursively applying the robust Bellman operator from some fixed initialization. In addition, this procedure converges rather fast due to the nice $\gamma$-contraction property of $\mathcal{T}^\sigma$ (Iyengar, 2005; Nilim and El Ghaoui, 2005) with respect to the $\ell_\infty$ norm.

**Specification of the divergence $\rho$.** We consider two popular choices of the uncertainty set measured in terms of the $f$-divergence: total variation and $\chi^2$ divergence, given respectively by

$$\rho_{\mathsf{TV}}\left(P_{s,a}, P_{s,a}^0\right) := \frac{1}{2}\left\|P_{s,a} - P_{s,a}^0\right\|_1 = \frac{1}{2}\sum_{s' \in \mathcal{S}} P^0(s'\,|\,s,a)\left|1 - \frac{P(s'\,|\,s,a)}{P^0(s'\,|\,s,a)}\right|, \tag{7}$$

$$\rho_{\chi^2}\left(P_{s,a}, P_{s,a}^0\right) := \sum_{s' \in \mathcal{S}} P^0(s'\,|\,s,a)\left(1 - \frac{P(s'\,|\,s,a)}{P^0(s'\,|\,s,a)}\right)^2. \tag{8}$$

Note that $\rho_{\mathsf{TV}}\left(P_{s,a}, P_{s,a}^0\right) \in [0,1]$ and $\rho_{\chi^2}\left(P_{s,a}, P_{s,a}^0\right) \in [0,\infty)$ in general. As we shall see, the two choices convey drastically different messages in the statistical complexity of RMDPs.

**Sampling mechanism: a generative model.** Following Zhou et al. (2021); Panaganti and Kalathil (2022), we assume the access to a generative model or a simulator (Kearns and Singh, 1999), which allows us to collect $N$ independent samples from the *nominal* kernel $P^0$ at each state-action pair:

$$\forall (s,a) \in \mathcal{S} \times \mathcal{A}, \qquad s_{i,s,a} \overset{i.i.d}{\sim} P^0(\cdot\,|\,s,a), \qquad i = 1,2,\cdots,N. \tag{9}$$

The total sample size therefore is $NSA$.

**Goal.** Given the collected samples, the task is to learn the robust optimal policy for the RMDP with some uncertainty set $\mathcal{U}_\rho^\sigma(P^0)$ around the nominal kernel accurately using as few samples as possible. Specifically, given some accuracy level $\varepsilon > 0$, the goal is to seek an $\varepsilon$-optimal robust policy $\widehat{\pi}$ obeying $V^{\star,\sigma}(s) - V^{\widehat{\pi},\sigma}(s) \leq \varepsilon$ for all $s \in \mathcal{S}$.

## 3 Model-based algorithm: distributionally robust value iteration

We consider a model-based strategy, which first constructs an empirical nominal transition kernel based on the collected samples, and then applies distributionally robust value iteration (DRVI) to recover the optimal robust policy.

**Empirical nominal kernel.** The empirical nominal transition kernel $\widehat{P}^0 \in \mathbb{R}^{SA \times S}$ can be constructed using the empirical frequency of visits, i.e.

$$\forall (s,a) \in \mathcal{S} \times \mathcal{A}: \quad \widehat{P}^0(s'\,|\,s,a) := \frac{1}{N}\sum_{i=1}^N \mathbb{1}\{s_{i,s,a} = s'\}, \tag{10}$$

which leads to an empirical RMDP $\widehat{\mathcal{M}}_{\mathsf{rob}} = \{\mathcal{S}, \mathcal{A}, \mathcal{U}_\rho^\sigma(\widehat{P}^0), \gamma, r\}$. Analogously, we can define the corresponding robust value function (resp. robust Q-function) of policy $\pi$ in $\widehat{\mathcal{M}}_{\mathsf{rob}}$ as $\widehat{V}^{\pi,\sigma}$ (resp. $\widehat{Q}^{\pi,\sigma}$) (cf. (5)). In addition, we denote the corresponding *optimal robust policy* as $\widehat{\pi}^\star$ and the *optimal robust value function* (resp. *optimal robust Q-function*) as $\widehat{V}^{\star,\sigma}$ (resp. $\widehat{Q}^{\star,\sigma}$) (cf. (5)), which satisfies the robust Bellman optimality equation:

$$\forall (s,a) \in \mathcal{S} \times \mathcal{A}: \quad \widehat{Q}^{\star,\sigma}(s,a) = r(s,a) + \gamma \inf_{\mathcal{P} \in \mathcal{U}_\rho^\sigma(\widehat{P}_{s,a}^0)} \mathcal{P}\widehat{V}^{\star,\sigma}. \tag{11}$$

---

**Algorithm 1:** Distributionally robust value iteration (DRVI) for infinite-horizon RMDPs.

---

1  **input:** empirical nominal transition kernel $\widehat{P}^0$; reward function $r$; uncertainty level $\sigma$; number of iterations $T$.
2  **initialization:** $\widehat{Q}_0(s, a) = 0$, $\widehat{V}_0(s) = 0$ for all $(s, a) \in \mathcal{S} \times \mathcal{A}$.
3  **for** $t = 1, 2, \cdots, T$ **do**
4     **for** $s \in \mathcal{S}, a \in \mathcal{A}$ **do**
5         Set $\widehat{Q}_t(s, a)$ according to (13);
6     **for** $s \in \mathcal{S}$ **do**
7         Set $\widehat{V}_t(s) = \max_a \widehat{Q}_t(s, a)$;

8  **output:** $\widehat{Q}_T$, $\widehat{V}_T$ and $\widehat{\pi}$ obeying $\widehat{\pi}(s) := \arg\max_a \widehat{Q}_T(s, a)$.

---

Equipped with $\widehat{P}^0$, define the empirical robust Bellman operator $\widehat{\mathcal{T}}^\sigma$ as

$$\forall (s, a) \in \mathcal{S} \times \mathcal{A}: \ \widehat{\mathcal{T}}^\sigma(Q)(s, a) := r(s, a) + \gamma \inf_{\mathcal{P} \in \mathcal{U}_\rho^\sigma(\widehat{P}_{s,a}^0)} \mathcal{P}V, \ \text{ with } V(s) := \max_a Q(s, a). \quad (12)$$

**DRVI: distributionally robust value iteration.** To solve for the fixed point of $\widehat{\mathcal{T}}^\sigma$, we introduce distributionally robust value iteration (DRVI), which is summarized in Algorithm 1. Starting from some initialization $\widehat{Q}_0 = 0$, the update rule at the $t$-th ($t \geq 1$) step can be formulated as:

$$\forall (s, a) \in \mathcal{S} \times \mathcal{A}: \quad \widehat{Q}_t(s, a) = \widehat{\mathcal{T}}^\sigma\left(\widehat{Q}_{t-1}\right)(s, a) = r(s, a) + \gamma \inf_{\mathcal{P} \in \mathcal{U}_\rho^\sigma(\widehat{P}_{s,a}^0)} \mathcal{P}\widehat{V}_{t-1}, \quad (13)$$

where $\widehat{V}_{t-1}(s) = \max_a \widehat{Q}_{t-1}(s, a)$ for all $s \in \mathcal{S}$. However, directly solving (13) is computationally prohibitive since it involves optimization over an $S$-dimensional probability simplex at each iteration, especially when the dimension of the state space $\mathcal{S}$ is prohibitive. Fortunately, in view of strong duality (Iyengar, 2005), (13) can be equivalently solved using its dual problem, which concerns optimizing of a *scalar* dual variable and thus can be solved efficiently. The specific form of the dual problem depends on the choice of the divergence $\rho$, which we discuss in a more detailed version.

## 4 Theoretical guarantees: sample complexity analyses

We now present our main results, which concern the sample complexities of learning RMDPs when the uncertainty set is specified using the TV distance or the $\chi^2$ divergence. Surprisingly, the choice of the uncertainty set can lead to dramatic consequence in the sample size requirement.

### 4.1 The case of TV distance: RMDP is easier than standard MDP

We start with the case when the uncertainty set is measured via the TV distance, where Theorem 1 presents the sample complexity upper bound above which DRVI is able to find an $\varepsilon$-optimal robust policy in a small number of iterations; the key challenge of the analysis is to carefully control the robust value function $V^{\pi,\sigma}$ as a function of uncertainty level $\sigma$.

**Theorem 1** (Upper bound using TV distance). *Fix the uncertainty set $\mathcal{U}_\rho^\sigma(\cdot) = \mathcal{U}_{\mathsf{TV}}^\sigma(\cdot)$ using the TV distance in (7). Consider any discount factor $\gamma \in \left[\frac{1}{4}, 1\right)$, uncertainty level $\sigma \in (0, 1)$, and $\delta \in (0, 1)$. With probability at least $1 - \delta$, the output $\widehat{\pi}$ from Algorithm 1 with at most $T = C_1 \log\left(N(1 - \gamma)\right)$ iterations yields $V^{\star,\sigma}(s) - V^{\widehat{\pi},\sigma}(s) \leq \varepsilon$ for any $\varepsilon \in \left(0, \sqrt{1/\max\{1 - \gamma, \sigma\}}\right]$, as long as the total number of samples obeys $NSA \geq \frac{C_2 SA}{(1-\gamma)^2 \max\{1-\gamma, \sigma\} \varepsilon^2} \log\left(\frac{SAN}{(1-\gamma)\delta}\right)$. Here, $C_1, C_2 > 0$ are some large enough universal constants.*

Before discussing the implications of Theorem 1, we present a matching minimax lower bound that confirms the optimality of the upper bound, which in turn pins down the sample complexity requirement for learning RMDPs with TV distance.

**Theorem 2** (Lower bound using TV distance). *Consider any tuple $(S, A, \gamma, \sigma, \varepsilon)$ obeying $\sigma \in (0, 1 - c_0]$ with $0 < c_0 \leq \frac{1}{8}$ being any small enough positive constant, $\gamma \in \left[\frac{1}{2}, 1\right)$, and $\varepsilon \in \left(0, \frac{c_0}{256(1-\gamma)}\right]$. We*

can construct two infinite-horizon RMDPs $\mathcal{M}_0, \mathcal{M}_1$ defined by the uncertainty set $\mathcal{U}_\rho^\sigma(\cdot) = \mathcal{U}_{\mathsf{TV}}^\sigma(\cdot)$, an initial state distribution $\varphi$, and a dataset with $N$ independent samples for each state-action pair over the nominal transition kernel (for $\mathcal{M}_0$ and $\mathcal{M}_1$ respectively), such that

$$\inf_{\widehat{\pi}} \max \left\{ \mathbb{P}_0 \left( V^{\star,\sigma}(\varphi) - V^{\widehat{\pi},\sigma}(\varphi) > \varepsilon \right), \mathbb{P}_1 \left( V^{\star,\sigma}(\varphi) - V^{\widehat{\pi},\sigma}(\varphi) > \varepsilon \right) \right\} \geq \frac{1}{8},$$

provided that $NSA \leq \frac{c_0 SA \log 2}{8192(1-\gamma)^2 \max\{1-\gamma, \sigma\}\varepsilon^2}$. The infimum is taken over all estimators $\widehat{\pi}$, and $\mathbb{P}_0$ (resp. $\mathbb{P}_1$) denotes the probability when the RMDP is $\mathcal{M}_0$ (resp. $\mathcal{M}_1$).

Below, we interpret the above theorems and highlight several key implications about the sample complexity requirements for learning RMDPs with TV distance.

**Near minimax-optimal sample complexity.**   Theorem 1 shows that the total number of samples required for DRVI to yield $\varepsilon$-accuracy is

$$\widetilde{O} \left( \frac{SA}{(1-\gamma)^2 \max\{1-\gamma, \sigma\}\varepsilon^2} \right). \tag{14}$$

Taking together with the minimax lower bound asserted in Theorem 2, this confirms the near minimax-optimality of the sample complexity up to some logarithmic factor almost over the full range of the uncertainty level $\sigma$, which scales linearly with respect to the size of the state-action space.

**RMDPs is easier than standard MDPs with TV distance.**   Recall that the sample complexity requirement for standard MDP (Agarwal et al., 2020; Li et al., 2020) to yield $\varepsilon$ accuracy is $\widetilde{O} \left( \frac{SA}{(1-\gamma)^3 \varepsilon^2} \right)$. Comparing with the sample complexity requirement in (14) for RMDPs with TV distance, this confirms that the latter is at least as easy as—if not easier—than standard MDPs. In particular, when $\sigma \lesssim 1 - \gamma$ is small, the sample complexity of RMDPs is the same as the standard MDPs, which is expected since the RMDP reduces to the standard MDP when $\sigma = 0$. On the other hand, when $1 - \gamma \lesssim \sigma < 1$, the sample complexity of RMDPs becomes $\widetilde{O} \left( \frac{SA}{(1-\gamma)^2 \sigma \varepsilon^2} \right)$, which is smaller than that of standard MDPs by a factor of $\sigma/(1-\gamma)$.

**Comparison with state-of-the-art bounds.**   For the upper bound, our results (cf. Theorem 1) significantly improves over the prior art $\widetilde{O} \left( \frac{S^2 A}{(1-\gamma)^4 \varepsilon^2} \right)$ of Panaganti and Kalathil (2022) by at least a factor of $\frac{S}{1-\gamma}$ and even $\frac{S}{(1-\gamma)^2}$ when the uncertainty level $1 - \gamma \lesssim \sigma < 1$ is large. Turning to the lower bound side, Yang et al. (2022) developed a lower bound for RMDPs under the TV distance, which scales as $\widetilde{O} \left( \frac{SA(1-\gamma)}{\varepsilon^2} \min \left\{ \frac{1}{(1-\gamma)^4}, \frac{1}{\sigma^4} \right\} \right)$. Clearly, this is worse than ours by a factor of $\frac{\sigma^3}{(1-\gamma)^3} \in \left( 1, \frac{1}{(1-\gamma)^3} \right)$ in the regime where $1 - \gamma \lesssim \sigma < 1$.

## 4.2   The case of $\chi^2$ distance: RMDP can be harder than standard MDP

We now move onto the case when the uncertainty set is measured via the $\chi^2$ distance, where Theorem 3 presents the sample complexity upper bound above which DRVI is able to find an $\varepsilon$-optimal robust policy in a small number of iterations.

**Theorem 3** (Upper bound using $\chi^2$ distance). *Fix the uncertainty set $\mathcal{U}_\rho^\sigma(\cdot) = \mathcal{U}_{\chi^2}^\sigma(\cdot)$ using the $\chi^2$ distance in (8). Consider any uncertainty level $\sigma \in (0, \infty)$, and $\delta \in (0, 1)$. With probability at least $1 - \delta$, the output policy $\widehat{\pi}$ from Algorithm 1 with at most $T = c_1 \log (N(1-\gamma))$ iterations yields $V^{\star,\sigma}(s) - V^{\widehat{\pi},\sigma}(s) \leq \varepsilon$ for any $\varepsilon \in \left( 0, \frac{1}{1-\gamma} \right]$, as long as the total number of samples obeying $NSA \geq \frac{c_2 SA(1+\sigma)}{(1-\gamma)^4 \varepsilon^2} \log \left( \frac{SAN}{\delta} \right)$. Here, $c_1, c_2 > 0$ are some large enough universal constants.*

In addition, in order to gauge the tightness of Theorem 3, and understand the minimal sample complexity requirement for learning RMDPs with $\chi^2$ divergence, we further develop a minimax lower bound as follows.

**Theorem 4** (Lower bound using $\chi^2$ divergence). *Consider any $(S, A, \gamma, \sigma, \varepsilon)$ obeying $\gamma \in [\frac{3}{4}, 1)$, $\sigma \in (0, \infty)$, and*

$$\varepsilon \leq c_3 \begin{cases} \frac{1}{1-\gamma} & \text{if } \sigma \in \left( 0, \frac{1-\gamma}{4} \right), \\ \max \left\{ \frac{1}{\sigma(1-\gamma)}, 1 \right\} & \text{if } \sigma \in \left[ \frac{1-\gamma}{4}, \infty \right), \end{cases} \tag{15}$$

*for some small universal constant $c_3 > 0$. Then we can construct two infinite-horizon RMDPs $\mathcal{M}_0, \mathcal{M}_1$ defined by the uncertainty set $\mathcal{U}_\rho^\sigma(\cdot) = \mathcal{U}_{\chi^2}^\sigma(\cdot)$, an initial state distribution $\rho$, and a dataset with $N$ independent samples for each $(s, a)$ pair over the nominal transition kernel (for $\mathcal{M}_0$ and $\mathcal{M}_1$ respectively), such that*

$$\inf_{\widehat{\pi}} \max \left\{ \mathbb{P}_0 \big( V^{\star,\sigma}(\rho) - V^{\widehat{\pi},\sigma}(\rho) > \varepsilon \big), \, \mathbb{P}_1 \big( V^{\star,\sigma}(\rho) - V^{\widehat{\pi},\sigma}(\rho) > \varepsilon \big) \right\} \geq \frac{1}{8}, \qquad (16)$$

*provided that the total number of samples*

$$NSA \leq c_4 \begin{cases} \frac{SA}{(1-\gamma)^3 \varepsilon^2} & \text{if } \sigma \in \left( 0, \frac{1-\gamma}{4} \right), \\ \frac{\sigma SA}{\min\{1, (1-\gamma)^4 (1+\sigma)^4\} \varepsilon^2} & \text{if } \sigma \in \left[ \frac{1-\gamma}{4}, \infty \right) \end{cases} \qquad (17)$$

*for some universal constant $c_4 > 0$.*

We are now positioned to highlight some key implications of the above theorems about the sample complexity requirements for learning RMDPs with $\chi^2$ divergence.

**Nearly tight sample complexity.** To achieve $\varepsilon$-accuracy for RMDPs with $\chi^2$ distance, Theorem 3 shows that a total number of samples on the order of $\widetilde{O}\left( \frac{SA(1+\sigma)}{(1-\gamma)^4 \varepsilon^2} \right)$ is sufficient for DRVI. Taking it together with the minimax lower bound in Theorem 4 confirms that the sample complexity is near-optimal up to a polynomial factor of the effective horizon $1/(1-\gamma)$ over the entire range of the uncertainty level $\sigma$. In particular, when $\sigma \asymp 1$, our sample complexity $\widetilde{O}\left( \frac{SA}{(1-\gamma)^4 \varepsilon^2} \right)$ is tight and matches with the lower bound; when $\sigma \gtrsim \frac{1}{(1-\gamma)^3}$, our sample complexity correctly predicts the linear dependency with $\sigma$, suggesting that more samples are needed when one plans for larger $\chi^2$-based uncertainty sets.

**RMDPs can be much harder than standard MDPs with $\chi^2$ divergence.** The minimax lower bound developed in Theorem 4 exhibits a surprising non-monotonic behavior of the sample size requirement over the entire range of the uncertainty level $\sigma \in (0, \infty)$ when the uncertainty set is measured via the $\chi^2$ divergence. When $\sigma \lesssim 1 - \gamma$, the lower bound reduces to $\widetilde{O}\left( \frac{SA}{(1-\gamma)^3 \varepsilon^2} \right)$, which matches with that of standard MDPs, as $\sigma = 0$ corresponds to standard MDP. However, two additional regimes are worth calling out:

$$1 - \gamma \lesssim \sigma \lesssim \frac{1}{(1-\gamma)^{1/3}} : \widetilde{O}\left( \frac{SA}{(1-\gamma)^4 \varepsilon^2} \min \left\{ \sigma, \frac{1}{\sigma^3} \right\} \right), \text{ and } \sigma \gtrsim \frac{1}{(1-\gamma)^3} : \ \widetilde{O}\left( \frac{SA\sigma}{\varepsilon^2} \right),$$

both of which are *greater* than that of standard MDPs, indicating learning RMDPs with $\chi^2$ divergence can be much harder.

**Comparison with state-of-the-art bounds.** Our upper bound significantly improves over the prior art $\widetilde{O}\left( \frac{S^2 A(1+\sigma)}{(1-\gamma)^4 \varepsilon^2} \right)$ of Panaganti and Kalathil (2022) by a factor of $S$, and provides the *first* finite-sample complexity that scales *linearly* with respect to $S$ for discounted infinite-horizon RMDPs, which typically exhibit more complicated statistical dependencies than the finite-horizon counterpart. On the other hand, Yang et al. (2022) established a lower bound on the order of $\widetilde{O}\left( \frac{SA}{(1-\gamma)^2 \sigma \varepsilon^2} \right)$ when $\sigma \gtrsim 1 - \gamma$, which is always smaller than the requirement of standard MDPs, and diminishes when $\sigma$ grows. Consequently, Yang et al. (2022) does not lead to the rigorous justification that RMDPs can be harder than standard MDPs, nor the correct linear scaling of the sample size when $\sigma$ grows towards infinity.

# 5    Other related works

**Finite-sample guarantees for standard RL.**    There has been a considerable amount of research into non-asymptotic sample analysis of standard RL for a variety of settings; partial examples include, but are not limited to, the works via probably approximately correct (PAC) bounds for the generative model setting (Kearns and Singh, 1999; Beck and Srikant, 2012; Li et al., 2022a; Chen et al., 2020; Azar et al., 2013; Sidford et al., 2018; Agarwal et al., 2020; Li et al., 2023, 2020; Wainwright, 2019) and the offline setting (Rashidinejad et al., 2021; Xie et al., 2021; Yin et al., 2021; Shi et al., 2022; Li

et al., 2022b; Jin et al., 2021; Yan et al., 2022), as well as the online setting via both regret-based and PAC-base analyses (Jin et al., 2018; Bai et al., 2019; Li et al., 2021; Zhang et al., 2020b; Dong et al., 2019; Jin et al., 2020; Li et al., 2022a; Jafarnia-Jahromi et al., 2020; Yang et al., 2021; Woo et al., 2023).

**Robustness in RL.** To address the challenges of deployed environment uncertainty, an emerging line of works begin to address robustness of RL algorithms with respect to the uncertainty or perturbation over different components of MDPs — state, action, reward, and the transition kernel; see Moos et al. (2022) for a recent review. Besides the framework of distributionally robust MDPs (RMDPs) (Iyengar, 2005) adopted by this work, to promote robustness in RL, there exist various other works including but not limited to Zhang et al. (2020a, 2021); Han et al. (2022); Qiaoben et al. (2021); Sun et al. (2021); Xiong et al. (2022) investigating the robustness w.r.t. state uncertainty. Besides, Tessler et al. (2019); Tan et al. (2020) considered the robustness w.r.t. the uncertainty of the action, and Ding et al. (2023) tackles robustness against spurious correlations.

**Distributionally robust RL.** Rooted in the literature of distributionally robust optimization, which has primarily been investigated in the context of supervised learning (Rahimian and Mehrotra, 2019; Gao, 2020; Bertsimas et al., 2018; Duchi and Namkoong, 2018; Blanchet and Murthy, 2019), distributionally robust dynamic programming and RMDPs have attracted considerable attention recently (Wolff et al., 2012; Kaufman and Schaefer, 2013; Ho et al., 2018; Smirnova et al., 2019; Ho et al., 2021; Goyal and Grand-Clement, 2022; Derman and Mannor, 2020; Tamar et al., 2014). In the context of RMDPs, both empirical and theoretical studies have been widely conducted, although most prior theoretical analyses focus on planning with an exact knowledge of the uncertainty set (Iyengar, 2005; Xu and Mannor, 2012; Tamar et al., 2014), or are asymptotic in nature (Roy et al., 2017).

Resorting to the tools of high-dimensional statistics, various recent works begin to shift attention to understand the finite-sample performance of provable robust RL algorithms, under diverse data generating mechanisms and forms of the uncertainty set over the transition kernel. Besides the infinite-horizon setting, finite-sample complexity bounds for RMDPs with the TV distance and the $\chi^2$ divergence are also developed for the finite-horizon setting in Xu et al. (2023); Dong et al. (2022). In addition, many other forms of uncertainty sets have been considered associated with different divergence function including but not limited to Wasserstein distance, R-contamination, KL divergence, Wang and Zou (2021); Yang et al. (2022); Panaganti and Kalathil (2022); Zhou et al. (2021); Shi and Chi (2022); Xu et al. (2023); Wang et al. (2023a); Blanchet et al. (2023); Liu et al. (2022); Wang et al. (2023c); Liang et al. (2023); Xu et al. (2023); Badrinath and Kalathil (2021); Ramesh et al. (2023); Panaganti et al. (2022); Ma et al. (2022). Moreover, various other related problems or issues have been explored such as the difference of various uncertainty types (Wang et al., 2023b), the iteration complexity of the policy-based methods (Li et al., 2022c; Kumar et al., 2023), the cases when the uncertainty level is instance-dependent small enough (Clavier et al., 2023), and regularization-based robust RL (Yang et al., 2023; Zhang et al., 2023).

# 6  Discussions

This work studies sample complexity bounds for learning RMDPs when the uncertainty set is measured via the TV distance and the $\chi^2$ divergence under the generative model. Our sample complexity bounds not only strengthen the prior art in both directions of upper and lower bounds, but also reveal new insights on how the additional consideration of distributional robustness fundamentally changes the sample complexity of RL in a surprising manner. It turns out that RMDPs are not necessarily harder nor easier than standard MDPs, but the answer is far more nuanced and highly dependent on both the size and shape of the uncertainty set under consideration. These findings could help to guide the practice of RMDPs, by raising awareness that the choice of the uncertainty set not only represents a preference in robustness, but also influences the statistical complexity of the problem.

## Acknowledgments and Disclosure of Funding

The work of L. Shi and Y. Chi is supported in part by the grants ONR N00014-19-1-2404, NSF CCF-2106778, DMS-2134080, and CNS-2148212. L. Shi is also gratefully supported by the Leo Finzi Memorial Fellowship, Wei Shen and Xuehong Zhang Presidential Fellowship, and Liang

Ji-Dian Graduate Fellowship at Carnegie Mellon University. The work of Y. Wei is supported in part by the the NSF grants DMS-2147546/2015447, CAREER award DMS-2143215, CCF-2106778, and the Google Research Scholar Award. The work of Y. Chen is supported in part by the Alfred P. Sloan Research Fellowship, the Google Research Scholar Award, the AFOSR grant FA9550-22-1-0198, the ONR grant N00014-22-1-2354, and the NSF grants CCF-2221009 and CCF-1907661. The authors also acknowledge Zuxin Liu and He Wang for valuable discussions.

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
