# Appendix

## Table of Contents

## A  Other related works

We limit our discussions primarily to provable RL algorithms in the tabular setting with finite state and action spaces, which are most related to our work.

**Finite-sample guarantees for standard RL.** A surge of recent research has utilized the toolkit of concentration inequalities to investigate the performance of standard RL algorithms in non-asymptotic settings. There has been a considerable amount of research into non-asymptotic sample analysis of standard RL for a variety of settings; a small set of samples include, but are not limited to, the works via probably approximately correct (PAC) bounds for the generative model setting (Kearns and Singh, 1999; Beck and Srikant, 2012; **?**; Chen et al., 2020; Azar et al., 2013; Sidford et al., 2018; Agarwal et al., 2020; Li et al., 2023, 2020; Wainwright, 2019), the offline setting (Rashidinejad et al., 2021; Xie et al., 2021; Yin et al., 2021; Shi et al., 2022; Li et al., 2022a; Jin et al., 2021; Yan et al., 2022), and the online setting via regret analysis (Jin et al., 2018; Bai et al., 2019; Li et al., 2021; Zhang et al., 2020b; Dong et al., 2019; Jafarnia-Jahromi et al., 2020; Yang et al., 2021).

**Robustness in RL.** Although standard RL has achieved remarkable success, current RL algorithms are still limited since the agent may fail catastrophically if the deployed environment is subject to perturbation, uncertainty, and even structural changes. To address these challenges, an emerging line of works begin to address robustness of RL algorithms with respect to the uncertainty or perturbation over different components of MDPs — state, action, reward, and the transition kernel; see Moos et al. (2022) for a recent review. Besides the framework of distributionally robust MDPs (RMDPs)

(Iyengar, 2005) adopted by this work, to promote robustness in RL, there exist various other works including but not limited to Zhang et al. (2020a, 2021); Han et al. (2022); Qiaoben et al. (2021); Sun et al. (2021); Xiong et al. (2022) investigating the robustness w.r.t. state uncertainty, where the agent's policy is chosen based on a perturbed observation generated from the state by adding restricted noise or adversarial attack. Besides, Tessler et al. (2019); Tan et al. (2020) considered the robustness to the uncertainty of the action, namely, the action is possibly distorted by an adversarial agent abruptly or smoothly.

**Distributionally robust RL.** Rooted in the literature of distributionally robust optimization, which has primarily been investigated in the context of supervised learning (Rahimian and Mehrotra, 2019; Gao, 2020; Bertsimas et al., 2018; Duchi and Namkoong, 2018; Blanchet and Murthy, 2019), distributionally robust dynamic programming and RMDPs have attracted considerable attention recently (Iyengar, 2005; Xu and Mannor, 2012; Wolff et al., 2012; Kaufman and Schaefer, 2013; Ho et al., 2018; Smirnova et al., 2019; Ho et al., 2021; Goyal and Grand-Clement, 2022; Derman and Mannor, 2020; Tamar et al., 2014; Badrinath and Kalathil, 2021). In the context of RMDPs, both empirical and theoretical studies have been widely conducted, although most prior theoretical analyses focus on planning with an exact knowledge of the uncertainty set (Iyengar, 2005; Xu and Mannor, 2012; Tamar et al., 2014), or are asymptotic in nature (Roy et al., 2017).

Resorting to the tools of high-dimensional statistics, various recent works begin to shift attention to understand the finite-sample performance of provable robust RL algorithms, under diverse data generating mechanisms and forms of the uncertainty set over the transition kernel. Besides the infinite-horizon setting, finite-sample complexity bounds for RMDPs with the TV distance and the $\chi^2$ divergence are also developed for the finite-horizon setting in Xu et al. (2023); Dong et al. (2022). In addition, many other forms of uncertainty sets have been considered. For example, Wang and Zou (2021) considered a R-contamination uncertain set and proposed a provable robust Q-learning algorithm for the online setting with similar guarantees as standard MDPs. The KL divergence is another popular choice widely considered, where Yang et al. (2022); Panaganti and Kalathil (2022); Zhou et al. (2021); Shi and Chi (2022); Xu et al. (2023); Wang et al. (2023); **?** investigated the sample complexity of both model-based and model-free algorithms under the simulator or offline settings. Xu et al. (2023) considered a variety of uncertainty sets including one associated with Wasserstein distance. Badrinath and Kalathil (2021) considered a general $(s, a)$-rectangular form of the uncertainty set and proposed a model-free algorithm for the online setting with linear function approximation to cope with large state spaces. Moreover, various other related issues have been explored such as the iteration complexity of the policy-based methods (Li et al., 2022b; Kumar et al., 2023), and regularization-based robust RL (Yang et al., 2023).

# B  Preliminaries

For convenience, we introduce the notation $[T] \coloneqq \{1, \cdots, T\}$ for any positive integer $T > 0$. Moreover, for any two vectors $x = [x_i]_{1 \le i \le n}$ and $y = [y_i]_{1 \le i \le n}$, the notation $x \le y$ (resp. $x \ge y$) means $x_i \le y_i$ (resp. $x_i \ge y_i$) for all $1 \le i \le n$. And for any vecvor $x$, we overload the notation by letting $x^{\circ 2} = \left[x(s, a)^2\right]_{(s,a) \in \mathcal{S} \times \mathcal{A}}$ (resp. $x^{\circ 2} = \left[x(s)^2\right]_{s \in \mathcal{S}}$). With slight abuse of notation, we denote 0 (resp. 1) as the all-zero (resp. all-one) vector, and drop the subscript $\rho$ to write $\mathcal{U}^\sigma(\cdot) = \mathcal{U}_\rho^\sigma(\cdot)$ whenever the argument holds for all divergence $\rho$.

**Matrix notation.** To continue, we recall or introduce some additional matrix notation that is useful throughout the analysis.

- $P^0 \in \mathbb{R}^{SA \times S}$: the matrix of the nominal transition kernel with $P_{s,a}^0$ as the $(s, a)$-th row.

- $\widehat{P}^0 \in \mathbb{R}^{SA \times S}$: the matrix of the estimated nomimal transition kernel with $\widehat{P}_{s,a}^0$ as the $(s, a)$-th row.

- $r \in \mathbb{R}^{SA}$: a vector representing the reward function $r$ (so that $r_{(s,a)} = r(s, a)$ for all $(s, a) \in \mathcal{S} \times \mathcal{A}$).

- $\Pi^\pi \in \{0,1\}^{S \times SA}$: a projection matrix associated with a given deterministic policy $\pi$ taking the following form

$$\Pi^\pi = \begin{pmatrix} \boldsymbol{e}_{\pi(1)}^\top & 0^\top & \cdots & 0^\top \\ 0^\top & \boldsymbol{e}_{\pi(2)}^\top & \cdots & 0^\top \\ \vdots & \vdots & \ddots & \vdots \\ 0^\top & 0^\top & \cdots & \boldsymbol{e}_{\pi(S)}^\top \end{pmatrix}, \tag{18}$$

where $\boldsymbol{e}_{\pi(1)}^\top, \boldsymbol{e}_{\pi(2)}^\top, \ldots, \boldsymbol{e}_{\pi(S)}^\top \in \mathbb{R}^A$ are standard basis vectors.

- $r_\pi \in \mathbb{R}^S$: a reward vector restricted to the actions chosen by the policy $\pi$, namely, $r_\pi(s) = r(s, \pi(s))$ for all $s \in \mathcal{S}$ (or simply, $r_\pi = \Pi^\pi r$).

- $\mathsf{Var}_P(V) \in \mathbb{R}^{SA}$: for any transition kernel $P \in \mathbb{R}^{SA \times S}$ and vector $V \in \mathbb{R}^S$, we denote the $(s,a)$-th row of $\mathsf{Var}_P(V)$ as

$$\mathsf{Var}_P(s,a) := \mathsf{Var}_{P_{s,a}}(V). \tag{19}$$

- $P^V \in \mathbb{R}^{SA \times S}$, $\widehat{P}^V \in \mathbb{R}^{SA \times S}$: the matrices representing the probability transition kernel in the uncertainty set that leads to the worst-case value for any vector $V \in \mathbb{R}^S$. We denote $P_{s,a}^V$ (resp. $\widehat{P}_{s,a}^V$) as the $(s,a)$-th row of the transition matrix $P^V$ (resp. $\widehat{P}^V$). In truth, the $(s,a)$-th rows of these transition matrices are defined as

$$P_{s,a}^V = \mathrm{argmin}_{\mathcal{P} \in \mathcal{U}^\sigma(P_{s,a}^0)} \mathcal{P}V, \qquad \text{and} \qquad \widehat{P}_{s,a}^V = \mathrm{argmin}_{\mathcal{P} \in \mathcal{U}^\sigma(\widehat{P}_{s,a}^0)} \mathcal{P}V. \tag{20a}$$

Furthermore, we make use of the following short-hand notation:

$$P_{s,a}^{\pi,V} := P_{s,a}^{V^{\pi,\sigma}} = \mathrm{argmin}_{\mathcal{P} \in \mathcal{U}^\sigma(P_{s,a}^0)} \mathcal{P}V^{\pi,\sigma},$$

$$P_{s,a}^{\pi,\widehat{V}} := P_{s,a}^{\widehat{V}^{\pi,\sigma}} = \mathrm{argmin}_{\mathcal{P} \in \mathcal{U}^\sigma(P_{s,a}^0)} \mathcal{P}\widehat{V}^{\pi,\sigma}, \tag{20b}$$

$$\widehat{P}_{s,a}^{\pi,V} := \widehat{P}_{s,a}^{V^{\pi,\sigma}} = \mathrm{argmin}_{P \in \mathcal{U}^\sigma(\widehat{P}_{s,a}^0)} P V^{\pi,\sigma},$$

$$\widehat{P}_{s,a}^{\pi,\widehat{V}} := \widehat{P}_{s,a}^{\widehat{V}^{\pi,\sigma}} = \mathrm{argmin}_{P \in \mathcal{U}^\sigma(\widehat{P}_{s,a}^0)} P \widehat{V}^{\pi,\sigma}. \tag{20c}$$

The corresponding probability transition matrices are denoted by $P^{\pi,V} \in \mathbb{R}^{SA \times S}$, $P^{\pi,\widehat{V}} \in \mathbb{R}^{SA \times S}$, $\widehat{P}^{\pi,V} \in \mathbb{R}^{SA \times S}$ and $\widehat{P}^{\pi,\widehat{V}} \in \mathbb{R}^{SA \times S}$, respectively.

- $P^\pi \in \mathbb{R}^{S \times S}$, $\widehat{P}^\pi \in \mathbb{R}^{S \times S}$, $\underline{P}^{\pi,V} \in \mathbb{R}^{S \times S}$, $\underline{P}^{\pi,\widehat{V}} \in \mathbb{R}^{S \times S}$, $\underline{\widehat{P}}^{\pi,V} \in \mathbb{R}^{S \times S}$ and $\underline{\widehat{P}}^{\pi,\widehat{V}} \in \mathbb{R}^{S \times S}$: six *square* probability transition matrices w.r.t. policy $\pi$ over the states, namely

$$P^\pi := \Pi^\pi P^0, \qquad \widehat{P}^\pi := \Pi^\pi \widehat{P}^0, \qquad \underline{P}^{\pi,V} := \Pi^\pi P^{\pi,V}, \qquad \underline{P}^{\pi,\widehat{V}} := \Pi^\pi P^{\pi,\widehat{V}},$$

$$\underline{\widehat{P}}^{\pi,V} := \Pi^\pi \widehat{P}^{\pi,V}, \qquad \text{and} \qquad \underline{\widehat{P}}^{\pi,\widehat{V}} := \Pi^\pi \widehat{P}^{\pi,\widehat{V}}. \tag{21}$$

We denote $P_s^\pi$ as the $s$-th row of the transition matrix $P^\pi$; similar quantities can be defined for the other matrices as well.

## B.1 Basic facts

**Kullback-Leibler (KL) divergence.** First, for any two distributions $P$ and $Q$, we denote by $\mathsf{KL}(P \parallel Q)$ the Kullback-Leibler (KL) divergence of $P$ and $Q$. Letting $\mathsf{Ber}(p)$ be the Bernoulli distribution with mean $p$, we also introduce

$$\mathsf{KL}(p \parallel q) := p \log \frac{p}{q} + (1-p) \log \frac{1-p}{1-q} \quad \text{and} \quad \chi^2(p \parallel q) := \frac{(p-q)^2}{q} + \frac{(p-q)^2}{1-q} = \frac{(p-q)^2}{q(1-q)}, \tag{22}$$

which represent respectively the KL divergence and the $\chi^2$ divergence of $\mathsf{Ber}(p)$ from $\mathsf{Ber}(q)$ (Tsybakov and Zaiats, 2009). We make note of the following useful property about the KL divergence in Tsybakov and Zaiats (2009, Lemma 2.7).

**Lemma 1.** *For any $p, q \in (0,1)$, it holds that*

$$\mathsf{KL}(p \parallel q) \le \frac{(p-q)^2}{q(1-q)}. \tag{23}$$

**Variance.** For any probability vector $P \in \mathbb{R}^{1 \times S}$ and vector $V \in \mathbb{R}^S$, we denote the variance

$$\mathrm{Var}_P(V) := P(V \circ V) - (PV) \circ (PV). \tag{24}$$

The following lemma bounds the Lipschitz constant of the variance function.

**Lemma 2.** *Consider any* $0 \leq V_1, V_2 \leq \frac{1}{1-\gamma}$ *obeying* $\|V_1 - V_2\|_\infty \leq x$ *and any probability vector* $P \in \Delta(S)$*, one has*

$$|\mathrm{Var}_P(V_1) - \mathrm{Var}_P(V_2)| \leq \frac{2x}{(1-\gamma)}. \tag{25}$$

*Proof.* It is immediate to check that

$$
\begin{aligned}
|\mathrm{Var}_P(V_1) - \mathrm{Var}_P(V_2)| &= |P(V_1 \circ V_1) - (PV_1) \circ (PV_1) - P(V_2 \circ V_2) + (PV_2) \circ (PV_2)| \\
&\leq \left|P\big(V_1 \circ V_1 - V_2 \circ V_2\big)\right| + |(PV_1 + PV_2)P(V_1 - V_2)| \\
&\leq 2\|V_1 + V_2\|_\infty \|V_1 - V_2\|_\infty \leq \frac{2x}{(1-\gamma)}. 
\end{aligned}
\tag{26}
$$

where the penultimate inequality holds by the triangle inequality. $\square$

## B.2 Properties of the robust Bellman operator

**$\gamma$-contraction of the robust Bellman operator.** It is worth noting that the robust Bellman operator (cf. (6)) shares the nice $\gamma$-contraction property of the standard Bellman operator, stated as below.

**Lemma 3** ($\gamma$-Contraction). *(Iyengar, 2005, Theorem 3.2) For any* $\gamma \in [0, 1)$*, the robust Bellman operator* $\mathcal{T}^\sigma(\cdot)$ *(cf. (6)) is a* $\gamma$*-contraction w.r.t.* $\|\cdot\|_\infty$*. Namely, for any* $Q_1, Q_2 \in \mathbb{R}^{SA}$ *s.t.* $Q_1(s,a), Q_2(s,a) \in \left[0, \frac{1}{1-\gamma}\right]$ *for all* $(s,a) \in \mathcal{S} \times \mathcal{A}$*, one has*

$$\|\mathcal{T}^\sigma(Q_1) - \mathcal{T}^\sigma(Q_2)\|_\infty \leq \gamma \|Q_1 - Q_2\|_\infty. \tag{27}$$

*Additionally,* $Q^{\star,\sigma}$ *is the unique fixed point of* $\mathcal{T}^\sigma(\cdot)$ *obeying* $0 \leq Q^{\star,\sigma}(s,a) \leq \frac{1}{1-\gamma}$ *for all* $(s,a) \in \mathcal{S} \times \mathcal{A}$*.*

**Dual equivalence of the robust Bellman operator.** Fortunately, the robust Bellman operator can be evaluated efficiently by resorting to its dual formulation (Iyengar, 2005). In what follows, we shall illustrate this for the two choices of the divergence $\rho$ of interest. Before continuing, for any $V \in \mathbb{R}^S$, we denote $[V]_\alpha$ as its clipped version by some non-negative value $\alpha$, namely,

$$[V]_\alpha(s) := \begin{cases} \alpha, & \text{if } V(s) > \alpha, \\ V(s), & \text{otherwise.} \end{cases} \tag{28}$$

- TV distance, where the uncertainty set is $\mathcal{U}_\rho^\sigma(\widehat{P}_{s,a}^0) := \mathcal{U}_{\mathsf{TV}}^\sigma(\widehat{P}_{s,a}^0) := \mathcal{U}_{\rho_{\mathsf{TV}}}^\sigma(\widehat{P}_{s,a}^0)$ w.r.t. the TV distance $\rho = \rho_{\mathsf{TV}}$ defined in (7). In particular, we have the following lemma due to strong duality, which is a direct consequence of Iyengar (2005, Lemma 4.3).

  **Lemma 4** (Strong duality for TV). *Consider any probability vector* $P \in \Delta(\mathcal{S})$*, any fixed uncertainty level* $\sigma$ *and the uncertainty set* $\mathcal{U}^\sigma(P) := \mathcal{U}_{\mathsf{TV}}^\sigma(P)$*. For any vector* $V \in \mathbb{R}^S$ *obeying* $V \geq 0$*, recalling the definition of* $[V]_\alpha$ *in (28), one has*

  $$\inf_{\mathcal{P} \in \mathcal{U}^\sigma(P)} \mathcal{P}V = \max_{\alpha \in [\min_s V(s), \max_s V(s)]} \left\{ P[V]_\alpha - \sigma\left(\alpha - \min_{s'}[V]_\alpha(s')\right) \right\}. \tag{29}$$

  In view of the above lemma, the following dual update rule is equivalent to (13) in DRVI:

  $$
  \begin{aligned}
  \widehat{Q}_t(s,a) &= r(s,a) \\
  &\quad + \gamma \max_{\alpha \in \left[\min_s \widehat{V}_{t-1}(s), \max_s \widehat{V}_{t-1}(s)\right]} \left\{ \widehat{P}_{s,a}^0\left[\widehat{V}_{t-1}\right]_\alpha - \sigma\left(\alpha - \min_{s'}\left[\widehat{V}_{t-1}\right]_\alpha(s')\right) \right\}. 
  \end{aligned}
  \tag{30}
  $$

- $\chi^2$ divergence, where the uncertainty set is $\mathcal{U}_\rho^\sigma(\widehat{P}_{s,a}^0) := \mathcal{U}_{\chi^2}^\sigma(\widehat{P}_{s,a}^0) := \mathcal{U}_{\rho_{\chi^2}}^\sigma(\widehat{P}_{s,a}^0)$ w.r.t. the $\chi^2$ divergence $\rho = \rho_{\chi^2}$ defined in (8). We introduce the following lemma which directly follows from (Iyengar, 2005, Lemma 4.2).

**Lemma 5** (Strong duality for $\chi^2$). *Consider any probability vector $P \in \Delta(\mathcal{S})$, any fixed uncertainty level $\sigma$ and the uncertainty set $\mathcal{U}^\sigma(P) := \mathcal{U}^\sigma_{\chi^2}(P)$. For any vector $V \in \mathbb{R}^S$ obeying $V \geq 0$, one has*

$$\inf_{\mathcal{P} \in \mathcal{U}^\sigma(P)} \mathcal{P}V = \max_{\alpha \in [\min_s V(s), \max_s V(s)]} \left\{ P[V]_\alpha - \sqrt{\sigma \mathsf{Var}_P\left([V]_\alpha\right)} \right\}, \qquad (31)$$

*where $\mathsf{Var}_P(\cdot)$ is defined as (24).*

In view of the above lemma, the update rule (13) in DRVI can be equivalently written as:

$$\widehat{Q}_t(s,a) = r(s,a)$$

$$+ \gamma \max_{\alpha \in \left[\min_s \widehat{V}_{t-1}(s), \max_s \widehat{V}_{t-1}(s)\right]} \left\{ \widehat{P}^0_{s,a} \left[\widehat{V}_{t-1}\right]_\alpha - \sqrt{\sigma \mathsf{Var}_{\widehat{P}^0_{s,a}}\left(\left[\widehat{V}_{t-1}\right]_\alpha\right)} \right\}. \quad (32)$$

The proofs of Lemma 4 and Lemma 5 are provided as follows.

*Proof of Lemma 4.* To begin with, applying (Iyengar, 2005, Lemma 4.3), the term of interest obeys

$$\inf_{\mathcal{P} \in \mathcal{U}^\sigma(P)} \mathcal{P}V = \max_{\mu \in \mathbb{R}^S, \mu \geq 0} \left\{ P(V - \mu) - \sigma \left( \max_{s'} \{V(s') - \mu(s')\} - \min_{s'} \{V(s') - \mu(s')\} \right) \right\}, \tag{33}$$

where $\mu(s')$ represents the $s'$-th entry of $\mu \in \mathbb{R}^S$. Denoting $\mu^\star$ as the optimal dual solution, taking $\alpha = \max_{s'} \{V(s') - \mu^\star(s')\}$, it is easily verified that $\mu^\star$ obeys

$$\mu^\star(s) = \begin{cases} V(s) - \alpha, & \text{if } V(s) > \alpha \\ 0, & \text{otherwise.} \end{cases} \tag{34}$$

Therefore, (33) can be solved by optimizing $\alpha$ as below (Iyengar, 2005, Lemma 4.3):

$$\inf_{\mathcal{P} \in \mathcal{U}^\sigma(P)} \mathcal{P}V = \max_{\alpha \in [\min_s V(s), \max_s V(s)]} \left\{ P[V]_\alpha - \sigma \left( \alpha - \min_{s'} [V]_\alpha(s') \right) \right\}. \tag{35}$$

$\square$

*Proof of Lemma 5.* Due to strong duality (Iyengar, 2005, Lemma 4.2), it holds that

$$\inf_{\mathcal{P} \in \mathcal{U}^\sigma(P)} \mathcal{P}V = \max_{\mu \in \mathbb{R}^S, \mu \geq 0} \left\{ P(V - \mu) - \sqrt{\sigma \mathsf{Var}_P(V - \mu)} \right\}, \tag{36}$$

and the optimal $\mu^\star$ obeys

$$\mu^\star(s) = \begin{cases} V(s) - \alpha, & \text{if } V(s) > \alpha \\ 0, & \text{otherwise.} \end{cases} \tag{37}$$

for some $\alpha \in [\min_s V(s), \max_s V(s)]$. As a result, solving (36) is equivalent to optimizing the scalar $\alpha$ as below:

$$\inf_{\mathcal{P} \in \mathcal{U}^\sigma(P)} \mathcal{P}V = \max_{\alpha \in [\min_s V(s), \max_s V(s)]} \left\{ P[V]_\alpha - \sqrt{\sigma \mathsf{Var}_P\left([V]_\alpha\right)} \right\}. \tag{38}$$

$\square$

## B.3 Additional facts of the empirical robust MDP

**Bellman equations of the empirical robust MDP $\widehat{\mathcal{M}}_{\mathsf{rob}}$.** To begin with, recall that the empirical robust MDP $\widehat{\mathcal{M}}_{\mathsf{rob}} = \{\mathcal{S}, \mathcal{A}, \gamma, \mathcal{U}^\sigma(\widehat{P}^0), r\}$ based on the estimated nominal distribution $\widehat{P}^0$ constructed in (10) and its corresponding robust value function (resp. robust Q-function) $\widehat{V}^{\pi,\sigma}$ (resp. $\widehat{Q}^{\pi,\sigma}$).

Note that $\widehat{Q}^{\star,\sigma}$ is the unique fixed point of $\widehat{\mathcal{T}}^\sigma(\cdot)$ (see Lemma 3), the empirical robust Bellman operator constructed using $\widehat{P}^0$. Moreover, similar to (**??**), for $\widehat{\mathcal{M}}_{\mathsf{rob}}$, the Bellman's optimality

principle gives the following *robust Bellman consistency equation* (resp. *robust Bellman optimality equation*):

$$\forall (s,a) \in \mathcal{S} \times \mathcal{A}: \quad \widehat{Q}^{\pi,\sigma}(s,a) = r(s,a) + \gamma \inf_{\mathcal{P} \in \mathcal{U}^\sigma(\widehat{P}^0_{s,a})} \mathcal{P}\widehat{V}^{\pi,\sigma}, \tag{39a}$$

$$\forall (s,a) \in \mathcal{S} \times \mathcal{A}: \quad \widehat{Q}^{\star,\sigma}(s,a) = r(s,a) + \gamma \inf_{\mathcal{P} \in \mathcal{U}^\sigma(\widehat{P}^0_{s,a})} \mathcal{P}\widehat{V}^{\star,\sigma}. \tag{39b}$$

With these in mind, combined with the matrix notation, for any policy $\pi$, we can write the robust Bellman consistency equations as

$$Q^{\pi,\sigma} = r + \gamma \inf_{\mathcal{P} \in \mathcal{U}^\sigma(P^0)} \mathcal{P}V^{\pi,\sigma} \quad \text{and} \quad \widehat{Q}^{\pi,\sigma} = r + \gamma \inf_{\mathcal{P} \in \mathcal{U}^\sigma(\widehat{P}^0)} \mathcal{P}\widehat{V}^{\pi,\sigma}, \tag{40}$$

which leads to

$$V^{\pi,\sigma} = r_\pi + \gamma \Pi^\pi \inf_{\mathcal{P} \in \mathcal{U}^\sigma(P^0)} \mathcal{P}V^{\pi,\sigma} \overset{\text{(i)}}{=} r_\pi + \gamma \underline{P}^{\pi,V} V^{\pi,\sigma},$$

$$\widehat{V}^{\pi,\sigma} = r_\pi + \gamma \Pi^\pi \inf_{\mathcal{P} \in \mathcal{U}^\sigma(\widehat{P}^0)} \mathcal{P}\widehat{V}^{\pi,\sigma} \overset{\text{(ii)}}{=} r_\pi + \gamma \underline{\widehat{P}}^{\pi,\widehat{V}} \widehat{V}^{\pi,\sigma}, \tag{41}$$

where (i) and (ii) holds by the definitions in (18), (20) and (21).

Encouragingly, the above property of the robust Bellman operator ensures the fast convergence of DRVI. We collect this consequence in the following lemma.

**Lemma 6.** *Let $\widehat{Q}_0 = 0$. The iterates $\{\widehat{Q}_t\}, \{\widehat{V}_t\}$ of* DRVI *obey*

$$\forall t \geq 0: \quad \left\| \widehat{Q}_t - \widehat{Q}^{\star,\sigma} \right\|_\infty \leq \frac{\gamma^t}{1-\gamma} \quad \text{and} \quad \left\| \widehat{V}_t - \widehat{V}^{\star,\sigma} \right\|_\infty \leq \frac{\gamma^t}{1-\gamma}. \tag{42}$$

*Furthermore, the output policy $\widehat{\pi}$ obeys*

$$\left\| \widehat{V}^{\star,\sigma} - \widehat{V}^{\widehat{\pi},\sigma} \right\|_\infty \leq \frac{2\gamma \varepsilon_{\mathsf{opt}}}{1-\gamma}, \quad \text{where} \quad \left\| \widehat{V}^{\star,\sigma} - \widehat{V}_{T-1} \right\|_\infty =: \varepsilon_{\mathsf{opt}}. \tag{43}$$

*Proof of Lemma 6.* Applying the $\gamma$-contraction property in Lemma 3 directly yields that for any $t \geq 0$,

$$\|\widehat{Q}_t - \widehat{Q}^{\star,\sigma}\|_\infty = \left\| \widehat{\mathcal{T}}^\sigma(\widehat{Q}_{t-1}) - \widehat{\mathcal{T}}^\sigma(\widehat{Q}^{\star,\sigma}) \right\|_\infty \leq \gamma \|\widehat{Q}_{t-1} - \widehat{Q}^{\star,\sigma}\|_\infty$$

$$\leq \cdots \leq \gamma^t \|\widehat{Q}_0 - \widehat{Q}^{\star,\sigma}\|_\infty = \gamma^t \|\widehat{Q}^{\star,\sigma}\|_\infty \leq \frac{\gamma^t}{1-\gamma},$$

where the last inequality holds by the fact $\|\widehat{Q}^{\star,\sigma}\|_\infty \leq \frac{1}{1-\gamma}$ (see Lemma 3). In addition,

$$\|\widehat{V}_t - \widehat{V}^{\star,\sigma}\|_\infty = \max_{s \in \mathcal{S}} \left\| \max_{a \in \mathcal{A}} \widehat{Q}_t(s,a) - \max_{a \in \mathcal{A}} \widehat{Q}^{\star,\sigma}(s,a) \right\|_\infty \leq \|\widehat{Q}_t - \widehat{Q}^{\star,\sigma}\|_\infty \leq \frac{\gamma^t}{1-\gamma},$$

where the penultimate inequality holds by the maximum operator is 1-Lipschitz. This completes the proof of (42).

We now move to establish (43). Note that there exists at least one state $s_0 \in \mathcal{S}$ that is associated with the maximum of the value gap, i.e.,

$$\left\| \widehat{V}^{\star,\sigma} - \widehat{V}^{\widehat{\pi},\sigma} \right\|_\infty = \widehat{V}^{\star,\sigma}(s_0) - \widehat{V}^{\widehat{\pi},\sigma}(s_0) \geq \widehat{V}^{\star,\sigma}(s) - \widehat{V}^{\widehat{\pi},\sigma}(s), \quad \forall s \in \mathcal{S}.$$

Recall $\widehat{\pi}^\star$ is the optimal robust policy for the empirical RMDP $\widehat{\mathcal{M}}_{\mathsf{rob}}$. For convenience, we denote $a_1 = \widehat{\pi}^\star(s_0)$ and $a_2 = \widehat{\pi}(s_0)$. Then, since $\widehat{\pi}$ is the greedy policy w.r.t. $\widehat{Q}_T$, one has

$$r(s_0, a_1) + \gamma \inf_{\mathcal{P} \in \mathcal{U}^\sigma(\widehat{P}^0_{s_0,a_1})} \mathcal{P}\widehat{V}_{T-1} = \widehat{Q}_T(s_0, a_1) \leq \widehat{Q}_T(s_0, a_2) = r(s_0, a_2) + \gamma \inf_{\mathcal{P} \in \mathcal{U}^\sigma(\widehat{P}^0_{s_0,a_2})} \mathcal{P}\widehat{V}_{T-1}. \tag{44}$$

Recalling the notation in (20), the above fact and (43) altogether yield

$$r(s_0, a_1) + \gamma \widehat{P}_{s_0, a_1}^{\widehat{V}_{T-1}} \left( \widehat{V}^{\star, \sigma} - \varepsilon_{\mathsf{opt}} 1 \right) \leq r(s_0, a_1) + \gamma \widehat{P}_{s_0, a_1}^{\widehat{V}_{T-1}} \widehat{V}_{T-1}$$

$$\leq r(s_0, a_2) + \gamma \inf_{\mathcal{P} \in \mathcal{U}^\sigma(\widehat{P}_{s_0, a_2}^0)} \mathcal{P} \widehat{V}_{T-1}$$

$$\overset{(i)}{\leq} r(s_0, a_2) + \gamma \widehat{P}_{s_0, a_2}^{\widehat{V}^{\widehat{\pi}, \sigma}} \widehat{V}_{T-1}$$

$$\leq r(s_0, a_2) + \gamma \widehat{P}_{s_0, a_2}^{\widehat{V}^{\widehat{\pi}, \sigma}} \left( \widehat{V}^{\star, \sigma} + \varepsilon_{\mathsf{opt}} 1 \right), \qquad (45)$$

where (i) follows from the optimality criteria. The term of interest can be controlled as

$$\left\| \widehat{V}^{\star, \sigma} - \widehat{V}^{\widehat{\pi}, \sigma} \right\|_\infty = \widehat{V}^{\star, \sigma}(s_0) - \widehat{V}^{\widehat{\pi}, \sigma}(s_0)$$

$$= r(s_0, a_1) + \gamma \inf_{\mathcal{P} \in \mathcal{U}^\sigma(\widehat{P}_{s_0, a_1}^0)} \mathcal{P} \widehat{V}^{\star, \sigma} - \left( r(s_0, a_2) + \gamma \inf_{\mathcal{P} \in \mathcal{U}^\sigma(\widehat{P}_{s_0, a_2}^0)} \mathcal{P} \widehat{V}^{\widehat{\pi}, \sigma} \right)$$

$$= r(s_0, a_1) - r(s_0, a_2) + \gamma \left( \inf_{\mathcal{P} \in \mathcal{U}^\sigma(\widehat{P}_{s_0, a_1}^0)} \mathcal{P} \widehat{V}^{\star, \sigma} - \inf_{\mathcal{P} \in \mathcal{U}^\sigma(\widehat{P}_{s_0, a_2}^0)} \mathcal{P} \widehat{V}^{\widehat{\pi}, \sigma} \right)$$

$$\overset{(i)}{\leq} 2\gamma \varepsilon_{\mathsf{opt}} + \gamma \left( \widehat{P}_{s_0, a_2}^{\widehat{V}^{\widehat{\pi}, \sigma}} \widehat{V}^{\star, \sigma} - \widehat{P}_{s_0, a_1}^{\widehat{V}_{T-1}} \widehat{V}^{\star, \sigma} + \inf_{\mathcal{P} \in \mathcal{U}^\sigma(\widehat{P}_{s_0, a_1}^0)} \mathcal{P} \widehat{V}^{\star, \sigma} - \inf_{\mathcal{P} \in \mathcal{U}^\sigma(\widehat{P}_{s_0, a_2}^0)} \mathcal{P} \widehat{V}^{\widehat{\pi}, \sigma} \right)$$

$$= 2\gamma \varepsilon_{\mathsf{opt}} + \gamma \left( \widehat{P}_{s_0, a_2}^{\widehat{V}^{\widehat{\pi}, \sigma}} \widehat{V}^{\star, \sigma} - \inf_{\mathcal{P} \in \mathcal{U}^\sigma(\widehat{P}_{s_0, a_2}^0)} \mathcal{P} \widehat{V}^{\widehat{\pi}, \sigma} \right) + \gamma \left( \inf_{\mathcal{P} \in \mathcal{U}^\sigma(\widehat{P}_{s_0, a_1}^0)} \mathcal{P} \widehat{V}^{\star, \sigma} - \widehat{P}_{s_0, a_1}^{\widehat{V}_{T-1}} \widehat{V}^{\star, \sigma} \right)$$

$$\overset{(ii)}{\leq} 2\gamma \varepsilon_{\mathsf{opt}} + \gamma \widehat{P}_{s_0, a_2}^{\widehat{V}^{\widehat{\pi}, \sigma}} \left( \widehat{V}^{\star, \sigma} - \widehat{V}^{\widehat{\pi}, \sigma} \right) + \gamma \left( \widehat{P}_{s_0, a_1}^{\widehat{V}_{T-1}} \widehat{V}^{\star, \sigma} - \widehat{P}_{s_0, a_1}^{\widehat{V}_{T-1}} \widehat{V}^{\star, \sigma} \right)$$

$$\leq 2\gamma \varepsilon_{\mathsf{opt}} + \gamma \left\| \widehat{V}^{\star, \sigma} - \widehat{V}^{\widehat{\pi}, \sigma} \right\|_\infty, \qquad (46)$$

where (i) holds by plugging in (45), and (ii) follows from $\inf_{\mathcal{P} \in \mathcal{U}^\sigma(\widehat{P}_{s_0, a_1}^0)} \mathcal{P} \widehat{V}^{\star, \sigma} \leq \mathcal{P} \widehat{V}^{\star, \sigma}$ for any $\mathcal{P} \in \mathcal{U}^\sigma(\widehat{P}_{s_0, a_1}^0)$. Rearranging (46) leads to

$$\left\| \widehat{V}^{\star, \sigma} - \widehat{V}^{\widehat{\pi}, \sigma} \right\|_\infty \leq \frac{2\gamma \varepsilon_{\mathsf{opt}}}{1 - \gamma}.$$

$\square$

# C  Proof of the upper bound with TV distance: Theorem 1

Throughout this section, for any transition kernel $P$, the uncertainty set is taken as (see (7))

$$\mathcal{U}^\sigma(P) := \mathcal{U}_{\mathsf{TV}}^\sigma(P) = \otimes \, \mathcal{U}_{\mathsf{TV}}^\sigma(P_{s,a}), \qquad \mathcal{U}_{\mathsf{TV}}^\sigma(P_{s,a}) := \left\{ P_{s,a}' \in \Delta(\mathcal{S}) : \frac{1}{2} \left\| P_{s,a}' - P_{s,a} \right\|_1 \leq \sigma \right\}. \tag{47}$$

## C.1  Technical lemmas

We begin with a key lemma concerning the dynamic range of the robust value function $V^{\pi, \sigma}$ (cf. (??)), which produces tighter control when $\sigma$ is large; the proof is deferred to Appendix C.3.1.

**Lemma 7.** *For any nominal transition kernel $P \in \mathbb{R}^{SA \times S}$, any fixed uncertainty level $\sigma$, and any policy $\pi$, its corresponding robust value function $V^{\pi, \sigma}$ (cf. (??)) satisfies*

$$\max_{s \in \mathcal{S}} V^{\pi, \sigma}(s) - \min_{s \in \mathcal{S}} V^{\pi, \sigma}(s) \leq \frac{1}{\gamma \max\{1 - \gamma, \sigma\}}.$$

Next, we introduce the following lemma, whose proof is postponed in Appendix C.3.2.

**Lemma 8.** *Consider an MDP with transition kernel matrix $P$ and reward function $0 \leq r \leq 1$. For any policy $\pi$ and its associated state transition matrix $P_\pi := \Pi^\pi P$ and value function $0 \leq V^{\pi,P} \leq \frac{1}{1-\gamma}$ (cf. (1)), one has*

$$(I - \gamma P_\pi)^{-1} \sqrt{\mathrm{Var}_{P_\pi}(V^{\pi,P})} \leq \sqrt{\frac{8(\max_s V^{\pi,P}(s) - \min_s V^{\pi,P}(s))}{\gamma^2(1-\gamma)^2}} \mathbf{1}.$$

## C.2  Proof of Theorem 1

The main proof idea of Theorem 1 is similar to that of Agarwal et al. (2020) and Li et al. (2020) while the argument needs essential adjustments in order to adapt to the robustness setting. Before proceeding, applying Lemma 6 yields that for any $\varepsilon_{\mathsf{opt}} > 0$, as long as $T \geq \log(\frac{1}{(1-\gamma)\varepsilon_{\mathsf{opt}}})$, one has

$$\left\| \widehat{V}^{\star,\sigma} - \widehat{V}^{\widehat{\pi},\sigma} \right\|_\infty \leq \frac{2\gamma\varepsilon_{\mathsf{opt}}}{1-\gamma}, \tag{48}$$

allowing us to justify the more general statement in Remark **??**. To control the performance gap $\left\| V^{\star,\sigma} - V^{\widehat{\pi},\sigma} \right\|_\infty$, the proof is divided into several key steps.

**Step 1: decomposing the error.**  Recall the optimal robust policy $\pi^\star$ w.r.t. $\mathcal{M}_{\mathsf{rob}}$ and the optimal robust policy $\widehat{\pi}^\star$, the optimal robust value function $\widehat{V}^{\star,\sigma}$ (resp. robust value function $\widehat{Q}^{\pi,\sigma}$) w.r.t. $\widehat{\mathcal{M}}_{\mathsf{rob}}$. The term of interest $V^{\star,\sigma} - V^{\widehat{\pi},\sigma}$ can be decomposed as

$$V^{\star,\sigma} - V^{\widehat{\pi},\sigma} = \left( V^{\pi^\star,\sigma} - \widehat{V}^{\pi^\star,\sigma} \right) + \left( \widehat{V}^{\pi^\star,\sigma} - \widehat{V}^{\widehat{\pi}^\star,\sigma} \right) + \left( \widehat{V}^{\widehat{\pi}^\star,\sigma} - \widehat{V}^{\widehat{\pi},\sigma} \right) + \left( \widehat{V}^{\widehat{\pi},\sigma} - V^{\widehat{\pi},\sigma} \right)$$

$$\overset{(i)}{\leq} \left( V^{\pi^\star,\sigma} - \widehat{V}^{\pi^\star,\sigma} \right) + \left( \widehat{V}^{\widehat{\pi}^\star,\sigma} - \widehat{V}^{\widehat{\pi},\sigma} \right) + \left( \widehat{V}^{\widehat{\pi},\sigma} - V^{\widehat{\pi},\sigma} \right)$$

$$\overset{(ii)}{\leq} \left( V^{\pi^\star,\sigma} - \widehat{V}^{\pi^\star,\sigma} \right) + \frac{2\gamma\varepsilon_{\mathsf{opt}}}{1-\gamma}\mathbf{1} + \left( \widehat{V}^{\widehat{\pi},\sigma} - V^{\widehat{\pi},\sigma} \right) \tag{49}$$

where (i) holds by $\widehat{V}^{\pi^\star,\sigma} - \widehat{V}^{\widehat{\pi}^\star,\sigma} \leq 0$ since $\widehat{\pi}^\star$ is the robust optimal policy for $\widehat{\mathcal{M}}_{\mathsf{rob}}$, and (ii) comes from the fact in (48).

To control the two important terms in (49), we first consider a more general term $\widehat{V}^{\pi,\sigma} - V^{\pi,\sigma}$ for any policy $\pi$. Towards this, plugging in (41) yields

$$\widehat{V}^{\pi,\sigma} - V^{\pi,\sigma} = r_\pi + \gamma \underline{\widehat{P}}^{\pi,\widehat{V}} \widehat{V}^{\pi,\sigma} - \left( r_\pi + \gamma \underline{P}^{\pi,V} V^{\pi,\sigma} \right)$$

$$= \left( \gamma \underline{\widehat{P}}^{\pi,\widehat{V}} \widehat{V}^{\pi,\sigma} - \gamma \underline{P}^{\pi,\widehat{V}} \widehat{V}^{\pi,\sigma} \right) + \left( \gamma \underline{P}^{\pi,\widehat{V}} \widehat{V}^{\pi,\sigma} - \gamma \underline{P}^{\pi,V} V^{\pi,\sigma} \right)$$

$$\overset{(i)}{\leq} \gamma \left( \underline{P}^{\pi,V} \widehat{V}^{\pi,\sigma} - \underline{P}^{\pi,V} V^{\pi,\sigma} \right) + \left( \gamma \underline{\widehat{P}}^{\pi,\widehat{V}} \widehat{V}^{\pi,\sigma} - \gamma \underline{P}^{\pi,\widehat{V}} \widehat{V}^{\pi,\sigma} \right),$$

where (i) holds by observing

$$\underline{P}^{\pi,\widehat{V}} \widehat{V}^{\pi,\sigma} \leq \underline{P}^{\pi,V} \widehat{V}^{\pi,\sigma}$$

due to the optimality of $\underline{P}^{\pi,\widehat{V}}$ (cf. (20)). Rearranging terms leads to

$$\widehat{V}^{\pi,\sigma} - V^{\pi,\sigma} \leq \gamma \left( I - \gamma \underline{P}^{\pi,V} \right)^{-1} \left( \underline{\widehat{P}}^{\pi,\widehat{V}} \widehat{V}^{\pi,\sigma} - \underline{P}^{\pi,\widehat{V}} \widehat{V}^{\pi,\sigma} \right). \tag{50}$$

Similarly, we can also deduce

$$\widehat{V}^{\pi,\sigma} - V^{\pi,\sigma} = r_\pi + \gamma \underline{\widehat{P}}^{\pi,\widehat{V}} \widehat{V}^{\pi,\sigma} - \left( r_\pi + \gamma \underline{P}^{\pi,V} V^{\pi,\sigma} \right)$$

$$= \left( \gamma \underline{\widehat{P}}^{\pi,\widehat{V}} \widehat{V}^{\pi,\sigma} - \gamma \underline{P}^{\pi,\widehat{V}} \widehat{V}^{\pi,\sigma} \right) + \left( \gamma \underline{P}^{\pi,\widehat{V}} \widehat{V}^{\pi,\sigma} - \gamma \underline{P}^{\pi,V} V^{\pi,\sigma} \right)$$

$$\geq \gamma \left( \underline{P}^{\pi,\widehat{V}} \widehat{V}^{\pi,\sigma} - \underline{P}^{\pi,\widehat{V}} V^{\pi,\sigma} \right) + \left( \gamma \underline{\widehat{P}}^{\pi,\widehat{V}} \widehat{V}^{\pi,\sigma} - \gamma \underline{P}^{\pi,\widehat{V}} \widehat{V}^{\pi,\sigma} \right)$$

$$\geq \gamma \left(I - \gamma \underline{P}^{\pi,\widehat{V}}\right)^{-1} \left(\widehat{\underline{P}}^{\pi,\widehat{V}} \widehat{V}^{\pi,\sigma} - \underline{P}^{\pi,\widehat{V}} \widehat{V}^{\pi,\sigma}\right). \tag{51}$$

Combining (50) and (51), we arrive at

$$\left\|\widehat{V}^{\pi,\sigma} - V^{\pi,\sigma}\right\|_\infty \leq \gamma \max \left\{ \left\|\left(I - \gamma \underline{P}^{\pi,V}\right)^{-1} \left(\widehat{\underline{P}}^{\pi,\widehat{V}} \widehat{V}^{\pi,\sigma} - \underline{P}^{\pi,\widehat{V}} \widehat{V}^{\pi,\sigma}\right)\right\|_\infty, \right.$$
$$\left. \left\|\left(I - \gamma \underline{P}^{\pi,\widehat{V}}\right)^{-1} \left(\widehat{\underline{P}}^{\pi,\widehat{V}} \widehat{V}^{\pi,\sigma} - \underline{P}^{\pi,\widehat{V}} \widehat{V}^{\pi,\sigma}\right)\right\|_\infty \right\}. \tag{52}$$

By decomposing the error in a symmetric way, we can similarly obtain

$$\left\|\widehat{V}^{\pi,\sigma} - V^{\pi,\sigma}\right\|_\infty \leq \gamma \max \left\{ \left\|\left(I - \gamma \widehat{\underline{P}}^{\pi,V}\right)^{-1} \left(\widehat{\underline{P}}^{\pi,V} V^{\pi,\sigma} - \underline{P}^{\pi,V} V^{\pi,\sigma}\right)\right\|_\infty, \right.$$
$$\left. \left\|\left(I - \gamma \widehat{\underline{P}}^{\pi,\widehat{V}}\right)^{-1} \left(\widehat{\underline{P}}^{\pi,V} V^{\pi,\sigma} - \underline{P}^{\pi,V} V^{\pi,\sigma}\right)\right\|_\infty \right\}. \tag{53}$$

With the above facts in mind, we are ready to control the two terms $\|\widehat{V}^{\pi^\star,\sigma} - V^{\pi^\star,\sigma}\|_\infty$ and $\|\widehat{V}^{\widehat{\pi},\sigma} - V^{\widehat{\pi},\sigma}\|_\infty$ in (49) separately. More specifically, taking $\pi = \pi^\star$, applying (53) leads to

$$\left\|\widehat{V}^{\pi^\star,\sigma} - V^{\pi^\star,\sigma}\right\|_\infty \leq \gamma \max \left\{ \left\|\left(I - \gamma \widehat{\underline{P}}^{\pi^\star,V}\right)^{-1} \left(\widehat{\underline{P}}^{\pi^\star,V} V^{\pi^\star,\sigma} - \underline{P}^{\pi^\star,V} V^{\pi^\star,\sigma}\right)\right\|_\infty, \right.$$
$$\left. \left\|\left(I - \gamma \widehat{\underline{P}}^{\pi^\star,\widehat{V}}\right)^{-1} \left(\widehat{\underline{P}}^{\pi^\star,V} V^{\pi^\star,\sigma} - \underline{P}^{\pi^\star,V} V^{\pi^\star,\sigma}\right)\right\|_\infty \right\}. \tag{54}$$

Similarly, taking $\pi = \widehat{\pi}$, applying (52) leads to

$$\left\|\widehat{V}^{\widehat{\pi},\sigma} - V^{\widehat{\pi},\sigma}\right\|_\infty \leq \gamma \max \left\{ \left\|\left(I - \gamma \underline{P}^{\widehat{\pi},\widehat{V}}\right)^{-1} \left(\widehat{\underline{P}}^{\widehat{\pi},\widehat{V}} \widehat{V}^{\widehat{\pi},\sigma} - \underline{P}^{\widehat{\pi},\widehat{V}} \widehat{V}^{\widehat{\pi},\sigma}\right)\right\|_\infty, \right.$$
$$\left. \left\|\left(I - \gamma \underline{P}^{\widehat{\pi},V}\right)^{-1} \left(\widehat{\underline{P}}^{\widehat{\pi},\widehat{V}} \widehat{V}^{\widehat{\pi},\sigma} - \underline{P}^{\widehat{\pi},\widehat{V}} \widehat{V}^{\widehat{\pi},\sigma}\right)\right\|_\infty \right\}. \tag{55}$$

**Step 2: controlling $\|\widehat{V}^{\pi^\star,\sigma} - V^{\pi^\star,\sigma}\|_\infty$: bounding the first term in** (54). To control the two terms in (54), we first introduce the following lemma whose proof is postponed to Appendix C.3.3.

**Lemma 9.** *Consider any $\delta \in (0,1)$. Setting $N \geq \log(\frac{18SAN}{\delta})$, with probability at least $1 - \delta$, one has*

$$\left|\widehat{\underline{P}}^{\pi^\star,V} V^{\pi^\star,\sigma} - \underline{P}^{\pi^\star,V} V^{\pi^\star,\sigma}\right| \leq 2\sqrt{\frac{\log(\frac{18SAN}{\delta})}{N}} \sqrt{\mathrm{Var}_{P^{\pi^\star}}(V^{\star,\sigma})} + \frac{\log(\frac{18SAN}{\delta})}{N(1-\gamma)} 1$$
$$\leq 3\sqrt{\frac{\log(\frac{18SAN}{\delta})}{(1-\gamma)^2 N}} 1, \tag{56}$$

*where $\mathrm{Var}_{P^{\pi^\star}}(V^{\star,\sigma})$ is defined in (19).*

Armed with the above lemma, now we control the first term on the right hand side of (54) as follows:

$$\left(I - \gamma \widehat{\underline{P}}^{\pi^\star,V}\right)^{-1} \left(\widehat{\underline{P}}^{\pi^\star,V} V^{\pi^\star,\sigma} - \underline{P}^{\pi^\star,V} V^{\pi^\star,\sigma}\right)$$
$$\overset{\text{(i)}}{\leq} \left(I - \gamma \widehat{\underline{P}}^{\pi^\star,V}\right)^{-1} \left\|\widehat{\underline{P}}^{\pi^\star,V} V^{\pi^\star,\sigma} - \underline{P}^{\pi^\star,V} V^{\pi^\star,\sigma}\right\|_\infty$$
$$\overset{\text{(ii)}}{\leq} \left(I - \gamma \widehat{\underline{P}}^{\pi^\star,V}\right)^{-1} \left(2\sqrt{\frac{\log(\frac{18SAN}{\delta})}{N}} \sqrt{\mathrm{Var}_{P^{\pi^\star}}(V^{\star,\sigma})} + \frac{\log(\frac{18SAN}{\delta})}{N(1-\gamma)} 1\right)$$

$$\leq \frac{\log(\frac{18SAN}{\delta})}{N(1-\gamma)}\left(I-\gamma\underline{\widehat{P}}^{\pi^\star,V}\right)^{-1}1 + \underbrace{2\sqrt{\frac{\log(\frac{18SAN}{\delta})}{N}}\left(I-\gamma\underline{\widehat{P}}^{\pi^\star,V}\right)^{-1}\sqrt{\mathrm{Var}_{\underline{\widehat{P}}^{\pi^\star,V}}(V^{\star,\sigma})}}_{=:\mathcal{C}_1}$$

$$+ \underbrace{2\sqrt{\frac{\log(\frac{18SAN}{\delta})}{N}}\left(I-\gamma\underline{\widehat{P}}^{\pi^\star,V}\right)^{-1}\sqrt{\left|\mathrm{Var}_{\widehat{P}^{\pi^\star}}(V^{\star,\sigma})-\mathrm{Var}_{\underline{\widehat{P}}^{\pi^\star,V}}(V^{\star,\sigma})\right|}}_{=:\mathcal{C}_2}$$

$$+ \underbrace{2\sqrt{\frac{\log(\frac{18SAN}{\delta})}{N}}\left(I-\gamma\underline{\widehat{P}}^{\pi^\star,V}\right)^{-1}\left(\sqrt{\mathrm{Var}_{P^{\pi^\star}}(V^{\star,\sigma})}-\sqrt{\mathrm{Var}_{\widehat{P}^{\pi^\star}}(V^{\star,\sigma})}\right)}_{=:\mathcal{C}_3}, \tag{57}$$

where (i) holds by $\left(I-\gamma\underline{\widehat{P}}^{\pi^\star,V}\right)^{-1} \geq 0$, (ii) follows from Lemma 9, and the last inequality arise from

$$\sqrt{\mathrm{Var}_{P^{\pi^\star}}(V^{\star,\sigma})} = \left(\sqrt{\mathrm{Var}_{P^{\pi^\star}}(V^{\star,\sigma})}-\sqrt{\mathrm{Var}_{\widehat{P}^{\pi^\star}}(V^{\star,\sigma})}\right) + \sqrt{\mathrm{Var}_{\widehat{P}^{\pi^\star}}(V^{\star,\sigma})}$$

$$\leq \left(\sqrt{\mathrm{Var}_{P^{\pi^\star}}(V^{\star,\sigma})}-\sqrt{\mathrm{Var}_{\widehat{P}^{\pi^\star}}(V^{\star,\sigma})}\right) + \sqrt{\left|\mathrm{Var}_{\widehat{P}^{\pi^\star}}(V^{\star,\sigma})-\mathrm{Var}_{\underline{\widehat{P}}^{\pi^\star,V}}(V^{\star,\sigma})\right|}$$

$$+ \sqrt{\mathrm{Var}_{\underline{\widehat{P}}^{\pi^\star,V}}(V^{\star,\sigma})}$$

by applying the triangle inequality.

To continue, observing that each row of $\widehat{\underline{P}}^{\pi^\star,V}$ is a probability distribution obeying that the sum is 1, we arrive at

$$\left(I-\gamma\underline{\widehat{P}}^{\pi^\star,V}\right)^{-1}1 = \left(I+\sum_{t=1}^{\infty}\gamma^t\left(\underline{\widehat{P}}^{\pi^\star,V}\right)^t\right)1 = \frac{1}{1-\gamma}1. \tag{58}$$

Armed with this fact, we shall control the other three terms $\mathcal{C}_1, \mathcal{C}_2, \mathcal{C}_3$ in (57) separately.

- Consider $\mathcal{C}_1$. We first introduce the following lemma, whose proof is postponed to Appendix C.3.4.

  **Lemma 10.** *Consider any $\delta \in (0,1)$. With probability at least $1-\delta$, one has*

  $$\left(I-\gamma\underline{\widehat{P}}^{\pi^\star,V}\right)^{-1}\sqrt{\mathrm{Var}_{\underline{\widehat{P}}^{\pi^\star,V}}(V^{\star,\sigma})} \quad \leq 4\sqrt{\frac{\left(1+\sqrt{\frac{\log(\frac{18SAN}{\delta})}{(1-\gamma)^2N}}\right)}{\gamma^3(1-\gamma)^2\max\{1-\gamma,\sigma\}}}1$$

  $$\leq 4\sqrt{\frac{\left(1+\sqrt{\frac{\log(\frac{18SAN}{\delta})}{(1-\gamma)^2N}}\right)}{\gamma^3(1-\gamma)^3}}1.$$

  Applying Lemma 10 and inserting back to (57) leads to

  $$\mathcal{C}_1 = 2\sqrt{\frac{\log(\frac{18SAN}{\delta})}{N}}\left(I-\gamma\underline{\widehat{P}}^{\pi^\star,V}\right)^{-1}\sqrt{\mathrm{Var}_{\underline{\widehat{P}}^{\pi^\star,V}}(V^{\star,\sigma})}$$

  $$\leq 8\sqrt{\frac{\log(\frac{18SAN}{\delta})}{\gamma^3(1-\gamma)^2\max\{1-\gamma,\sigma\}N}\left(1+\sqrt{\frac{\log(\frac{18SAN}{\delta})}{(1-\gamma)^2N}}\right)}1. \tag{59}$$

- Consider $\mathcal{C}_2$. First, denote $V' := V^{\star,\sigma} - \min_{s'\in\mathcal{S}}V^{\star,\sigma}(s')1$, by Lemma 7, it follows that

  $$0 \leq V' \leq \frac{1}{\gamma\max\{1-\gamma,\sigma\}}1. \tag{60}$$

Then, we have for all $(s, a) \in \mathcal{S} \times \mathcal{A}$, and $P_{s,a} \in \Delta(\mathcal{S})$, and $\widetilde{P}_{s,a} \in \mathcal{U}^\sigma(P_{s,a})$:

$$
\begin{aligned}
\left| \mathsf{Var}_{\widetilde{P}_{s,a}}(V^{\star,\sigma}) - \mathsf{Var}_{P_{s,a}}(V^{\star,\sigma}) \right| &= \left| \mathsf{Var}_{\widetilde{P}_{s,a}}(V') - \mathsf{Var}_{P_{s,a}}(V') \right| \\
&\leq \|\widetilde{P}_{s,a} - P_{s,a}\|_1 \|V'\|_\infty^2 \\
&\leq \frac{2\sigma}{\gamma^2 (\max\{1 - \gamma, \sigma\})^2} 1 \leq \frac{2}{\gamma^2 \max\{1 - \gamma, \sigma\}} 1.
\end{aligned}
\tag{61}
$$

Applying the above relation we obtain

$$
\begin{aligned}
\mathcal{C}_2 &= 2 \sqrt{\frac{\log(\frac{18SAN}{\delta})}{N}} \left( I - \gamma \widehat{\underline{P}}^{\pi^\star, V} \right)^{-1} \sqrt{\left| \mathsf{Var}_{\widehat{P}^{\pi^\star}}(V^{\star,\sigma}) - \mathsf{Var}_{\widehat{\underline{P}}^{\pi^\star,V}}(V^{\star,\sigma}) \right|} \\
&= 2 \sqrt{\frac{\log(\frac{18SAN}{\delta})}{N}} \left( I - \gamma \widehat{\underline{P}}^{\pi^\star, V} \right)^{-1} \sqrt{\left| \Pi^{\pi^\star} \left( \mathsf{Var}_{\widehat{P}^0}(V^{\star,\sigma}) - \mathsf{Var}_{\widehat{P}^{\pi^\star},V}(V^{\star,\sigma}) \right) \right|} \\
&\leq 2 \sqrt{\frac{\log(\frac{18SAN}{\delta})}{N}} \left( I - \gamma \widehat{\underline{P}}^{\pi^\star, V} \right)^{-1} \sqrt{\left\| \mathsf{Var}_{\widehat{P}^0}(V^{\star,\sigma}) - \mathsf{Var}_{\widehat{P}^{\pi^\star},V}(V^{\star,\sigma}) \right\|_\infty} 1 \\
&\leq 2 \sqrt{\frac{\log(\frac{18SAN}{\delta})}{N}} \left( I - \gamma \widehat{\underline{P}}^{\pi^\star, V} \right)^{-1} \sqrt{\frac{2}{\gamma^2 \max\{1 - \gamma, \sigma\}}} 1 \\
&= 2 \sqrt{\frac{2 \log(\frac{18SAN}{\delta})}{\gamma^2 (1 - \gamma)^2 \max\{1 - \gamma, \sigma\} N}} 1,
\end{aligned}
\tag{62}
$$

where the last equality uses $\left( I - \gamma \widehat{\underline{P}}^{\pi^\star, V} \right)^{-1} 1 = \frac{1}{1 - \gamma}$ (cf. (58)).

- Consider $\mathcal{C}_3$. The following lemma plays an important role.

**Lemma 11.** *([Panaganti and Kalathil, 2022](#), Lemma 6) Consider any $\delta \in (0, 1)$. For any fixed policy $\pi$ and fixed value vector $V \in \mathbb{R}^S$, one has with probability at least $1 - \delta$,*

$$
\left| \sqrt{\mathsf{Var}_{\widehat{P}^\pi}(V)} - \sqrt{\mathsf{Var}_{P^\pi}(V)} \right| \leq \sqrt{\frac{2\|V\|_\infty^2 \log(\frac{2SA}{\delta})}{N}} 1.
$$

Applying Lemma 11 with $\pi = \pi^\star$ and $V = V^{\star,\sigma}$ leads to

$$
\sqrt{\mathsf{Var}_{P^{\pi^\star}}(V^{\star,\sigma})} - \sqrt{\mathsf{Var}_{\widehat{P}^{\pi^\star}}(V^{\star,\sigma})} \leq \sqrt{\frac{2\|V^{\star,\sigma}\|_\infty^2 \log(\frac{2SA}{\delta})}{N}} 1,
$$

which can be plugged in (57) to verify

$$
\begin{aligned}
\mathcal{C}_3 &= 2 \sqrt{\frac{\log(\frac{18SAN}{\delta})}{N}} \left( I - \gamma \widehat{\underline{P}}^{\pi^\star, V} \right)^{-1} \left( \sqrt{\mathsf{Var}_{P^{\pi^\star}}(V^{\star,\sigma})} - \sqrt{\mathsf{Var}_{\widehat{P}^{\pi^\star}}(V^{\star,\sigma})} \right) \\
&\leq \frac{4}{(1 - \gamma)} \frac{\log(\frac{SAN}{\delta}) \|V^{\star,\sigma}\|_\infty}{N} 1 \leq \frac{4 \log(\frac{18SAN}{\delta})}{(1 - \gamma)^2 N} 1,
\end{aligned}
\tag{63}
$$

where the last line uses $\left( I - \gamma \widehat{\underline{P}}^{\pi^\star, V} \right)^{-1} 1 = \frac{1}{1 - \gamma}$ (cf. (58)).

Finally, inserting the results of $\mathcal{C}_1$ in (59), $\mathcal{C}_2$ in (62), $\mathcal{C}_3$ in (63), and (58) back into (57) gives

$$
\begin{aligned}
&\left( I - \gamma \widehat{\underline{P}}^{\pi^\star, V} \right)^{-1} \left( \widehat{\underline{P}}^{\pi^\star, V} V^{\pi^\star, \sigma} - \underline{P}^{\pi^\star, V} V^{\pi^\star, \sigma} \right) \\
&\leq 8 \sqrt{\frac{\log(\frac{18SAN}{\delta})}{\gamma^3 (1 - \gamma)^2 \max\{1 - \gamma, \sigma\} N}} \left( 1 + \sqrt{\frac{\log(\frac{18SAN}{\delta})}{(1 - \gamma)^2 N}} \right) 1
\end{aligned}
$$

$$+ 2\sqrt{\frac{2\log(\frac{18SAN}{\delta})}{\gamma^2(1-\gamma)^2\max\{1-\gamma,\sigma\}N}}\mathbb{1} + \frac{4\log(\frac{18SAN}{\delta})}{(1-\gamma)^2N}\mathbb{1} + \frac{\log(\frac{18SAN}{\delta})}{N(1-\gamma)^2}\mathbb{1}$$

$$\leq 10\sqrt{\frac{2\log(\frac{18SAN}{\delta})}{\gamma^3(1-\gamma)^2\max\{1-\gamma,\sigma\}N}}\left(1 + \sqrt{\frac{\log(\frac{SAN}{\delta})}{(1-\gamma)^2N}}\right)\mathbb{1} + \frac{5\log(\frac{18SAN}{\delta})}{(1-\gamma)^2N}\mathbb{1}$$

$$\leq 160\sqrt{\frac{\log(\frac{18SAN}{\delta})}{(1-\gamma)^2\max\{1-\gamma,\sigma\}N}}\mathbb{1} + \frac{5\log(\frac{18SAN}{\delta})}{(1-\gamma)^2N}\mathbb{1}, \tag{64}$$

where the last inequality holds by the fact $\gamma \geq \frac{1}{4}$ and letting $N \geq \frac{\log(\frac{SAN}{\delta})}{(1-\gamma)^2}$.

**Step 3: controlling $\|\widehat{V}^{\pi^\star,\sigma} - V^{\pi^\star,\sigma}\|_\infty$: bounding the second term in** (54). To proceed, applying Lemma 9 on the second term of the right hand side of (54) leads to

$$\left(I - \gamma\underline{\widehat{P}}^{\pi^\star,\widehat{V}}\right)^{-1}\left(\underline{\widehat{P}}^{\pi^\star,V}V^{\pi^\star,\sigma} - \underline{P}^{\pi^\star,V}V^{\pi^\star,\sigma}\right)$$

$$\leq 2\left(I - \gamma\underline{\widehat{P}}^{\pi^\star,\widehat{V}}\right)^{-1}\left(\sqrt{\frac{\log(\frac{18SAN}{\delta})}{N}}\sqrt{\mathrm{Var}_{P^{\pi^\star}}(V^{\star,\sigma})} + \frac{\log(\frac{18SAN}{\delta})}{N(1-\gamma)}\mathbb{1}\right)$$

$$\leq \frac{2\log(\frac{18SAN}{\delta})}{N(1-\gamma)}\left(I - \gamma\underline{\widehat{P}}^{\pi^\star,\widehat{V}}\right)^{-1}\mathbb{1} + \underbrace{2\sqrt{\frac{\log(\frac{18SAN}{\delta})}{N}}\left(I - \gamma\underline{\widehat{P}}^{\pi^\star,\widehat{V}}\right)^{-1}\sqrt{\mathrm{Var}_{\underline{\widehat{P}}^{\pi^\star,\hat{v}}}(\widehat{V}^{\pi^\star,\sigma})}}_{=:\mathcal{C}_4}$$

$$+ \underbrace{2\sqrt{\frac{\log(\frac{18SAN}{\delta})}{N}}\left(I - \gamma\underline{\widehat{P}}^{\pi^\star,\widehat{V}}\right)^{-1}\left(\sqrt{\mathrm{Var}_{\underline{\widehat{P}}^{\pi^\star,\hat{v}}}(V^{\pi^\star,\sigma} - \widehat{V}^{\pi^\star,\sigma})}\right)}_{=:\mathcal{C}_5}$$

$$+ \underbrace{2\sqrt{\frac{\log(\frac{18SAN}{\delta})}{N}}\left(I - \gamma\underline{\widehat{P}}^{\pi^\star,\widehat{V}}\right)^{-1}\left(\sqrt{\left|\mathrm{Var}_{\widehat{P}^{\pi^\star}}(V^{\star,\sigma}) - \mathrm{Var}_{\underline{\widehat{P}}^{\pi^\star,\hat{v}}}(V^{\star,\sigma})\right|}\right)}_{=:\mathcal{C}_6}$$

$$+ \underbrace{2\sqrt{\frac{\log(\frac{18SAN}{\delta})}{N}}\left(I - \gamma\underline{\widehat{P}}^{\pi^\star,\widehat{V}}\right)^{-1}\left(\sqrt{\mathrm{Var}_{P^{\pi^\star}}(V^{\star,\sigma})} - \sqrt{\mathrm{Var}_{\widehat{P}^{\pi^\star}}(V^{\star,\sigma})}\right)}_{=:\mathcal{C}_7}, \tag{65}$$

where the last term $\widetilde{\mathcal{C}}_3$ can be controlled the same as $\mathcal{C}_3$ in (63). We now bound the above terms separately.

- Applying Lemma 8 with $P = \widehat{P}^{\pi^\star,\widehat{V}}$, $\pi = \pi^\star$ and taking $V = \widehat{V}^{\pi^\star,\sigma}$ which obeys $\widehat{V}^{\pi^\star,\sigma} = r_{\pi^\star} + \gamma\underline{\widehat{P}}^{\pi^\star,\widehat{V}}\widehat{V}^{\pi^\star,\sigma}$, and in view of (58), the term $\mathcal{C}_4$ in (65) can be controlled as follows:

$$\mathcal{C}_4 = 2\sqrt{\frac{\log(\frac{18SAN}{\delta})}{N}}\left(I - \gamma\underline{\widehat{P}}^{\pi^\star,\widehat{V}}\right)^{-1}\sqrt{\mathrm{Var}_{\underline{\widehat{P}}^{\pi^\star,\hat{v}}}(\widehat{V}^{\pi^\star,\sigma})}$$

$$\leq 2\sqrt{\frac{\log(\frac{18SAN}{\delta})}{N}}\sqrt{\frac{8(\max_s \widehat{V}^{\pi^\star,\sigma}(s) - \min_s \widehat{V}^{\pi^\star,\sigma}(s))}{\gamma^2(1-\gamma)^2}}\mathbb{1}$$

$$\leq 8\sqrt{\frac{\log(\frac{18SAN}{\delta})}{\gamma^3(1-\gamma)^2\max\{1-\gamma,\sigma\}N}}\mathbb{1}, \tag{66}$$

where the last inequality holds by applying Lemma 7.

- To continue, considering $\mathcal{C}_5$, we directly observe that (in view of (58))

$$\mathcal{C}_5 = 2\sqrt{\frac{\log(\frac{18SAN}{\delta})}{N}}\left(I - \gamma\widehat{\underline{P}}^{\pi^\star,\widehat{V}}\right)^{-1}\sqrt{\mathrm{Var}_{\widehat{\underline{P}}^{\pi^\star,\widehat{V}}}\left(V^{\pi^\star,\sigma} - \widehat{V}^{\pi^\star,\sigma}\right)}$$

$$\leq 2\sqrt{\frac{\log(\frac{18SAN}{\delta})}{(1-\gamma)^2 N}}\left\|V^{\star,\sigma} - \widehat{V}^{\pi^\star,\sigma}\right\|_\infty 1. \tag{67}$$

- Then, it is easily verified that $\mathcal{C}_6$ can be controlled similarly as (62) as follows:

$$\mathcal{C}_6 \leq 2\sqrt{\frac{2\log(\frac{18SAN}{\delta})}{\gamma^2(1-\gamma)^2\max\{1-\gamma,\sigma\}N}}1. \tag{68}$$

- Similarly, $\mathcal{C}_7$ can be controlled the same as (63) shown below:

$$\mathcal{C}_7 \leq \frac{4\log(\frac{18SAN}{\delta})}{(1-\gamma)^2 N}1. \tag{69}$$

Combining the results in (66), (67), (68), and (69) and inserting back to (65) leads to

$$\left(I - \gamma\widehat{\underline{P}}^{\pi^\star,\widehat{V}}\right)^{-1}\left(\widehat{\underline{P}}^{\pi^\star,V}V^{\pi^\star,\sigma} - \underline{P}^{\pi^\star,V}V^{\pi^\star,\sigma}\right) \leq 8\sqrt{\frac{\log(\frac{18SAN}{\delta})}{\gamma^3(1-\gamma)^2\max\{1-\gamma,\sigma\}N}}1$$

$$+ 2\sqrt{\frac{\log(\frac{18SAN}{\delta})}{(1-\gamma)^2 N}}\left\|V^{\star,\sigma} - \widehat{V}^{\pi^\star,\sigma}\right\|_\infty 1 + 2\sqrt{\frac{2\log(\frac{18SAN}{\delta})}{\gamma^2(1-\gamma)^2\max\{1-\gamma,\sigma\}N}}1 + \frac{4\log(\frac{18SAN}{\delta})}{(1-\gamma)^2 N}1$$

$$\leq 80\sqrt{\frac{\log(\frac{18SAN}{\delta})}{(1-\gamma)^2\max\{1-\gamma,\sigma\}N}}1 + 2\sqrt{\frac{\log(\frac{18SAN}{\delta})}{(1-\gamma)^2 N}}\left\|V^{\star,\sigma} - \widehat{V}^{\pi^\star,\sigma}\right\|_\infty 1 + \frac{4\log(\frac{18SAN}{\delta})}{(1-\gamma)^2 N}1, \tag{70}$$

where the last inequality follows from the assumption $\gamma \geq \frac{1}{4}$.

Finally, inserting (64) and (70) back to (54) yields

$$\left\|\widehat{V}^{\pi^\star,\sigma} - V^{\pi^\star,\sigma}\right\|_\infty \leq \max\left\{160\sqrt{\frac{\log(\frac{18SAN}{\delta})}{(1-\gamma)^2\max\{1-\gamma,\sigma\}N}} + \frac{5\log(\frac{18SAN}{\delta})}{(1-\gamma)^2 N},\right.$$

$$\left.80\sqrt{\frac{\log(\frac{18SAN}{\delta})}{(1-\gamma)^2\max\{1-\gamma,\sigma\}N}} + 2\sqrt{\frac{\log(\frac{18SAN}{\delta})}{(1-\gamma)^2 N}}\left\|V^{\star,\sigma} - \widehat{V}^{\pi^\star,\sigma}\right\|_\infty + \frac{4\log(\frac{18SAN}{\delta})}{(1-\gamma)^2 N}\right\}$$

$$\leq 160\sqrt{\frac{\log(\frac{18SAN}{\delta})}{(1-\gamma)^2\max\{1-\gamma,\sigma\}N}} + \frac{8\log(\frac{18SAN}{\delta})}{(1-\gamma)^2 N}, \tag{71}$$

where the last inequality holds by taking $N \geq \frac{16\log(\frac{SAN}{\delta})}{(1-\gamma)^2}$.

**Step 4: controlling $\|\widehat{V}^{\widehat{\pi},\sigma} - V^{\widehat{\pi},\sigma}\|_\infty$: bounding the first term in** (55). Unlike the earlier term, we now need to deal with the complicated statistical dependency between $\widehat{\pi}$ and the empirical RMDP. To begin with, we introduce the following lemma which controls the main term on the right hand side of (55), which is proved in Appendix C.3.5.

**Lemma 12.** *Consider any $\delta \in (0,1)$. Taking $N \geq \log\left(\frac{54SAN^2}{(1-\gamma)\delta}\right)$, with probability at least $1 - \delta$, one has*

$$\left|\widehat{\underline{P}}^{\widehat{\pi},\widehat{V}}\widehat{V}^{\widehat{\pi},\sigma} - \underline{P}^{\widehat{\pi},\widehat{V}}\widehat{V}^{\widehat{\pi},\sigma}\right| \leq 2\sqrt{\frac{\log(\frac{54SAN^2}{(1-\gamma)\delta})}{N}}\sqrt{\mathrm{Var}_{P^0_{s,a}}(\widehat{V}^{\star,\sigma})}1 + \frac{8\log(\frac{54SAN^2}{(1-\gamma)\delta})}{N(1-\gamma)}1 + \frac{2\gamma\varepsilon_{\mathsf{opt}}}{1-\gamma}1$$

$$\leq 10\sqrt{\frac{\log(\frac{54SAN^2}{(1-\gamma)\delta})}{(1-\gamma)^2 N}}1 + \frac{2\gamma\varepsilon_{\mathsf{opt}}}{1-\gamma}1. \tag{72}$$

With Lemma 12 in hand, we have

$$\left(I - \gamma\underline{P}^{\widehat{\pi},\widehat{V}}\right)^{-1}\left(\underline{\widehat{P}}^{\widehat{\pi},\widehat{V}}\widehat{V}^{\widehat{\pi},\sigma} - \underline{P}^{\widehat{\pi},\widehat{V}}\widehat{V}^{\widehat{\pi},\sigma}\right)$$

$$\overset{(i)}{\leq} \left(I - \gamma\underline{P}^{\widehat{\pi},\widehat{V}}\right)^{-1}\left|\underline{\widehat{P}}^{\widehat{\pi},\widehat{V}}\widehat{V}^{\widehat{\pi},\sigma} - \underline{P}^{\widehat{\pi},\widehat{V}}\widehat{V}^{\widehat{\pi},\sigma}\right|$$

$$\leq 2\sqrt{\frac{\log(\frac{54SAN^2}{(1-\gamma)\delta})}{N}}\left(I - \gamma\underline{P}^{\widehat{\pi},\widehat{V}}\right)^{-1}\sqrt{\mathrm{Var}_{P^{\widehat{\pi}}}(\widehat{V}^{\star,\sigma})}$$

$$+ \left(I - \gamma P_Q^{\widehat{\pi},V^{\widehat{\pi}}}\right)^{-1}\left(\frac{8\log(\frac{54SAN^2}{(1-\gamma)\delta})}{N(1-\gamma)} + \frac{2\gamma\varepsilon_{\mathsf{opt}}}{1-\gamma}\right)\mathbf{1}$$

$$\overset{(ii)}{\leq} \left(\frac{8\log(\frac{54SAN^2}{(1-\gamma)\delta})}{N(1-\gamma)^2} + \frac{2\gamma\varepsilon_{\mathsf{opt}}}{(1-\gamma)^2}\right)\mathbf{1} + \underbrace{2\sqrt{\frac{\log(\frac{54SAN^2}{(1-\gamma)\delta})}{N}}\left(I - \gamma\underline{P}^{\widehat{\pi},\widehat{V}}\right)^{-1}\sqrt{\mathrm{Var}_{\underline{P}^{\widehat{\pi},\widehat{v}}}(\widehat{V}^{\widehat{\pi},\sigma})}}_{=:\mathcal{D}_1}$$

$$+ \underbrace{2\sqrt{\frac{\log(\frac{54SAN^2}{(1-\gamma)\delta})}{N}}\left(I - \gamma\underline{P}^{\widehat{\pi},\widehat{V}}\right)^{-1}\sqrt{\left|\mathrm{Var}_{\underline{P}^{\widehat{\pi},\widehat{v}}}(\widehat{V}^{\star,\sigma}) - \mathrm{Var}_{\underline{P}^{\widehat{\pi},\widehat{v}}}(\widehat{V}^{\widehat{\pi},\sigma})\right|}}_{=:\mathcal{D}_2}$$

$$+ \underbrace{2\sqrt{\frac{\log(\frac{54SAN^2}{(1-\gamma)\delta})}{N}}\left(I - \gamma\underline{P}^{\widehat{\pi},\widehat{V}}\right)^{-1}\sqrt{\left|\mathrm{Var}_{P^{\widehat{\pi}}}(\widehat{V}^{\star,\sigma}) - \mathrm{Var}_{\underline{P}^{\widehat{\pi},\widehat{v}}}(\widehat{V}^{\star,\sigma})\right|}}_{=:\mathcal{D}_3}, \tag{73}$$

where (i) and (ii) hold by the fact that each row of $(1-\gamma)\left(I - \gamma\underline{P}^{\widehat{\pi},\widehat{V}}\right)^{-1}$ is a probability vector that falls into $\Delta(\mathcal{S})$.

The remainder of the proof will focus on controlling the three terms in (73) separately.

- For $\mathcal{D}_1$, we introduce the following lemma, whose proof is postponed to C.3.6.

  **Lemma 13.** *Consider any $\delta \in (0,1)$. Taking $N \geq \frac{\log(\frac{54SAN^2}{(1-\gamma)\delta})}{(1-\gamma)^2}$ and $\varepsilon_{\mathsf{opt}} \leq \frac{1-\gamma}{\gamma}$, one has with probability at least $1 - \delta$,*

  $$\left(I - \gamma\underline{P}^{\widehat{\pi},\widehat{V}}\right)^{-1}\sqrt{\mathrm{Var}_{\underline{P}^{\widehat{\pi},\widehat{v}}}(\widehat{V}^{\widehat{\pi},\sigma})} \leq 6\sqrt{\frac{1}{\gamma^3(1-\gamma)^2\max\{1-\gamma,\sigma\}}}\mathbf{1} \leq 6\sqrt{\frac{1}{(1-\gamma)^3\gamma^2}}\mathbf{1}.$$

  Applying Lemma 13 and (58) to (73) leads to

  $$\mathcal{D}_1 = 2\sqrt{\frac{\log(\frac{54SAN^2}{(1-\gamma)\delta})}{N}}\left(I - \gamma\underline{P}^{\widehat{\pi},\widehat{V}}\right)^{-1}\sqrt{\mathrm{Var}_{\underline{P}^{\widehat{\pi},\widehat{v}}}(\widehat{V}^{\widehat{\pi},\sigma})}$$

  $$\leq 12\sqrt{\frac{\log(\frac{54SAN^2}{(1-\gamma)\delta})}{\gamma^3(1-\gamma)^2\max\{1-\gamma,\sigma\}N}}\mathbf{1}. \tag{74}$$

- Applying Lemma 2 with $\|\widehat{V}^{\star,\sigma} - \widehat{V}^{\widehat{\pi},\sigma}\|_\infty \leq \frac{2\gamma\varepsilon_{\mathsf{opt}}}{1-\gamma}$ and (58), $\mathcal{D}_2$ can be controlled as

  $$\mathcal{D}_2 = 2\sqrt{\frac{\log(\frac{54SAN^2}{(1-\gamma)\delta})}{N}}\left(I - \gamma\underline{P}^{\widehat{\pi},\widehat{V}}\right)^{-1}\sqrt{\left|\mathrm{Var}_{\underline{P}^{\widehat{\pi},\widehat{v}}}(\widehat{V}^{\star,\sigma}) - \mathrm{Var}_{\underline{P}^{\widehat{\pi},\widehat{v}}}(\widehat{V}^{\widehat{\pi},\sigma})\right|}$$

  $$\leq 4\sqrt{\frac{\log(\frac{54SAN^2}{(1-\gamma)\delta})}{N}}\left(I - \gamma\underline{P}^{\widehat{\pi},\widehat{V}}\right)^{-1}\frac{\sqrt{\gamma\varepsilon_{\mathsf{opt}}}}{1-\gamma} \leq 4\sqrt{\frac{\gamma\varepsilon_{\mathsf{opt}}\log(\frac{54SAN^2}{(1-\gamma)\delta})}{(1-\gamma)^4N}}\mathbf{1}. \tag{75}$$

- $\mathcal{D}_3$ can be controlled similar to $\mathcal{C}_2$ in (62) as follows:

  $$\mathcal{D}_3 = 2\sqrt{\frac{\log(\frac{54SAN^2}{(1-\gamma)\delta})}{N}}\left(I - \gamma\underline{P}^{\widehat{\pi},\widehat{V}}\right)^{-1}\sqrt{\left|\mathrm{Var}_{P^{\widehat{\pi}}}(\widehat{V}^{\star,\sigma}) - \mathrm{Var}_{\underline{P}^{\widehat{\pi},\widehat{v}}}(\widehat{V}^{\star,\sigma})\right|}$$

$$\leq 4\sqrt{\frac{\log(\frac{54SAN^2}{(1-\gamma)\delta})}{N}}\left(I - \gamma\underline{P}^{\widehat{\pi},\widehat{V}}\right)^{-1}\sqrt{\frac{1}{\gamma^2\max\{1-\gamma,\sigma\}}}1$$

$$\leq 4\sqrt{\frac{\log(\frac{54SAN^2}{(1-\gamma)\delta})}{\gamma^2(1-\gamma)^2\max\{1-\gamma,\sigma\}N}}1 \tag{76}$$

Finally, summing up the results in (74), (75), and (76) and inserting them back to (73) yields: taking $N \geq \frac{\log(\frac{54SAN^2}{(1-\gamma)\delta})}{(1-\gamma)^2}$ and $\varepsilon_{\mathsf{opt}} \leq \frac{1-\gamma}{\gamma}$, with probability at least $1-\delta$,

$$\left(I - \gamma\underline{P}^{\widehat{\pi},\widehat{V}}\right)^{-1}\left(\underline{\widehat{P}}^{\widehat{\pi},\widehat{V}}\widehat{V}^{\widehat{\pi},\sigma} - \underline{P}^{\widehat{\pi},\widehat{V}}\widehat{V}^{\widehat{\pi},\sigma}\right) \leq \left(\frac{8\log(\frac{54SAN^2}{(1-\gamma)\delta})}{N(1-\gamma)^2} + \frac{2\gamma\varepsilon_{\mathsf{opt}}}{(1-\gamma)^2}\right)1$$

$$+ 12\sqrt{\frac{\log(\frac{54SAN^2}{(1-\gamma)\delta})}{\gamma^3(1-\gamma)^2\max\{1-\gamma,\sigma\}N}}1 + 4\sqrt{\frac{\gamma\varepsilon_{\mathsf{opt}}\log(\frac{54SAN^2}{(1-\gamma)\delta})}{(1-\gamma)^4N}}1$$

$$+ 4\sqrt{\frac{\log(\frac{54SAN^2}{(1-\gamma)\delta})}{\gamma^2(1-\gamma)^2\max\{1-\gamma,\sigma\}N}}1$$

$$\leq 16\sqrt{\frac{\log(\frac{54SAN^2}{(1-\gamma)\delta})}{\gamma^3(1-\gamma)^2\max\{1-\gamma,\sigma\}N}}1 + \frac{14\log(\frac{54SAN^2}{(1-\gamma)\delta})}{N(1-\gamma)^2}1, \tag{77}$$

where the last inequality holds by taking $\varepsilon_{\mathsf{opt}} \leq \min\left\{\frac{1-\gamma}{\gamma}, \frac{\log(\frac{54SAN^2}{(1-\gamma)\delta})}{\gamma N}\right\} = \frac{\log(\frac{54SAN^2}{(1-\gamma)\delta})}{\gamma N}$.

**Step 5: controlling $\|\widehat{V}^{\widehat{\pi},\sigma} - V^{\widehat{\pi},\sigma}\|_\infty$: bounding the second term in (55).** Towards this, applying Lemma 12 leads to

$$\left(I - \gamma\underline{P}^{\widehat{\pi},V}\right)^{-1}\left(\underline{\widehat{P}}^{\widehat{\pi},\widehat{V}}\widehat{V}^{\widehat{\pi},\sigma} - \underline{P}^{\widehat{\pi},\widehat{V}}\widehat{V}^{\widehat{\pi},\sigma}\right) \leq \left(I - \gamma\underline{P}^{\widehat{\pi},V}\right)^{-1}\left|\underline{\widehat{P}}^{\widehat{\pi},\widehat{V}}\widehat{V}^{\widehat{\pi},\sigma} - \underline{P}^{\widehat{\pi},\widehat{V}}\widehat{V}^{\widehat{\pi},\sigma}\right|$$

$$\leq 2\sqrt{\frac{\log(\frac{54SAN^2}{(1-\gamma)\delta})}{N}}\left(I - \gamma\underline{P}^{\widehat{\pi},V}\right)^{-1}\sqrt{\mathrm{Var}_{\underline{P}^{\widehat{\pi}}}(\widehat{V}^{\star,\sigma})}$$

$$+ \left(I - \gamma\underline{P}^{\widehat{\pi},V}\right)^{-1}\left(\frac{8\log(\frac{54SAN^2}{(1-\gamma)\delta})}{N(1-\gamma)} + \frac{2\gamma\varepsilon_{\mathsf{opt}}}{1-\gamma}\right)1$$

$$\leq \left(\frac{8\log(\frac{54SAN^2}{(1-\gamma)\delta})}{N(1-\gamma)^2} + \frac{2\gamma\varepsilon_{\mathsf{opt}}}{(1-\gamma)^2}\right)1 + \underbrace{2\sqrt{\frac{\log(\frac{54SAN^2}{(1-\gamma)\delta})}{N}}\left(I - \gamma\underline{P}^{\widehat{\pi},V}\right)^{-1}\sqrt{\mathrm{Var}_{\underline{P}^{\widehat{\pi},V}}(V^{\widehat{\pi},\sigma})}}_{=:\mathcal{D}_4}$$

$$+ \underbrace{2\sqrt{\frac{\log(\frac{54SAN^2}{(1-\gamma)\delta})}{N}}\left(I - \gamma\underline{P}^{\widehat{\pi},V}\right)^{-1}\sqrt{\mathrm{Var}_{\underline{P}^{\widehat{\pi},V}}(\widehat{V}^{\widehat{\pi},\sigma} - V^{\widehat{\pi},\sigma})}}_{=:\mathcal{D}_5}$$

$$+ \underbrace{2\sqrt{\frac{\log(\frac{54SAN^2}{(1-\gamma)\delta})}{N}}\left(I - \gamma\underline{P}^{\widehat{\pi},\widehat{V}}\right)^{-1}\sqrt{\left|\mathrm{Var}_{\underline{P}^{\widehat{\pi},V}}(\widehat{V}^{\star,\sigma}) - \mathrm{Var}_{\underline{P}^{\widehat{\pi},V}}(\widehat{V}^{\widehat{\pi},\sigma})\right|}}_{=:\mathcal{D}_6}$$

$$+ \underbrace{2\sqrt{\frac{\log(\frac{54SAN^2}{(1-\gamma)\delta})}{N}}\left(I - \gamma\underline{P}^{\widehat{\pi},\widehat{V}}\right)^{-1}\sqrt{\left|\mathrm{Var}_{\underline{P}^{\widehat{\pi}}}(\widehat{V}^{\star,\sigma}) - \mathrm{Var}_{\underline{P}^{\widehat{\pi},V}}(\widehat{V}^{\star,\sigma})\right|}}_{=:\mathcal{D}_7}. \tag{78}$$

We shall bound each of the terms separately.

- Applying Lemma 8 with $P = \underline{P}^{\widehat{\pi}, V}$, $\pi = \widehat{\pi}$, and taking $V = V^{\widehat{\pi}, \sigma}$ which obeys $V^{\widehat{\pi}, \sigma} = r_{\widehat{\pi}} + \gamma \underline{P}^{\widehat{\pi}, V} V^{\widehat{\pi}, \sigma}$, the term $\mathcal{D}_4$ can be controlled similar to (66) as follows:

$$\mathcal{D}_4 \leq 8\sqrt{\frac{\log(\frac{54SAN^2}{\delta})}{\gamma^3(1-\gamma)^2 \max\{1-\gamma, \sigma\}N}}1. \tag{79}$$

- For $\mathcal{D}_5$, it is observed that

$$\mathcal{D}_5 = 2\sqrt{\frac{\log(\frac{54SAN^2}{(1-\gamma)\delta})}{N}}\left(I - \gamma \underline{P}^{\widehat{\pi}, V}\right)^{-1}\sqrt{\mathrm{Var}_{\underline{P}^{\widehat{\pi}, V}}(\widehat{V}^{\widehat{\pi}, \sigma} - V^{\widehat{\pi}, \sigma})}$$

$$\leq 2\sqrt{\frac{\log(\frac{54SAN^2}{\delta})}{(1-\gamma)^2 N}}\left\|V^{\widehat{\pi}, \sigma} - \widehat{V}^{\widehat{\pi}, \sigma}\right\|_{\infty}1. \tag{80}$$

- Next, observing that $\mathcal{D}_6$ and $\mathcal{D}_7$ are almost the same as the terms $\mathcal{D}_2$ (controlled in (75)) and $\mathcal{D}_3$ (controlled in (76)) in (73), it is easily verified that they can be controlled as follows

$$\mathcal{D}_6 \leq 4\sqrt{\frac{\gamma\varepsilon_{\mathsf{opt}}\log(\frac{54SAN^2}{(1-\gamma)\delta})}{(1-\gamma)^4 N}}1, \qquad \mathcal{D}_7 \leq 4\sqrt{\frac{\log(\frac{54SAN^2}{(1-\gamma)\delta})}{\gamma^2(1-\gamma)^2 \max\{1-\gamma, \sigma\}N}}1. \tag{81}$$

Then inserting the results in (79), (80), and (81) back to (78) leads to

$$\left(I - \gamma\underline{P}^{\widehat{\pi}, V}\right)^{-1}\left(\widehat{\underline{P}}^{\widehat{\pi}, \widehat{V}}\widehat{V}^{\widehat{\pi}, \sigma} - \underline{P}^{\widehat{\pi}, \widehat{V}}\widehat{V}^{\widehat{\pi}, \sigma}\right)$$

$$\leq \left(\frac{8\log(\frac{54SAN^2}{(1-\gamma)\delta})}{N(1-\gamma)^2} + \frac{2\gamma\varepsilon_{\mathsf{opt}}}{(1-\gamma)^2}\right)1 + 8\sqrt{\frac{\log(\frac{54SAN^2}{\delta})}{\gamma^3(1-\gamma)^2 \max\{1-\gamma, \sigma\}N}}1$$

$$+ 2\sqrt{\frac{\log(\frac{54SAN^2}{\delta})}{(1-\gamma)^2 N}}\left\|V^{\widehat{\pi}, \sigma} - \widehat{V}^{\widehat{\pi}, \sigma}\right\|_{\infty}1 + 4\sqrt{\frac{\gamma\varepsilon_{\mathsf{opt}}\log(\frac{54SAN^2}{(1-\gamma)\delta})}{(1-\gamma)^4 N}}1$$

$$+ 4\sqrt{\frac{\log(\frac{54SAN^2}{(1-\gamma)\delta})}{\gamma^2(1-\gamma)^2 \max\{1-\gamma, \sigma\}N}}1$$

$$\leq 12\sqrt{\frac{2\log(\frac{8SAN^2}{(1-\gamma)\delta})}{\gamma^3(1-\gamma)^2 \max\{1-\gamma, \sigma\}N}}1 + \frac{14\log(\frac{54SAN^2}{(1-\gamma)\delta})}{N(1-\gamma)^2}1 + 2\sqrt{\frac{\log(\frac{54SAN^2}{\delta})}{(1-\gamma)^2 N}}\left\|V^{\widehat{\pi}, \sigma} - \widehat{V}^{\widehat{\pi}, \sigma}\right\|_{\infty}1, \tag{82}$$

where the last inequality holds by letting $\varepsilon_{\mathsf{opt}} \leq \frac{\log(\frac{54SAN^2}{(1-\gamma)\delta})}{\gamma N}$, which directly satisfies $\varepsilon_{\mathsf{opt}} \leq \frac{1-\gamma}{\gamma}$ by letting $N \geq \frac{\log(\frac{54SAN^2}{\delta})}{1-\gamma}$.

Finally, inserting (77) and (82) back to (55) yields: taking $\varepsilon_{\mathsf{opt}} \leq \frac{\log(\frac{54SAN^2}{(1-\gamma)\delta})}{\gamma N}$ and $N \geq \frac{16\log(\frac{54SAN^2}{\delta})}{(1-\gamma)^2}$, with probability at least $1 - \delta$, one has

$$\left\|\widehat{V}^{\widehat{\pi}, \sigma} - V^{\widehat{\pi}, \sigma}\right\|_{\infty} \leq \max\left\{16\sqrt{\frac{\log(\frac{54SAN^2}{(1-\gamma)\delta})}{\gamma^3(1-\gamma)^2 \max\{1-\gamma, \sigma\}N}} + \frac{14\log(\frac{54SAN^2}{(1-\gamma)\delta})}{N(1-\gamma)^2}, \right.$$

$$\left. 12\sqrt{\frac{2\log(\frac{8SAN^2}{(1-\gamma)\delta})}{\gamma^3(1-\gamma)^2 \max\{1-\gamma, \sigma\}N}} + \frac{14\log(\frac{54SAN^2}{(1-\gamma)\delta})}{N(1-\gamma)^2} + 2\sqrt{\frac{\log(\frac{54SAN^2}{\delta})}{(1-\gamma)^2 N}}\left\|V^{\widehat{\pi}, \sigma} - \widehat{V}^{\widehat{\pi}, \sigma}\right\|_{\infty}\right\}$$

$$\leq 24\sqrt{\frac{\log(\frac{54SAN^2}{(1-\gamma)\delta})}{\gamma^3(1-\gamma)^2 \max\{1-\gamma, \sigma\}N}} + \frac{28\log(\frac{54SAN^2}{(1-\gamma)\delta})}{N(1-\gamma)^2}. \tag{83}$$

**Step 6: summing up the results.** Summing up the results in (71) and (83) and inserting back to (49) complete the proof as follows: taking $\varepsilon_{\mathsf{opt}} \leq \frac{\log(\frac{54SAN^2}{(1-\gamma)\delta})}{\gamma N}$ and $N \geq \frac{16\log(\frac{54SAN^2}{\delta})}{(1-\gamma)^2}$, with probability at least $1 - \delta$,

$$
\begin{aligned}
\|V^{\star,\sigma} - V^{\widehat{\pi},\sigma}\|_\infty &\leq \left\|V^{\pi^\star,\sigma} - \widehat{V}^{\pi^\star,\sigma}\right\|_\infty + \frac{2\gamma\varepsilon_{\mathsf{opt}}}{1-\gamma} + \left\|\widehat{V}^{\widehat{\pi},\sigma} - V^{\widehat{\pi},\sigma}\right\|_\infty \\
&\leq \frac{2\gamma\varepsilon_{\mathsf{opt}}}{1-\gamma} + 160\sqrt{\frac{\log(\frac{18SAN}{\delta})}{(1-\gamma)^2\max\{1-\gamma,\sigma\}N}} + \frac{8\log(\frac{18SAN}{\delta})}{(1-\gamma)^2 N} \\
&\quad + 24\sqrt{\frac{\log(\frac{54SAN^2}{(1-\gamma)\delta})}{\gamma^3(1-\gamma)^2\max\{1-\gamma,\sigma\}N}} + \frac{28\log(\frac{54SAN^2}{(1-\gamma)\delta})}{N(1-\gamma)^2} \\
&\leq 184\sqrt{\frac{\log(\frac{54SAN^2}{(1-\gamma)\delta})}{\gamma^3(1-\gamma)^2\max\{1-\gamma,\sigma\}N}} + \frac{36\log(\frac{54SAN^2}{(1-\gamma)\delta})}{N(1-\gamma)^2} \\
&\leq 1508\sqrt{\frac{\log(\frac{54SAN^2}{(1-\gamma)\delta})}{(1-\gamma)^2\max\{1-\gamma,\sigma\}N}}, \quad\quad (84)
\end{aligned}
$$

where the last inequality holds by $\gamma \geq \frac{1}{4}$ and $N \geq \frac{16\log(\frac{54SAN^2}{\delta})}{(1-\gamma)^2}$.

### C.3   Proof of the auxiliary lemmas

#### C.3.1   Proof of Lemma 7

To begin, note that there at leasts exist one state $s_0$ for any $V^{\pi,\sigma}$ such that $V^{\pi,\sigma}(s_0) = \min_{s\in\mathcal{S}} V^{\pi,\sigma}(s)$. With this in mind, for any policy $\pi$, one has by the definition in (**??**) and the Bellman's equation (**??**),

$$
\begin{aligned}
\max_{s\in\mathcal{S}} V^{\pi,\sigma}(s) &= \max_{s\in\mathcal{S}} \mathbb{E}_{a\sim\pi(\cdot\,|\,s)}\left[r(s,a) + \gamma \inf_{\mathcal{P}\in\mathcal{U}^\sigma(P_{s,a})} \mathcal{P}V^{\pi,\sigma}\right] \\
&\leq \max_{(s,a)\in\mathcal{S}\times\mathcal{A}}\left(1 + \gamma\inf_{\mathcal{P}\in\mathcal{U}^\sigma(P_{s,a})}\mathcal{P}V^{\pi,\sigma}\right),
\end{aligned}
$$

where the second line holds since the reward function $r(s,a) \in [0,1]$ for all $(s,a) \in \mathcal{S}\times\mathcal{A}$. To continue, note that for any $(s,a) \in \mathcal{S}\times\mathcal{A}$, there exists some $\widetilde{P}_{s,a} \in \mathbb{R}^S$ constructed by reducing the values of some elements of $P_{s,a}$ to obey $P_{s,a} \geq \widetilde{P}_{s,a} \geq 0$ and $\sum_{s'}(P_{s,a}(s') - \widetilde{P}_{s,a}(s')) = \sigma$. This implies $\widetilde{P}_{s,a} + \sigma e_{s_0}^\top \in \mathcal{U}^\sigma(P_{s,a})$, where $e_{s_0}$ is the standard basis vector supported on $s_0$, since $\frac{1}{2}\|\widetilde{P}_{s,a} + \sigma e_{s_0}^\top - P_{s,a}\|_1 \leq \frac{1}{2}\|\widetilde{P}_{s,a} - P_{s,a}\|_1 + \frac{\sigma}{2} = \sigma$. Consequently,

$$
\begin{aligned}
\inf_{\mathcal{P}\in\mathcal{U}^\sigma(P_{s,a})}\mathcal{P}V^{\pi,\sigma} &\leq \left(\widetilde{P}_{s,a} + \sigma e_{s_0}^\top\right)V^{\pi,\sigma} \leq \|\widetilde{P}_{s,a}\|_1\|V^{\pi,\sigma}\|_\infty + \sigma V^{\pi,\sigma}(s_0) \\
&\leq (1-\sigma)\max_{s\in\mathcal{S}} V^{\pi,\sigma}(s) + \sigma\min_{s\in\mathcal{S}} V^{\pi,\sigma}(s), \quad\quad (85)
\end{aligned}
$$

where the second inequality holds by $\|\widetilde{P}_{s,a}\|_1 = \sum_{s'}\widetilde{P}_{s,a}(s') = -\sum_{s'}\left(P_{s,a}(s') - \widetilde{P}_{s,a}(s')\right) + \sum_{s'}P_{s,a}(s') = 1 - \sigma$. Plugging this back to the previous relation gives

$$
\max_{s\in\mathcal{S}} V^{\pi,\sigma}(s) \leq 1 + \gamma(1-\sigma)\max_{s\in\mathcal{S}} V^{\pi,\sigma}(s) + \gamma\sigma\min_{s\in\mathcal{S}} V^{\pi,\sigma}(s),
$$

which, by rearranging terms, immediately yields

$$
\begin{aligned}
\max_{s\in\mathcal{S}} V^{\pi,\sigma}(s) &\leq \frac{1 + \gamma\sigma\min_{s\in\mathcal{S}} V^{\pi,\sigma}(s)}{1 - \gamma(1-\sigma)} \\
&\leq \frac{1}{(1-\gamma) + \gamma\sigma} + \min_{s\in\mathcal{S}} V^{\pi,\sigma}(s) \leq \frac{1}{\gamma\max\{1-\gamma,\sigma\}} + \min_{s\in\mathcal{S}} V^{\pi,\sigma}(s).
\end{aligned}
$$

 **C.3.2   Proof of Lemma 8**

811   Observing that each row of $P_\pi$ belongs to $\Delta(S)$, it can be directly verified that each row of $(1 -$
812   $\gamma)(I - \gamma P_\pi)^{-1}$ falls into $\Delta(S)$. As a result,

$$
\begin{aligned}
(I - \gamma P_\pi)^{-1} \sqrt{\mathrm{Var}_{P_\pi}(V^{\pi,P})} &= \frac{1}{1-\gamma}(1-\gamma)(I - \gamma P_\pi)^{-1} \sqrt{\mathrm{Var}_{P_\pi}(V^{\pi,P})} \\
&\overset{(i)}{\leq} \frac{1}{1-\gamma}\sqrt{(1-\gamma)(I - \gamma P_\pi)^{-1}\mathrm{Var}_{P_\pi}(V^{\pi,P})} \\
&= \sqrt{\frac{1}{1-\gamma}}\sqrt{\sum_{t=0}^{\infty}\gamma^t(P_\pi)^t\,\mathrm{Var}_{P_\pi}(V^{\pi,P})},
\end{aligned}
\tag{86}
$$

813   where (i) holds by Jensen's inequality.

814   To continue, denoting the minimum value of $V$ as $V_{\min} = \min_{s\in\mathcal{S}} V^{\pi,P}(s)$ and $V' := V^{\pi,P} - V_{\min}\mathbf{1}$.
815   We control $\mathrm{Var}_{P_\pi}(V^{\pi,P})$ as follows:

$$
\begin{aligned}
&\mathrm{Var}_{P_\pi}(V^{\pi,P}) \\
&\overset{(i)}{=} \mathrm{Var}_{P_\pi}(V') = P_\pi(V'\circ V') - (P_\pi V')\circ(P_\pi V') \\
&\overset{(ii)}{=} P_\pi(V'\circ V') - \frac{1}{\gamma^2}(V' - r_\pi + (1-\gamma)V_{\min}\mathbf{1})\circ(V' - r_\pi + (1-\gamma)V_{\min}\mathbf{1}) \\
&= P_\pi(V'\circ V') - \frac{1}{\gamma^2}V'\circ V' + \frac{2}{\gamma^2}V'\circ(r_\pi - (1-\gamma)V_{\min}\mathbf{1}) \\
&\quad - \frac{1}{\gamma^2}(r_\pi - (1-\gamma)V_{\min}\mathbf{1})\circ(r_\pi - (1-\gamma)V_{\min}\mathbf{1}) \\
&\leq P_\pi(V'\circ V') - \frac{1}{\gamma}V'\circ V' + \frac{2}{\gamma^2}\|V'\|_\infty\mathbf{1},
\end{aligned}
\tag{87}
$$

816   where (i) holds by the fact that $\mathrm{Var}_{P_\pi}(V^{\pi,P} - b\mathbf{1}) = \mathrm{Var}_{P_\pi}(V^{\pi,P})$ for any scalar $b$ and $V^{\pi,P} \in \mathbb{R}^S$,
817   (ii) follows from $V' = r_\pi + \gamma P_\pi V^{\pi,P} - V_{\min}\mathbf{1} = r_\pi - (1-\gamma)V_{\min}\mathbf{1} + \gamma P_\pi V'$, and the last line
818   arises from $\frac{1}{\gamma^2}V'\circ V' \geq \frac{1}{\gamma}V'\circ V'$ and $\|r_\pi - (1-\gamma)V_{\min}\mathbf{1}\|_\infty \leq 1$. Plugging (87) back to (86)
819   leads to

$$
\begin{aligned}
(I - \gamma P_\pi)^{-1}\sqrt{\mathrm{Var}_{P_\pi}(V^{\pi,P})} &\leq \sqrt{\frac{1}{1-\gamma}}\sqrt{\sum_{t=0}^{\infty}\gamma^t(P_\pi)^t\left(P_\pi(V'\circ V') - \frac{1}{\gamma}V'\circ V' + \frac{2}{\gamma^2}\|V'\|_\infty\mathbf{1}\right)} \\
&\overset{(i)}{\leq} \sqrt{\frac{1}{1-\gamma}}\sqrt{\left|\sum_{t=0}^{\infty}\gamma^t(P_\pi)^t\left(P_\pi(V'\circ V') - \frac{1}{\gamma}V'\circ V'\right)\right|} + \sqrt{\frac{1}{1-\gamma}}\sqrt{\sum_{t=0}^{\infty}\gamma^t(P_\pi)^t\frac{2}{\gamma^2}\|V'\|_\infty\mathbf{1}} \\
&\leq \sqrt{\frac{1}{1-\gamma}}\sqrt{\left|\left(\sum_{t=0}^{\infty}\gamma^t(P_\pi)^{t+1} - \sum_{t=0}^{\infty}\gamma^{t-1}(P_\pi)^t\right)(V'\circ V')\right|} + \sqrt{\frac{2\|V'\|_\infty\mathbf{1}}{\gamma^2(1-\gamma)^2}} \\
&\overset{(ii)}{\leq} \sqrt{\frac{\|V'\|_\infty^2\mathbf{1}}{\gamma(1-\gamma)}} + \sqrt{\frac{2\|V'\|_\infty\mathbf{1}}{\gamma^2(1-\gamma)^2}} \\
&\leq \sqrt{\frac{8\|V'\|_\infty\mathbf{1}}{\gamma^2(1-\gamma)^2}},
\end{aligned}
\tag{88}
$$

820   where (i) holds by the triangle inequality, (ii) holds by following recursion, and the last inequality
821   holds by $\|V'\|_\infty \leq \frac{1}{1-\gamma}$.

822   **C.3.3   Proof of Lemma 9**

823   **Step 1: controlling the point-wise concentration.**   We first consider a more general term w.r.t. any
824   fixed (independent from $\widehat{P}^0$) value vector $V$ obeying $0 \leq V \leq \frac{1}{1-\gamma}\mathbf{1}$ and any policy $\pi$. Invoking

Lemma 4 leads to that for any $(s, a) \in \mathcal{S} \times \mathcal{A}$,

$$
\left| \widehat{P}_{s,a}^{\pi,V} V - P_{s,a}^{\pi,V} V \right| \leq \left| \max_{\alpha \in [\min_s V(s), \max_s V(s)]} \left\{ \widehat{P}_{s,a}^0 [V]_\alpha - \sigma \left( \alpha - \min_{s'} [V]_\alpha (s') \right) \right\} \right.
$$

$$
\left. - \max_{\alpha \in [\min_s V(s), \max_s V(s)]} \left\{ P_{s,a}^0 [V]_\alpha - w\sigma \left( \alpha - \min_{s'} [V]_\alpha (s') \right) \right\} \right|
$$

$$
\leq \max_{\alpha \in [\min_s V(s), \max_s V(s)]} \underbrace{\left| \left( P_{s,a}^0 - \widehat{P}_{s,a}^0 \right) [V]_\alpha \right|}_{=: g_{s,a}(\alpha, V)}, \tag{89}
$$

where the last inequality holds by that the maximum operator is 1-Lipschitz.

Then for a fixed $\alpha$ and any vector $V$ that is independent with $\widehat{P}^0$, using the Bernstein's inequality, one has with probability at least $1 - \delta$,

$$
g_{s,a}(\alpha, V) = \left| \left( P_{s,a}^0 - \widehat{P}_{s,a}^0 \right) [V]_\alpha \right| \leq \sqrt{\frac{2 \log(\frac{2}{\delta})}{N}} \sqrt{\mathrm{Var}_{P_{s,a}^0}([V]_\alpha)} + \frac{2 \log(\frac{2}{\delta})}{3N(1-\gamma)}
$$

$$
\leq \sqrt{\frac{2 \log(\frac{2}{\delta})}{N}} \sqrt{\mathrm{Var}_{P_{s,a}^0}(V)} + \frac{2 \log(\frac{2}{\delta})}{3N(1-\gamma)}. \tag{90}
$$

**Step 2: deriving the uniform concentration.** To obtain the union bound, we first notice that $g_{s,a}(\alpha, V)$ is 1-Lipschitz w.r.t. $\alpha$ for any $V$ obeying $\|V\|_\infty \leq \frac{1}{1-\gamma}$. In addition, we can construct an $\varepsilon_1$-net $N_{\varepsilon_1}$ over $[0, \frac{1}{1-\gamma}]$ whose size satisfies $|N_{\varepsilon_1}| \leq \frac{3}{\varepsilon_1(1-\gamma)}$ (Vershynin, 2018). By the union bound and (90), it holds with probability at least $1 - \frac{\delta}{SA}$ that for all $\alpha \in N_{\varepsilon_1}$,

$$
g_{s,a}(\alpha, V) \leq \sqrt{\frac{2 \log(\frac{2SA|N_{\varepsilon_1}|}{\delta})}{N}} \sqrt{\mathrm{Var}_{P_{s,a}^0}(V)} + \frac{2 \log(\frac{2SA|N_{\varepsilon_1}|}{\delta})}{3N(1-\gamma)}. \tag{91}
$$

Combined with (89), it yields that,

$$
\left| \widehat{P}_{s,a}^{\pi,V} V - P_{s,a}^{\pi,V} V \right| \leq \max_{\alpha \in [\min_s V(s), \max_s V(s)]} \left| \left( P_{s,a}^0 - \widehat{P}_{s,a}^0 \right) [V]_\alpha \right|
$$

$$
\overset{(i)}{\leq} \varepsilon_1 + \sup_{\alpha \in N_{\varepsilon_1}} \left| \left( P_{s,a}^0 - \widehat{P}_{s,a}^0 \right) [V]_\alpha \right|
$$

$$
\overset{(ii)}{\leq} \varepsilon_1 + \sqrt{\frac{2 \log(\frac{2SA|N_{\varepsilon_1}|}{\delta})}{N}} \sqrt{\mathrm{Var}_{P_{s,a}^0}(V)} + \frac{2 \log(\frac{2SA|N_{\varepsilon_1}|}{\delta})}{3N(1-\gamma)} \tag{92}
$$

$$
\overset{(iii)}{\leq} \sqrt{\frac{2 \log(\frac{2SA|N_{\varepsilon_1}|}{\delta})}{N}} \sqrt{\mathrm{Var}_{P_{s,a}^0}(V)} + \frac{\log(\frac{2SA|N_{\varepsilon_1}|}{\delta})}{N(1-\gamma)}
$$

$$
\overset{(iv)}{\leq} 2\sqrt{\frac{\log(\frac{18SAN}{\delta})}{N}} \sqrt{\mathrm{Var}_{P_{s,a}^0}(V)} + \frac{\log(\frac{18SAN}{\delta})}{N(1-\gamma)} \tag{93}
$$

$$
\leq 2\sqrt{\frac{\log(\frac{18SAN}{\delta})}{N}} \|V\|_\infty + \frac{\log(\frac{18SAN}{\delta})}{N(1-\gamma)}
$$

$$
\leq 3\sqrt{\frac{\log(\frac{18SAN}{\delta})}{(1-\gamma)^2 N}} \tag{94}
$$

where (i) follows from that the optimal $\alpha^\star$ falls into the $\varepsilon_1$-ball centered around some point inside $N_{\varepsilon_1}$ and $g_{s,a}(\alpha, V)$ is 1-Lipschitz, (ii) holds by (91), (iii) arises from taking $\varepsilon_1 = \frac{\log(\frac{2SA|N_{\varepsilon_1}|}{\delta})}{3N(1-\gamma)}$, (iv) is verified by $|N_{\varepsilon_1}| \leq \frac{3}{\varepsilon_1(1-\gamma)} \leq 9N$, and the last inequality is due to the fact $\|V^{\star,\sigma}\|_\infty \leq \frac{1}{1-\gamma}$ and letting $N \geq \log(\frac{18SAN}{\delta})$.

To continue, applying (93) and (94) with $\pi = \pi^\star$ and $V = V^{\star,\sigma}$ (independent with $\widehat{P}^0$) and taking the union bound over $(s,a) \in \mathcal{S} \times \mathcal{A}$ gives that with probability at least $1-\delta$, it holds simultaneously for all $(s,a) \in \mathcal{S} \times \mathcal{A}$ that

$$\left| \widehat{P}_{s,a}^{\pi^\star,V} V^{\star,\sigma} - P_{s,a}^{\pi^\star,V} V^{\star,\sigma} \right| \leq 2\sqrt{\frac{\log(\frac{18SAN}{\delta})}{N}} \sqrt{\operatorname{Var}_{P_{s,a}^0}(V^{\star,\sigma})} + \frac{\log(\frac{18SAN}{\delta})}{N(1-\gamma)}$$

$$\leq 3\sqrt{\frac{\log(\frac{18SAN}{\delta})}{(1-\gamma)^2 N}}. \tag{95}$$

By converting (95) to the matrix form, one has with probability at least $1-\delta$,

$$\left| \widehat{\underline{P}}^{\pi^\star,V} V^{\pi^\star,\sigma} - \underline{P}^{\pi^\star,V} V^{\pi^\star,\sigma} \right| \leq 2\sqrt{\frac{\log(\frac{18SAN}{\delta})}{N}} \sqrt{\operatorname{Var}_{P^{\pi^\star}}(V^{\star,\sigma})} + \frac{\log(\frac{18SAN}{\delta})}{N(1-\gamma)} 1$$

$$\leq 3\sqrt{\frac{\log(\frac{18SAN}{\delta})}{(1-\gamma)^2 N}} 1. \tag{96}$$

### C.3.4   Proof of Lemma 10

Following the same argument as (86), it follows

$$\left( I - \gamma \widehat{\underline{P}}^{\pi^\star,V} \right)^{-1} \sqrt{\operatorname{Var}_{\widehat{\underline{P}}^{\pi^\star,V}}(V^{\star,\sigma})} = \sqrt{\frac{1}{1-\gamma}} \sqrt{\sum_{t=0}^{\infty} \gamma^t \left( \widehat{\underline{P}}^{\pi^\star,V} \right)^t \operatorname{Var}_{\widehat{\underline{P}}^{\pi^\star,V}}(V^{\star,\sigma})}. \tag{97}$$

To continue, we first focus on controlling $\operatorname{Var}_{\widehat{\underline{P}}^{\pi^\star,V}}(V^{\star,\sigma})$. Towards this, denoting the minimum value of $V^{\star,\sigma}$ as $V_{\min} := \min_{s \in \mathcal{S}} V^{\star,\sigma}(s)$ and $V' := V^{\star,\sigma} - V_{\min}1$, we arrive at (see the robust Bellman's consistency equation in (41))

$$V' = V^{\star,\sigma} - V_{\min}1 = r_{\pi^\star} + \gamma \underline{P}^{\pi^\star,V} V^{\star,\sigma} - V_{\min}1$$

$$= r_{\pi^\star} + \gamma \widehat{\underline{P}}^{\pi^\star,V} V^{\star,\sigma} + \gamma \left( \underline{P}^{\pi^\star,V} - \widehat{\underline{P}}^{\pi^\star,V} \right) V^{\star,\sigma} - V_{\min}1$$

$$= r_{\pi^\star} - (1-\gamma)V_{\min}1 + \gamma \widehat{\underline{P}}^{\pi^\star,V} V' + \gamma \left( \underline{P}^{\pi^\star,V} - \widehat{\underline{P}}^{\pi^\star,V} \right) V^{\star,\sigma}$$

$$= r'_{\pi^\star} + \gamma \widehat{\underline{P}}^{\pi^\star,V} V' + \gamma \left( \underline{P}^{\pi^\star,V} - \widehat{\underline{P}}^{\pi^\star,V} \right) V^{\star,\sigma}, \tag{98}$$

where the last line holds by letting $r'_{\pi^\star} := r_{\pi^\star} - (1-\gamma)V_{\min}1 \leq r_{\pi^\star}$. With the above fact in hand, we control $\operatorname{Var}_{\widehat{\underline{P}}^{\pi^\star,V}}(V^{\star,\sigma})$ as follows:

$$\operatorname{Var}_{\widehat{\underline{P}}^{\pi^\star,V}}(V^{\star,\sigma}) \overset{(i)}{=} \operatorname{Var}_{\widehat{\underline{P}}^{\pi^\star,V}}(V') = \widehat{\underline{P}}^{\pi^\star,V}(V' \circ V') - (\widehat{\underline{P}}^{\pi^\star,V} V') \circ (\widehat{\underline{P}}^{\pi^\star,V} V')$$

$$\overset{(ii)}{=} \widehat{\underline{P}}^{\pi^\star,V}(V' \circ V') - \frac{1}{\gamma^2} \left( V' - r'_{\pi^\star} - \gamma \left( \underline{P}^{\pi^\star,V} - \widehat{\underline{P}}^{\pi^\star,V} \right) V^{\star,\sigma} \right)^{\circ 2}$$

$$= \widehat{\underline{P}}^{\pi^\star,V}(V' \circ V') - \frac{1}{\gamma^2}V' \circ V' + \frac{2}{\gamma^2}V' \circ \left( r'_{\pi^\star} + \gamma \left( \underline{P}^{\pi^\star,V} - \widehat{\underline{P}}^{\pi^\star,V} \right) V^{\star,\sigma} \right)$$

$$\quad - \frac{1}{\gamma^2}\left( r'_{\pi^\star} + \gamma \left( \underline{P}^{\pi^\star,V} - \widehat{\underline{P}}^{\pi^\star,V} \right) V^{\star,\sigma} \right)^{\circ 2}$$

$$\overset{(iii)}{\leq} \widehat{\underline{P}}^{\pi^\star,V}(V' \circ V') - \frac{1}{\gamma}V' \circ V' + \frac{2}{\gamma^2}\|V'\|_\infty 1 + \frac{2}{\gamma}\|V'\|_\infty \left| \left( \underline{P}^{\pi^\star,V} - \widehat{\underline{P}}^{\pi^\star,V} \right) V^{\star,\sigma} \right| \tag{99}$$

$$\leq \widehat{\underline{P}}^{\pi^\star,V}(V' \circ V') - \frac{1}{\gamma}V' \circ V' + \frac{2}{\gamma^2}\|V'\|_\infty 1 + \frac{6}{\gamma}\|V'\|_\infty \sqrt{\frac{\log(\frac{18SAN}{\delta})}{(1-\gamma)^2 N}} 1, \tag{100}$$

where (i) holds by the fact that $\operatorname{Var}_{P_\pi}(V - b1) = \operatorname{Var}_{P_\pi}(V)$ for any scalar $b$ and $V \in \mathbb{R}^S$, (ii) follows from (98), (iii) arises from $\frac{1}{\gamma^2}V' \circ V' \geq \frac{1}{\gamma}V' \circ V'$ and $-1 \leq r_{\pi^\star} - (1-\gamma)V_{\min}1 = r'_{\pi^\star} \leq r_{\pi^\star} \leq 1$, and the last inequality holds by Lemma 9.

Plugging (100) into (97) leads to

$$
\left(I - \gamma \underline{\widehat{P}}^{\pi^\star, V}\right)^{-1} \sqrt{\mathrm{Var}_{\widehat{\underline{P}}^{\pi^\star, V}}(V^{\star, \sigma})}
$$

$$
\leq \sqrt{\frac{1}{1-\gamma}} \sqrt{\sum_{t=0}^{\infty} \gamma^t \left(\underline{\widehat{P}}^{\pi^\star, V}\right)^t \left(\underline{\widehat{P}}^{\pi^\star, V}(V' \circ V') - \frac{1}{\gamma} V' \circ V' + \frac{2}{\gamma^2}\|V'\|_\infty 1 + \frac{6}{\gamma}\|V'\|_\infty \sqrt{\frac{\log(\frac{18SAN}{\delta})}{(1-\gamma)^2 N}} 1 \right)}
$$

$$
\overset{(i)}{\leq} \sqrt{\frac{1}{1-\gamma}} \sqrt{\left| \sum_{t=0}^{\infty} \gamma^t \left(\underline{\widehat{P}}^{\pi^\star, V}\right)^t \left(\underline{\widehat{P}}^{\pi^\star, V}(V' \circ V') - \frac{1}{\gamma} V' \circ V'\right) \right|}
$$

$$
+ \sqrt{\frac{1}{1-\gamma}} \sqrt{\sum_{t=0}^{\infty} \gamma^t \left(\underline{\widehat{P}}^{\pi^\star, V}\right)^t \left(\frac{2}{\gamma^2}\|V'\|_\infty 1 + \frac{6}{\gamma}\|V'\|_\infty \sqrt{\frac{\log(\frac{18SAN}{\delta})}{(1-\gamma)^2 N}} 1 \right)}
$$

$$
\leq \sqrt{\frac{1}{1-\gamma}} \sqrt{\left| \sum_{t=0}^{\infty} \gamma^t \left(\underline{\widehat{P}}^{\pi^\star, V}\right)^t \left[\underline{\widehat{P}}^{\pi^\star, V}(V' \circ V') - \frac{1}{\gamma} V' \circ V'\right] \right|} + \sqrt{\frac{\left(2 + 6\sqrt{\frac{\log(\frac{18SAN}{\delta})}{(1-\gamma)^2 N}}\right)\|V'\|_\infty}{(1-\gamma)^2 \gamma^2}} 1,
$$

(101)

where (i) holds by the triangle inequality. Therefore, the remainder of the proof shall focus on the first term, which follows

$$
\left| \sum_{t=0}^{\infty} \gamma^t \left(\underline{\widehat{P}}^{\pi^\star, V}\right)^t \left(\underline{\widehat{P}}^{\pi^\star, V}(V' \circ V') - \frac{1}{\gamma} V' \circ V'\right) \right|
$$

$$
= \left| \left(\sum_{t=0}^{\infty} \gamma^t \left(\underline{\widehat{P}}^{\pi^\star, V}\right)^{t+1} - \sum_{t=0}^{\infty} \gamma^{t-1} \left(\underline{\widehat{P}}^{\pi^\star, V}\right)^t \right)(V' \circ V') \right| \leq \frac{1}{\gamma} \|V'\|_\infty^2 1
$$

(102)

by recursion. Inserting (102) back to (101) leads to

$$
\left(I - \gamma \underline{\widehat{P}}^{\pi^\star, V}\right)^{-1} \sqrt{\mathrm{Var}_{\widehat{\underline{P}}^{\pi^\star, V}}(V^{\star, \sigma})}
$$

$$
\leq \sqrt{\frac{\|V'\|_\infty^2}{\gamma(1-\gamma)}} 1 + 3\sqrt{\frac{\left(1 + \sqrt{\frac{\log(\frac{18SAN}{\delta})}{(1-\gamma)^2 N}}\right)\|V'\|_\infty}{(1-\gamma)^2 \gamma^2}} 1
$$

$$
\leq 4\sqrt{\frac{\left(1 + \sqrt{\frac{\log(\frac{18SAN}{\delta})}{(1-\gamma)^2 N}}\right)\|V'\|_\infty}{(1-\gamma)^2 \gamma^2}} 1 \leq 4\sqrt{\frac{\left(1 + \sqrt{\frac{\log(\frac{18SAN}{\delta})}{(1-\gamma)^2 N}}\right)}{\gamma^3(1-\gamma)^2 \max\{1-\gamma, \sigma\}}} 1
$$

$$
\leq 4\sqrt{\frac{\left(1 + \sqrt{\frac{\log(\frac{18SAN}{\delta})}{(1-\gamma)^2 N}}\right)}{\gamma^3(1-\gamma)^3}} 1,
$$

(103)

where the penultimate inequality follows from applying Lemma 7 with $P = P^0$ and $\pi = \pi^\star$:

$$
\|V'\|_\infty = \max_{s \in \mathcal{S}} V^{\star, \sigma}(s) - \min_{s \in \mathcal{S}} V^{\star, \sigma}(s) \leq \frac{1}{\gamma \max\{1-\gamma, \sigma\}}.
$$

### C.3.5 Proof of Lemma 12

To begin with, for any $(s, a) \in \mathcal{S} \times \mathcal{A}$, invoking the results in (89), we have

$$
\left| \widehat{P}_{s,a}^{\widehat{\pi}, \widehat{V}} \widehat{V}^{\widehat{\pi}, \sigma} - P_{s,a}^{\widehat{\pi}, \widehat{V}} \widehat{V}^{\widehat{\pi}, \sigma} \right| \leq \max_{\alpha \in [\min_s \widehat{V}^{\widehat{\pi}, \sigma}(s), \max_s \widehat{V}^{\widehat{\pi}, \sigma}(s)]} \left| \left(P_{s,a}^0 - \widehat{P}_{s,a}^0\right)\left[\widehat{V}^{\widehat{\pi}, \sigma}\right]_\alpha \right|
$$

$$
\overset{(i)}{\leq} \max_{\alpha \in [\min_s \widehat{V}^{\widehat{\pi}, \sigma}(s), \max_s \widehat{V}^{\widehat{\pi}, \sigma}(s)]} \left( \left| \left(P_{s,a}^0 - \widehat{P}_{s,a}^0\right)\left[\widehat{V}^{\star, \sigma}\right]_\alpha \right| + \left| \left(P_{s,a}^0 - \widehat{P}_{s,a}^0\right)\left(\left[\widehat{V}^{\widehat{\pi}, \sigma}\right]_\alpha - \left[\widehat{V}^{\star, \sigma}\right]_\alpha\right) \right| \right)
$$

$$\leq \max_{\alpha \in [\min_s \widehat{V}^{\widehat{\pi},\sigma}(s), \max_s \widehat{V}^{\widehat{\pi},\sigma}(s)]} \left( \left| \left( P_{s,a}^0 - \widehat{P}_{s,a}^0 \right) \left[ \widehat{V}^{\star,\sigma} \right]_\alpha \right| + \left\| P_{s,a}^0 - \widehat{P}_{s,a}^0 \right\|_1 \left\| \left[ \widehat{V}^{\widehat{\pi},\sigma} \right]_\alpha - \left[ \widehat{V}^{\star,\sigma} \right]_\alpha \right\|_\infty \right)$$

$$\overset{\text{(ii)}}{\leq} \max_{\alpha \in [\min_s \widehat{V}^{\widehat{\pi},\sigma}(s), \max_s \widehat{V}^{\widehat{\pi},\sigma}(s)]} \left| \left( P_{s,a}^0 - \widehat{P}_{s,a}^0 \right) \left[ \widehat{V}^{\star,\sigma} \right]_\alpha \right| + 2 \left\| \widehat{V}^{\widehat{\pi},\sigma} - \widehat{V}^{\star,\sigma} \right\|_\infty$$

$$\overset{\text{(iii)}}{\leq} \max_{\alpha \in [\min_s \widehat{V}^{\widehat{\pi},\sigma}(s), \max_s \widehat{V}^{\widehat{\pi},\sigma}(s)]} \left| \left( P_{s,a}^0 - \widehat{P}_{s,a}^0 \right) [\widehat{V}^{\star,\sigma}]_\alpha \right| + \frac{2\gamma\varepsilon_{\text{opt}}}{1-\gamma}, \tag{104}$$

where (i) holds by the triangle inequality, and (ii) follows from $\left\| P_{s,a}^0 - \widehat{P}_{s,a}^0 \right\|_1 \leq 2$ and $\left\| [\widehat{V}^{\widehat{\pi},\sigma}]_\alpha - [\widehat{V}^{\star,\sigma}]_\alpha \right\|_\infty \leq \left\| \widehat{V}^{\widehat{\pi},\sigma} - \widehat{V}^{\star,\sigma} \right\|_\infty$, and (iii) follows from (48).

To control $\left| \left( P_{s,a}^0 - \widehat{P}_{s,a}^0 \right) [\widehat{V}^{\star,\sigma}]_\alpha \right|$ in (104) for any given $\alpha \in \left[ 0, \frac{1}{1-\gamma} \right]$, and tame the dependency between $\widehat{V}^{\star,\sigma}$ and $\widehat{P}^0$, we resort to the following leave-one-out argument motivated by (Agarwal et al., 2020; Li et al., 2022a; Shi and Chi, 2022). Specifically, we first construct a set of auxiliary RMDPs which simultaneously have the desired statistical independence between robust value functions and the estimated nominal transition kernel, and are minimally different from the original RMDPs under consideration. Then we control the term of interest associated with these auxiliary RMDPs and show the value is close to the target quantity for the desired RMDP. The process is divided into several steps as below.

**Step 1: construction of auxiliary RMDPs with deterministic empirical nominal transitions.** Recall that we target the empirical infinite-horizon robust MDP $\widehat{\mathcal{M}}_{\text{rob}}$ with the nominal transition kernel $\widehat{P}^0$. Towards this, we can construct an auxiliary robust MDP $\widehat{\mathcal{M}}_{\text{rob}}^{s,u}$ for each state $s$ and any non-negative scalar $u \geq 0$, so that it is the same as $\widehat{\mathcal{M}}_{\text{rob}}$ except for the transition properties in state $s$. In particular, we define the nominal transition kernel and reward function of $\widehat{\mathcal{M}}_{\text{rob}}^{s,u}$ as $P^{s,u}$ and $r^{s,u}$, which are expressed as follows

$$\begin{cases} P^{s,u}(s' \,|\, s, a) = \mathbb{1}(s' = s) & \text{for all } (s', a) \in \mathcal{S} \times \mathcal{A}, \\ P^{s,u}(\cdot \,|\, \widetilde{s}, a) = \widehat{P}^0(\cdot \,|\, \widetilde{s}, a) & \text{for all } (\widetilde{s}, a) \in \mathcal{S} \times \mathcal{A} \text{ and } \widetilde{s} \neq s, \end{cases} \tag{105}$$

and

$$\begin{cases} r^{s,u}(s, a) = u & \text{for all } a \in \mathcal{A}, \\ r^{s,u}(\widetilde{s}, a) = r(\widetilde{s}, a) & \text{for all } (\widetilde{s}, a) \in \mathcal{S} \times \mathcal{A} \text{ and } \widetilde{s} \neq s. \end{cases} \tag{106}$$

It is evident that the nominal transition probability at state $s$ of the auxiliary $\widehat{\mathcal{M}}_{\text{rob}}^{s,u}$, i.e. it never leaves state $s$ once entered. This useful property removes the randomness of $\widehat{P}_{s,a}^0$ for all $a \in \mathcal{A}$ in state $s$, which will be leveraged later.

Correspondingly, the robust Bellman operator $\widehat{\mathcal{T}}_{s,u}^\sigma(\cdot)$ associated with the RMDP $\widehat{\mathcal{M}}_{\text{rob}}^{s,u}$ is defined as

$$\forall (\tilde{s}, a) \in \mathcal{S} \times \mathcal{A}: \quad \widehat{\mathcal{T}}_{s,u}^\sigma(Q)(\tilde{s}, a) = r^{s,u}(\tilde{s}, a) + \gamma \inf_{\mathcal{P} \in \mathcal{U}^\sigma(P_{\tilde{s},a}^{s,u})} \mathcal{P}V, \quad \text{with } V(\tilde{s}) = \max_a Q(\tilde{s}, a). \tag{107}$$

**Step 2: fixed-point equivalence between $\widehat{\mathcal{M}}_{\text{rob}}$ and the auxiliary RMDP $\widehat{\mathcal{M}}_{\text{rob}}^{s,u}$.** Recall that $\widehat{Q}^{\star,\sigma}$ is the unique fixed point of $\widehat{\mathcal{T}}^\sigma(\cdot)$ with the corresponding robust value $\widehat{V}^{\star,\sigma}$. We assert that the corresponding robust value function $\widehat{V}_{s,u^\star}^{\star,\sigma}$ obtained from the fixed point of $\widehat{\mathcal{T}}_{s,u}^\sigma(\cdot)$ aligns with the robust value function $\widehat{V}^{\star,\sigma}$ derived from $\widehat{\mathcal{T}}^\sigma(\cdot)$, as long as we choose $u$ in the following manner:

$$u^\star := u^\star(s) = \widehat{V}^{\star,\sigma}(s) - \gamma \inf_{\mathcal{P} \in \mathcal{U}^\sigma(e_s)} \mathcal{P}\widehat{V}^{\star,\sigma}. \tag{108}$$

where $e_s$ is the $s$-th standard basis vector in $\mathbb{R}^S$. Towards verifying this, we shall break our arguments in two different cases.

- **For state $s$:** One has for any $a \in \mathcal{A}$:

$$r^{s,u^\star}(s, a) + \gamma \inf_{\mathcal{P} \in \mathcal{U}^\sigma(P_{s,a}^{s,u^\star})} \mathcal{P}\widehat{V}^{\star,\sigma} = u^\star + \gamma \inf_{\mathcal{P} \in \mathcal{U}^\sigma(e_s)} \mathcal{P}\widehat{V}^{\star,\sigma}$$

$$= \widehat{V}^{\star,\sigma}(s) - \gamma \inf_{\mathcal{P} \in \mathcal{U}^\sigma(e_s)} \mathcal{P}\widehat{V}^{\star,\sigma} + \gamma \inf_{\mathcal{P} \in \mathcal{U}^\sigma(e_s)} \mathcal{P}\widehat{V}^{\star,\sigma} = \widehat{V}^{\star,\sigma}(s), \qquad (109)$$

where the first equality follows from the definition of $P_{s,a}^{s,u^\star}$ in (105), and the second equality follows from plugging in the definition of $u^\star$ in (108).

- **For state** $s' \neq s$: It is easily verified that for all $a \in \mathcal{A}$,

$$r^{s,u^\star}(s',a) + \gamma \inf_{\mathcal{P} \in \mathcal{U}^\sigma(P_{s',a}^{s,u^\star})} \mathcal{P}\widehat{V}^{\star,\sigma} = r(s',a) + \gamma \inf_{\mathcal{P} \in \mathcal{U}^\sigma(\widehat{P}_{s',a}^0)} \mathcal{P}\widehat{V}^{\star,\sigma}$$

$$= \widehat{\mathcal{T}}^\sigma(\widehat{Q}^{\star,\sigma})(s',a) = \widehat{Q}^{\star,\sigma}(s',a), \qquad (110)$$

where the first equality follows from the definitions in (106) and (105), and the last line arises from the definition of the robust Bellman operator in (12), and that $\widehat{Q}^{\star,\sigma}$ is the fixed point of $\widehat{\mathcal{T}}^\sigma(\cdot)$ (see Lemma 3).

Combining the facts in the above two cases, we establish that there exists a fixed point $\widehat{Q}_{s,u^\star}^{\star,\sigma}$ of the operator $\widehat{\mathcal{T}}_{s,u^\star}^\sigma(\cdot)$ by taking

$$\begin{cases} \widehat{Q}_{s,u^\star}^{\star,\sigma}(s,a) = \widehat{V}^{\star,\sigma}(s) & \text{for all } a \in \mathcal{A}, \\ \widehat{Q}_{s,u^\star}^{\star,\sigma}(s',a) = \widehat{Q}^{\star,\sigma}(s',a) & \text{for all } s' \neq s \text{ and } a \in \mathcal{A}. \end{cases} \qquad (111)$$

Consequently, we confirm the existence of a fixed point of the operator $\widehat{\mathcal{T}}_{s,u^\star}^\sigma(\cdot)$. In addition, its corresponding value function $\widehat{V}_{s,u^\star}^{\star,\sigma}$ also coincides with $\widehat{V}^{\star,\sigma}$. Note that the corresponding facts between $\widehat{\mathcal{M}}_{\text{rob}}$ and $\widehat{\mathcal{M}}_{\text{rob}}^{s,u}$ in Step 1 and step 2 holds in fact for any uncertainty set.

**Step 3: building an $\varepsilon$-net for all reward values $u$.** It is easily verified that

$$0 \leq u^\star \leq \widehat{V}^{\star,\sigma}(s) \leq \frac{1}{1-\gamma}. \qquad (112)$$

We can construct a $N_{\varepsilon_2}$-net over the interval $\left[0, \frac{1}{1-\gamma}\right]$, where the size is bounded by $|N_{\varepsilon_2}| \leq \frac{3}{\varepsilon_2(1-\gamma)}$ (Vershynin, 2018). Following the same arguments in the proof of Lemma 3, we can demonstrate that for each $u \in N_{\varepsilon_2}$, there exists a unique fixed point $\widehat{Q}_{s,u}^{\star,\sigma}$ of the operator $\widehat{\mathcal{T}}_{s,u}^\sigma(\cdot)$, which satisfies $0 \leq \widehat{Q}_{s,u}^{\star,\sigma} \leq \frac{1}{1-\gamma} \cdot \mathbf{1}$. Consequently, the corresponding robust value function also satisfies $\left\| \widehat{V}_{s,u}^{\star,\sigma} \right\|_\infty \leq \frac{1}{1-\gamma}$.

By the definitions in (105) and (106), we observe that for all $u \in N_{\varepsilon_2}$, $\widehat{\mathcal{M}}_{\text{rob}}^{s,u}$ is statistically independent from $\widehat{P}_{s,a}^0$. This independence indicates that $[\widehat{V}_{s,u}^{\star,\sigma}]_\alpha$ and $\widehat{P}_{s,a}^0$ are independent for a fixed $\alpha$. With this in mind, invoking the fact in (93) and (94) and taking the union bound over all $(s,a,\alpha) \in \mathcal{S} \times \mathcal{A} \times N_{\varepsilon_1}$, $u \in N_{\varepsilon_2}$ yields that, with probability at least $1 - \delta$, it holds for all $(s,a,u) \in \mathcal{S} \times \mathcal{A} \times N_{\varepsilon_2}$ that

$$\max_{\alpha \in [0, 1/(1-\gamma)]} \left| \left( P_{s,a}^0 - \widehat{P}_{s,a}^0 \right) \left[ \widehat{V}_{s,u}^{\star,\sigma} \right]_\alpha \right|$$

$$\leq \varepsilon_2 + 2\sqrt{\frac{\log(\frac{18SAN|N_{\varepsilon_2}|}{\delta})}{N}} \sqrt{\text{Var}_{P_{s,a}^0}(\widehat{V}_{s,u}^{\star,\sigma})} + \frac{2\log(\frac{18SAN|N_{\varepsilon_2}|}{\delta})}{3N(1-\gamma)}$$

$$\leq \varepsilon_2 + 3\sqrt{\frac{\log(\frac{18SAN|N_{\varepsilon_2}|}{\delta})}{(1-\gamma)^2 N}}, \qquad (113)$$

where the last inequality holds by the fact $\text{Var}_{P_{s,a}^0}(\widehat{V}_{s,u}^{\star,\sigma}) \leq \|\widehat{V}_{s,u}^{\star,\sigma}\|_\infty \leq \frac{1}{1-\gamma}$ and letting $N \geq \log\left(\frac{18SAN|N_{\varepsilon_2}|}{\delta}\right)$.

 **Step 4: uniform concentration.** Recalling that $u^\star \in \left[0, \frac{1}{1-\gamma}\right]$ (see (112)), we can always find
some $\overline{u} \in N_{\varepsilon_2}$ such that $|\overline{u} - u^\star| \le \varepsilon_2$. Consequently, plugging in the operator $\widehat{\mathcal{T}}_{s,u}^\sigma(\cdot)$ in (107) yields

$$\forall Q \in \mathbb{R}^{SA} : \quad \left\| \widehat{\mathcal{T}}_{s,\overline{u}}^\sigma(Q) - \widehat{\mathcal{T}}_{s,u^\star}^\sigma(Q) \right\|_\infty = |\overline{u} - u^\star| \le \varepsilon_2$$

With this in mind, we observe that the fixed points of $\widehat{\mathcal{T}}_{s,\overline{u}}^\sigma(\cdot)$ and $\widehat{\mathcal{T}}_{s,u^\star}^\sigma(\cdot)$ obey

$$
\begin{aligned}
\left\| \widehat{Q}_{s,\overline{u}}^{\star,\sigma} - \widehat{Q}_{s,u^\star}^{\star,\sigma} \right\|_\infty &= \left\| \widehat{\mathcal{T}}_{s,\overline{u}}^\sigma(\widehat{Q}_{s,\overline{u}}^{\star,\sigma}) - \widehat{\mathcal{T}}_{s,u^\star}^\sigma(\widehat{Q}_{s,u^\star}^{\star,\sigma}) \right\|_\infty \\
&\le \left\| \widehat{\mathcal{T}}_{s,\overline{u}}^\sigma(\widehat{Q}_{s,\overline{u}}^{\star,\sigma}) - \widehat{\mathcal{T}}_{s,\overline{u}}^\sigma(\widehat{Q}_{s,u^\star}^{\star,\sigma}) \right\|_\infty + \left\| \widehat{\mathcal{T}}_{s,\overline{u}}^\sigma(\widehat{Q}_{s,u^\star}^{\star,\sigma}) - \widehat{\mathcal{T}}_{s,u^\star}^\sigma(\widehat{Q}_{s,u^\star}^{\star,\sigma}) \right\|_\infty \\
&\le \gamma \left\| \widehat{Q}_{s,\overline{u}}^{\star,\sigma} - \widehat{Q}_{s,u^\star}^{\star,\sigma} \right\|_\infty + \varepsilon_2,
\end{aligned}
$$

where the last inequality holds by the fact that $\widehat{\mathcal{T}}_{s,u}^\sigma(\cdot)$ is a $\gamma$-contraction. It directly indicates that

$$\left\| \widehat{Q}_{s,\overline{u}}^{\star,\sigma} - \widehat{Q}_{s,u^\star}^{\star,\sigma} \right\|_\infty \le \frac{\varepsilon_2}{(1-\gamma)} \quad \text{and} \quad \left\| \widehat{V}_{s,\overline{u}}^{\star,\sigma} - \widehat{V}_{s,u^\star}^{\star,\sigma} \right\|_\infty \le \left\| \widehat{Q}_{s,\overline{u}}^{\star,\sigma} - \widehat{Q}_{s,u^\star}^{\star,\sigma} \right\|_\infty \le \frac{\varepsilon_2}{(1-\gamma)}. \tag{114}$$

Armed with the above facts, to control the first term in (104), invoking the identity $\widehat{V}^{\star,\sigma} = \widehat{V}_{s,u^\star}^{\star,\sigma}$
established in Step 2 gives that: for all $(s,a) \in \mathcal{S} \times \mathcal{A}$,

$$
\begin{aligned}
&\max_{\alpha \in [\min_s \widehat{V}^{\widehat{\pi},\sigma}(s), \max_s \widehat{V}^{\widehat{\pi},\sigma}(s)]} \left| \left( P_{s,a}^0 - \widehat{P}_{s,a}^0 \right) [\widehat{V}^{\star,\sigma}]_\alpha \right| \\
&\le \max_{\alpha \in [0,1/(1-\gamma)]} \left| \left( P_{s,a}^0 - \widehat{P}_{s,a}^0 \right) [\widehat{V}^{\star,\sigma}]_\alpha \right| = \max_{\alpha \in [0,1/(1-\gamma)]} \left| \left( P_{s,a}^0 - \widehat{P}_{s,a}^0 \right) [\widehat{V}_{s,u^\star}^{\star,\sigma}]_\alpha \right| \\
&\overset{(i)}{\le} \max_{\alpha \in [0,1/(1-\gamma)]} \left\{ \left| \left( P_{s,a}^0 - \widehat{P}_{s,a}^0 \right) [\widehat{V}_{s,\overline{u}}^{\star,\sigma}]_\alpha \right| + \left| \left( P_{s,a}^0 - \widehat{P}_{s,a}^0 \right) \left( [\widehat{V}_{s,\overline{u}}^{\star,\sigma}]_\alpha - [\widehat{V}_{s,u^\star}^{\star,\sigma}]_\alpha \right) \right| \right\} \\
&\overset{(ii)}{\le} \max_{\alpha \in [0,1/(1-\gamma)]} \left| \left( P_{s,a}^0 - \widehat{P}_{s,a}^0 \right) [\widehat{V}_{s,\overline{u}}^{\star,\sigma}]_\alpha \right| + \frac{2\varepsilon_2}{(1-\gamma)} \\
&\overset{(iii)}{\le} \frac{2\varepsilon_2}{(1-\gamma)} + \varepsilon_2 + 2\sqrt{\frac{\log(\frac{18SAN|N_{\varepsilon_2}|}{\delta})}{N}}\sqrt{\mathrm{Var}_{P_{s,a}^0}(\widehat{V}_{s,u}^{\star,\sigma})} + \frac{2\log(\frac{18SAN|N_{\varepsilon_2}|}{\delta})}{3N(1-\gamma)} \\
&\le \frac{3\varepsilon_2}{(1-\gamma)} + 2\sqrt{\frac{\log(\frac{18SAN|N_{\varepsilon_2}|}{\delta})}{N}}\sqrt{\mathrm{Var}_{P_{s,a}^0}(\widehat{V}^{\star,\sigma})} + \frac{2\log(\frac{18SAN|N_{\varepsilon_2}|}{\delta})}{3N(1-\gamma)} \\
&\quad + 2\sqrt{\frac{\log(\frac{18SAN|N_{\varepsilon_2}|}{\delta})}{N}}\sqrt{\left|\mathrm{Var}_{P_{s,a}^0}(\widehat{V}^{\star,\sigma}) - \mathrm{Var}_{P_{s,a}^0}(\widehat{V}_{s,\overline{u}}^{\star,\sigma})\right|} \\
&\overset{(iv)}{\le} \frac{3\varepsilon_2}{(1-\gamma)} + 2\sqrt{\frac{\log(\frac{18SAN|N_{\varepsilon_2}|}{\delta})}{N}}\sqrt{\mathrm{Var}_{P_{s,a}^0}(\widehat{V}^{\star,\sigma})} \\
&\quad + \frac{2\log(\frac{18SAN|N_{\varepsilon_2}|}{\delta})}{3N(1-\gamma)} + 2\sqrt{\frac{2\varepsilon_2\log(\frac{18SAN|N_{\varepsilon_2}|}{\delta})}{N(1-\gamma)^2}} \\
&\le 2\sqrt{\frac{\log(\frac{54SAN^2}{(1-\gamma)\delta})}{N}}\sqrt{\mathrm{Var}_{P_{s,a}^0}(\widehat{V}^{\star,\sigma})} + \frac{8\log(\frac{54SAN^2}{(1-\gamma)\delta})}{N(1-\gamma)} \tag{115} \\
&\le 10\sqrt{\frac{\log(\frac{54SAN^2}{(1-\gamma)\delta})}{(1-\gamma)^2 N}}, \tag{116}
\end{aligned}
$$

where (i) holds by the triangle inequality, (ii) arises from (the last inequality holds by (114))

$$\left| \left( P_{s,a}^0 - \widehat{P}_{s,a}^0 \right) \left( [\widehat{V}_{s,\overline{u}}^{\star,\sigma}]_\alpha - [\widehat{V}_{s,u^\star}^{\star,\sigma}]_\alpha \right) \right| \le \left\| P_{s,a}^0 - \widehat{P}_{s,a}^0 \right\|_1 \left\| [\widehat{V}_{s,\overline{u}}^{\star,\sigma}]_\alpha - [\widehat{V}_{s,u^\star}^{\star,\sigma}]_\alpha \right\|_\infty$$

$$\leq 2\left\|\widehat{V}_{s,\overline{u}}^{\star,\sigma} - \widehat{V}_{s,u^\star}^{\star,\sigma}\right\|_\infty \leq \frac{2\varepsilon_2}{(1-\gamma)}, \tag{117}$$

(iii) follows from (113), (iv) can be verified by applying Lemma 2 with (114). Here, the penultimate inequality holds by letting $\varepsilon_2 = \frac{\log(\frac{18SAN|N_{\varepsilon_2}|}{\delta})}{N}$, which leads to $|N_{\varepsilon_2}| \leq \frac{3}{\varepsilon_2(1-\gamma)} \leq \frac{3N}{1-\gamma}$, and the last inequality holds by the fact $\operatorname{Var}_{P_{s,a}^0}(\widehat{V}^{\star,\sigma}) \leq \|\widehat{V}^{\star,\sigma}\|_\infty \leq \frac{1}{1-\gamma}$ and letting $N \geq \log\left(\frac{54SAN^2}{(1-\gamma)\delta}\right)$.

**Step 5: finishing up.** Inserting (115) and (116) back into (104) and combining with (116) give that with probability at least $1-\delta$,

$$\left|\widehat{P}_{s,a}^{\widehat{\pi},\widehat{V}}\widehat{V}^{\widehat{\pi},\sigma} - P_{s,a}^{\widehat{\pi},\widehat{V}}\widehat{V}^{\widehat{\pi},\sigma}\right| \leq \max_{\alpha \in [\min_s \widehat{V}^{\widehat{\pi},\sigma}(s),\max_s \widehat{V}^{\widehat{\pi},\sigma}(s)]} \left|\left(P_{s,a}^0 - \widehat{P}_{s,a}^0\right)[\widehat{V}^{\star,\sigma}]_\alpha\right| + \frac{2\gamma\varepsilon_{\mathsf{opt}}}{1-\gamma}$$

$$\leq \max_{\alpha \in [0,1/(1-\gamma)]} \left|\left(P_{s,a}^0 - \widehat{P}_{s,a}^0\right)[\widehat{V}^{\star,\sigma}]_\alpha\right| + \frac{2\gamma\varepsilon_{\mathsf{opt}}}{1-\gamma}$$

$$\leq 2\sqrt{\frac{\log(\frac{54SAN^2}{(1-\gamma)\delta})}{N}}\sqrt{\operatorname{Var}_{P_{s,a}^0}(\widehat{V}^{\star,\sigma})} + \frac{8\log(\frac{54SAN^2}{(1-\gamma)\delta})}{N(1-\gamma)} + \frac{2\gamma\varepsilon_{\mathsf{opt}}}{1-\gamma}$$

$$\leq 10\sqrt{\frac{\log(\frac{54SAN^2}{(1-\gamma)\delta})}{(1-\gamma)^2 N}} + \frac{2\gamma\varepsilon_{\mathsf{opt}}}{1-\gamma} \tag{118}$$

holds for all $(s,a) \in \mathcal{S} \times \mathcal{A}$.

Finally, we complete the proof by compiling everything into the matrix form as follows:

$$\left|\widehat{\underline{P}}^{\widehat{\pi},\widehat{V}}\widehat{V}^{\widehat{\pi},\sigma} - \underline{P}^{\widehat{\pi},\widehat{V}}\widehat{V}^{\widehat{\pi},\sigma}\right| \leq 2\sqrt{\frac{\log(\frac{54SAN^2}{(1-\gamma)\delta})}{N}}\sqrt{\operatorname{Var}_{P_{s,a}^0}(\widehat{V}^{\star,\sigma})}\mathbf{1} + \frac{8\log(\frac{54SAN^2}{(1-\gamma)\delta})}{N(1-\gamma)}\mathbf{1} + \frac{2\gamma\varepsilon_{\mathsf{opt}}}{1-\gamma}\mathbf{1}$$

$$\leq 10\sqrt{\frac{\log(\frac{54SAN^2}{(1-\gamma)\delta})}{(1-\gamma)^2 N}}\mathbf{1} + \frac{2\gamma\varepsilon_{\mathsf{opt}}}{1-\gamma}\mathbf{1}. \tag{119}$$

### C.3.6 Proof of Lemma 13

The proof can be achieved by directly applying the same routine as Appendix C.3.4. Towards this, similar to (97), we arrive at

$$\left(I - \gamma\underline{P}^{\widehat{\pi},\widehat{V}}\right)^{-1}\sqrt{\operatorname{Var}_{\underline{P}^{\widehat{\pi},\widehat{V}}}(\widehat{V}^{\widehat{\pi},\sigma})} \leq \sqrt{\frac{1}{1-\gamma}}\sqrt{\sum_{t=0}^\infty \gamma^t\left(\underline{P}^{\widehat{\pi},\widehat{V}}\right)^t\operatorname{Var}_{\underline{P}^{\widehat{\pi},\widehat{V}}}(\widehat{V}^{\widehat{\pi},\sigma})}. \tag{120}$$

To control $\operatorname{Var}_{\underline{P}^{\widehat{\pi},\widehat{V}}}(\widehat{V}^{\widehat{\pi},\sigma})$, we denote the minimum value of $\widehat{V}^{\widehat{\pi},\sigma}$ as $V_{\min} = \min_{s\in\mathcal{S}}\widehat{V}^{\widehat{\pi},\sigma}(s)$ and $V' := \widehat{V}^{\widehat{\pi},\sigma} - V_{\min}\mathbf{1}$. By the same argument as (99), we arrive at

$$\operatorname{Var}_{\underline{P}^{\widehat{\pi},\widehat{V}}}(\widehat{V}^{\widehat{\pi},\sigma}) \leq \underline{P}^{\widehat{\pi},\widehat{V}}(V' \circ V') - \frac{1}{\gamma}V' \circ V' + \frac{2}{\gamma^2}\|V'\|_\infty\mathbf{1} + \frac{2}{\gamma}\|V'\|_\infty\left|\left(\widehat{\underline{P}}^{\widehat{\pi},\widehat{V}} - \underline{P}^{\widehat{\pi},\widehat{V}}\right)\widehat{V}^{\widehat{\pi},\sigma}\right|$$

$$\leq \underline{P}^{\widehat{\pi},\widehat{V}}(V' \circ V') - \frac{1}{\gamma}V' \circ V' + \frac{2}{\gamma^2}\|V'\|_\infty\mathbf{1} + \frac{2}{\gamma}\|V'\|_\infty\left(10\sqrt{\frac{\log(\frac{54SAN^2}{(1-\gamma)\delta})}{(1-\gamma)^2 N}} + \frac{2\gamma\varepsilon_{\mathsf{opt}}}{1-\gamma}\right)\mathbf{1}, \tag{121}$$

where the last inequality makes use of Lemma 12. Plugging (121) back into (120) leads to

$$\left(I - \gamma\underline{P}^{\widehat{\pi},\widehat{V}}\right)^{-1}\sqrt{\operatorname{Var}_{\underline{P}^{\widehat{\pi},\widehat{V}}}(\widehat{V}^{\widehat{\pi},\sigma})} \overset{\text{(i)}}{\leq} \sqrt{\frac{1}{1-\gamma}}\sqrt{\left|\sum_{t=0}^\infty \gamma^t\left(\underline{P}^{\widehat{\pi},\widehat{V}}\right)^t\left(\underline{P}^{\widehat{\pi},\widehat{V}}(V' \circ V') - \frac{1}{\gamma}V' \circ V'\right)\right|}$$

$$+ \sqrt{\frac{1}{(1-\gamma)^2\gamma^2}\left(2 + 20\sqrt{\frac{\log(\frac{54SAN^2}{(1-\gamma)\delta})}{(1-\gamma)^2 N}} + \frac{2\gamma\varepsilon_{\mathsf{opt}}}{1-\gamma}\right)\|V'\|_\infty\mathbf{1}}$$

$$\overset{\text{(ii)}}{\leq} \sqrt{\frac{\|V'\|_\infty^2}{\gamma(1-\gamma)}} 1 + \sqrt{\frac{\left(2 + 20\sqrt{\frac{\log(\frac{54SAN^2}{(1-\gamma)\delta})}{(1-\gamma)^2 N}} + \frac{2\gamma\varepsilon_{\mathsf{opt}}}{1-\gamma}\right) \|V'\|_\infty}{(1-\gamma)^2\gamma^2}} 1$$

$$\overset{\text{(iii)}}{\leq} \sqrt{\frac{\|V'\|_\infty^2}{\gamma(1-\gamma)}} 1 + \sqrt{\frac{24\|V'\|_\infty}{(1-\gamma)^2\gamma^2}} 1 \leq 6\sqrt{\frac{\|V'\|_\infty}{(1-\gamma)^2\gamma^2}} 1,$$

$$(122)$$

where (i) arises from following the routine of (101), (ii) holds by repeating the argument of (102), (iii) follows by taking $N \geq \frac{\log(\frac{54SAN^2}{(1-\gamma)\delta})}{(1-\gamma)^2}$ and $\varepsilon_{\mathsf{opt}} \leq \frac{1-\gamma}{\gamma}$, and the last inequality holds by $\|V'\|_\infty \leq \|V^{\star,\sigma}\|_\infty \leq \frac{1}{1-\gamma}$.

Finally, applying Lemma 7 with $P = \widehat{P}^0$ and $\pi = \widehat{\pi}$ yields

$$\|V'\|_\infty \leq \max_{s \in \mathcal{S}} \widehat{V}^{\widehat{\pi},\sigma}(s) - \min_{s \in \mathcal{S}} \widehat{V}^{\widehat{\pi},\sigma}(s) \leq \frac{1}{\gamma \max\{1-\gamma,\sigma\}},$$

which can be inserted into (122) and gives

$$\left(I - \gamma \underline{P}^{\widehat{\pi},\widehat{V}}\right)^{-1} \sqrt{\mathrm{Var}_{\underline{P}^{\widehat{\pi},\widehat{v}}}(\widehat{V}^{\widehat{\pi},\sigma})} \leq 6\sqrt{\frac{1}{\gamma^3(1-\gamma)^2 \max\{1-\gamma,\sigma\}}} 1 \leq 6\sqrt{\frac{1}{(1-\gamma)^3\gamma^2}} 1.$$

# D  Proof of the lower bound with TV distance: Theorem 2

To prove Theorem 2, we shall first construct some hard instances and then characterize the sample complexity requirements over these instances. Note that the hard instances for robust MDPs are different from those for standard MDPs, due to the asymmetric structure induced by the robust RL problem formulation to consider the worst-case performance. By constructing a new class of hard instances inspired by the asymmetric structure of the RMDP, we develop a new lower bound in Theorem 2 that is tighter than prior art (Yang et al., 2022).

## D.1  Construction of the hard problem instances

**Construction of two hard MDPs.**  Suppose there are two standard MDPs defined as below:

$$\left\{\mathcal{M}_\phi = \left(\mathcal{S}, \mathcal{A}, P^\phi, r, \gamma\right) \mid \phi = \{0,1\}\right\}.$$

Here, $\gamma$ is the discount parameter, $\mathcal{S} = \{0, 1, \ldots, S-1\}$ is the state space. Given any state $s \in \{2, 3, \cdots, S-1\}$, the corresponding action space are $\mathcal{A} = \{0, 1, 2, \cdots, A-1\}$. While for states $s = 0$ or $s = 1$, the action space is only $\mathcal{A}' = \{0, 1\}$. For any $\phi \in \{0, 1\}$, the transition kernel $P^\phi$ of the constructed MDP $\mathcal{M}_\phi$ is defined as

$$P^\phi(s' \mid s, a) = \begin{cases} p\mathbb{1}(s'=1) + (1-p)\mathbb{1}(s'=0) & \text{if} \quad (s,a) = (0, \phi) \\ q\mathbb{1}(s'=1) + (1-q)\mathbb{1}(s'=0) & \text{if} \quad (s,a) = (0, 1-\phi) \\ \mathbb{1}(s'=1) & \text{if} \quad s \geq 1 \end{cases}, \quad (123)$$

where $p$ and $q$ are set to satisfy

$$0 \leq p \leq 1 \quad \text{and} \quad 0 \leq q = p - \Delta \quad (124)$$

for some $p$ and $\Delta > 0$ that shall be introduced later. The above transition kernel $P^\phi$ implies that state 1 is an absorbing state, namely, the MDP will always stay after it arrives at 1.

Then, we define the reward function as

$$r(s, a) = \begin{cases} 1 & \text{if } s = 1 \\ 0 & \text{otherwise} \end{cases}. \quad (125)$$

Additionally, we choose the following initial state distribution:

$$\varphi(s) = \begin{cases} 1, & \text{if } s = 0 \\ 0, & \text{otherwise} \end{cases}. \quad (126)$$

Here, the constructed two instances are set with different probability transition from state 0 with reward 0 but not state 1 with reward 1 (which were used in standard MDPs (Li et al., 2022a)), yielding a larger gap between the value functions of the two instances.

**Uncertainty set of the transition kernels.** Recalling the uncertainty set assumed throughout this section is defined as $\mathcal{U}^\sigma(P^\phi)$ with TV distance:

$$\mathcal{U}^\sigma(P) \coloneqq \mathcal{U}^\sigma_{\mathsf{TV}}(P) = \otimes\, \mathcal{U}^\sigma_{\mathsf{TV}}(P_{s,a}), \qquad \mathcal{U}^\sigma_{\mathsf{TV}}(P_{s,a}) \coloneqq \left\{ P'_{s,a} \in \Delta(\mathcal{S}) : \frac{1}{2} \left\| P'_{s,a} - P_{s,a} \right\|_1 \le \sigma \right\}, \tag{127}$$

where $P^\phi_{s,a} \coloneqq P^\phi(\cdot \,|\, s, a)$ is defined similar to (4). In addition, without loss of generality, we recall the radius $\sigma \in (0, 1 - c_0]$ with $0 < c_0 < 1$. With the uncertainty level in hand, taking $c_1 \coloneqq \frac{c_0}{2}$, $p$ and $\Delta$ which determines the instances obey

$$p = (1 + c_1) \max\{1 - \gamma, \sigma\} \qquad \text{and} \qquad \Delta \le c_1 \max\{1 - \gamma, \sigma\}, \tag{128}$$

which ensure $0 \le p \le 1$ as follows:

$$(1 + c_1)\,\sigma \le 1 - c_0 + c_1 \sigma \le 1 - \frac{c_0}{2} < 1, \qquad (1 + c_1)\,(1 - \gamma) \le \frac{3}{2}(1 - \gamma) \le \frac{3}{4} < 1. \tag{129}$$

Consequently, applying (124) directly leads to

$$p \ge q \ge \max\{1 - \gamma, \sigma\}. \tag{130}$$

To continue, for any $(s, a, s') \in \mathcal{S} \times \mathcal{A} \times \mathcal{S}$, we denote the infimum probability of moving to the next state $s'$ associated with any perturbed transition kernel $P_{s,a} \in \mathcal{U}^\sigma(P^\phi_{s,a})$ as

$$\underline{P}^\phi(s' \,|\, s, a) \coloneqq \inf_{P_{s,a} \in \mathcal{U}^\sigma(P^\phi_{s,a})} P(s' \,|\, s, a) = \max\{P(s' \,|\, s, a) - \sigma, 0\}, \tag{131}$$

where the last equation can be easily verified by the definition of $\mathcal{U}^\sigma(P^\phi)$ in (127). As shall be seen, the transition from state 0 to state 1 plays an important role in the analysis, for convenience, we denote

$$\underline{p} \coloneqq \underline{P}^\phi(1 \,|\, 0, \phi) = p - \sigma, \qquad \underline{q} \coloneqq \underline{P}^\phi(1 \,|\, 0, 1 - \phi) = q - \sigma, \tag{132}$$

which follows from the fact that $p \ge q \ge \sigma$ in (130).

**Robust value functions and robust optimal policies.** To proceed, we are ready to derive the corresponding robust value functions, identify the optimal policies, and characterize the optimal values. For any MDP $\mathcal{M}_\phi$ with the above uncertainty set, we denote $\pi^\star_\phi$ as the optimal policy, and the robust value function of any policy $\pi$ (resp. the optimal policy $\pi^\star_\phi$) as $V^{\pi,\sigma}_\phi$ (resp. $V^{\star,\sigma}_\phi$). Then, we introduce the following lemma which describes some important properties of the robust (optimal) value functions and optimal policies. The proof is postponed to Appendix D.3.1.

**Lemma 14.** *For any $\phi = \{0, 1\}$ and any policy $\pi$, the robust value function obeys*

$$V^{\pi,\sigma}_\phi(0) = \frac{\gamma\left(z^\pi_\phi - \sigma\right)}{(1 - \gamma)\left(1 + \frac{\gamma(z^\pi_\phi - \sigma)}{1 - \gamma(1 - \sigma)}\right)(1 - \gamma(1 - \sigma))}, \tag{133}$$

*where $z^\pi_\phi$ is defined as*

$$z^\pi_\phi \coloneqq p\pi(\phi \,|\, 0) + q\pi(1 - \phi \,|\, 0). \tag{134}$$

*In addition, the robust optimal value functions and the robust optimal policies satisfy*

$$V^{\star,\sigma}_\phi(0) = \frac{\gamma\,(p - \sigma)}{(1 - \gamma)\left(1 + \frac{\gamma(p - \sigma)}{1 - \gamma(1 - \sigma)}\right)(1 - \gamma(1 - \sigma))}, \tag{135a}$$

$$\pi^\star_\phi(\phi \,|\, s) = 1, \qquad \text{for } s \in \mathcal{S}. \tag{135b}$$

## D.2 Establishing the minimax lower bound

Note that our goal is to control the quantity w.r.t. any policy estimator $\widehat{\pi}$ based on the chosen initial distribution $\varphi$ in (126) and the dataset consisting of $N$ samples over each state-action pair generated from the nominal transition kernel $P^\phi$, which gives

$$\left\langle \varphi, V^{\star,\sigma}_\phi - V^{\widehat{\pi},\sigma}_\phi \right\rangle = V^{\star,\sigma}_\phi(0) - V^{\widehat{\pi},\sigma}_\phi(0).$$

**Step 1: converting the goal to estimate $\phi$.** We make the following useful claim which shall be verified in Appendix D.3.2: With $\varepsilon \leq \frac{c_1}{32(1-\gamma)}$, letting

$$\Delta = 32(1-\gamma)\max\{1-\gamma,\sigma\}\varepsilon \leq c_1 \max\{1-\gamma,\sigma\} \tag{136}$$

which satisfies (128), it leads to that for any policy $\widehat{\pi}$,

$$\langle \varphi, V_\phi^{\star,\sigma} - V_\phi^{\widehat{\pi},\sigma} \rangle \geq 2\varepsilon\big(1 - \widehat{\pi}(\phi \,|\, 0)\big). \tag{137}$$

With this connection established between the policy $\widehat{\pi}$ and its sub-optimality gap as depicted in (137), we can now proceed to build an estimate for $\phi$. Here, we denote $\mathbb{P}_\phi$ as the probability distribution when the MDP is $\mathcal{M}_\phi$, where $\phi$ can take on values in the set $\{0,1\}$.

Let's assume momentarily that an estimated policy $\widehat{\pi}$ achieves

$$\mathbb{P}_\phi \left\{ \langle \varphi, V_\phi^{\star,\sigma} - V_\phi^{\widehat{\pi},\sigma} \rangle \leq \varepsilon \right\} \geq \frac{7}{8}, \tag{138}$$

then in view of (137), we necessarily have $\widehat{\pi}(\phi \,|\, 0) \geq \frac{1}{2}$ with probability at least $\frac{7}{8}$. With this in mind, we are motivated to construct the following estimate $\widehat{\phi}$ for $\phi \in \{0,1\}$:

$$\widehat{\phi} = \arg \max_{a \in \{0,1\}} \widehat{\pi}(a \,|\, 0), \tag{139}$$

which obeys

$$\mathbb{P}_\phi \big\{ \widehat{\phi} = \phi \big\} \geq \mathbb{P}_\phi \big\{ \widehat{\pi}(\phi \,|\, 0) > 1/2 \big\} \geq \frac{7}{8}. \tag{140}$$

Subsequently, our aim is to demonstrate that (140) cannot occur without an adequate number of samples, which would in turn contradict (137).

**Step 2: probability of error in testing two hypotheses.** Equipped with the aforementioned groundwork, we can now delve into differentiating between the two hypotheses $\phi \in \{0,1\}$. To achieve this, we consider the concept of minimax probability of error, defined as follows:

$$p_{\mathrm{e}} := \inf_\psi \max \big\{ \mathbb{P}_0(\psi \neq 0), \, \mathbb{P}_1(\psi \neq 1) \big\}. \tag{141}$$

Here, the infimum is taken over all possible tests $\psi$ constructed from the samples generated from the nominal transition kernel $P^\phi$.

Moving forward, let us denote $\mu_\phi$ (resp. $\mu_\phi(s)$) as the distribution of a sample tuple $(s_i, a_i, s_i')$ under the nominal transition kernel $P^\phi$ associated with $\mathcal{M}_\phi$ and the samples are generated independently. Applying standard results from Tsybakov and Zaiats (2009, Theorem 2.2) and the additivity of the KL divergence (cf. Tsybakov and Zaiats (2009, Page 85)), we obtain

$$p_{\mathrm{e}} \geq \frac{1}{4} \exp\Big( - NSA\mathsf{KL}\big(\mu_0 \,\|\, \mu_1\big) \Big)$$
$$= \frac{1}{4} \exp\left\{ - N\Big( \mathsf{KL}\big(P^0(\cdot \,|\, 0, 0) \,\|\, P^1(\cdot \,|\, 0, 0)\big) + \mathsf{KL}\big(P^0(\cdot \,|\, 0, 1) \,\|\, P^1(\cdot \,|\, 0, 1)\big) \Big) \right\}, \tag{142}$$

where the last inequality holds by observing that

$$\mathsf{KL}\big(\mu_0 \,\|\, \mu_1\big) = \frac{1}{SA} \sum_{s,a,s'} \mathsf{KL}\big(P^0(s' \,|\, s, a) \,\|\, P^1(s' \,|\, s, a)\big)$$
$$= \frac{1}{SA} \sum_{a \in \{0,1\}} \mathsf{KL}\big(P^0(\cdot \,|\, 0, a) \,\|\, P^1(\cdot \,|\, 0, a)\big),$$

Here, the last equality holds by the fact that $P^0(\cdot \,|\, s, a)$ and $P^1(\cdot \,|\, s, a)$ only differ when $s = 0$.

Now, our focus shifts towards bounding the terms involving the KL divergence in (142). Given $p \geq q \geq \max\{1-\gamma, \sigma\}$ (cf. (130)), applying Lemma 1 (cf. (23)) gives

$$\mathsf{KL}\big(P^0(\cdot \,|\, 0, 1) \,\|\, P^1(\cdot \,|\, 0, 1)\big) = \mathsf{KL}\big(p \,\|\, q\big) \leq \frac{(p-q)^2}{(1-p)p} \overset{\text{(i)}}{=} \frac{\Delta^2}{p(1-p)}$$

$$\overset{\text{(ii)}}{=} \frac{1024(1-\gamma)^2 \max\{1-\gamma, \sigma\}^2 \varepsilon^2}{p(1-p)}$$

$$\leq \frac{1024(1-\gamma)^2 \max\{1-\gamma, \sigma\}\varepsilon^2}{1-p} \leq \frac{4096}{c_1}(1-\gamma)^2 \max\{1-\gamma, \sigma\}\varepsilon^2, \tag{143}$$

where (i) stems from the definition in (124), (ii) follows by the expression of $\Delta$ in (136), and the last inequality arises from $1 - q \geq 1 - p \geq \frac{c_0}{4}$ (see (129)).

Note that it can be shown that $\mathsf{KL}\big(P^0(\cdot \,|\, 0, 0) \,\|\, P^1(\cdot \,|\, 0, 0)\big)$ can be upper bounded in a same manner. Substituting (143) back into (142) demonstrates that: if the sample size is selected as

$$N \leq \frac{c_1 \log 2}{8192(1-\gamma)^2 \max\{1-\gamma, \sigma\}\varepsilon^2}, \tag{144}$$

then one necessarily has

$$p_{\mathrm{e}} \geq \frac{1}{4} \exp\left\{ -N \frac{8192}{c_1}(1-\gamma)^2 \max\{1-\gamma, \sigma\}\varepsilon^2 \right\} \geq \frac{1}{8}, \tag{145}$$

**Step 3: putting the results together.** Lastly, suppose that there exists an estimator $\widehat{\pi}$ such that

$$\mathbb{P}_0\big\{ \langle \varphi, V_0^{\star,\sigma} - V_0^{\widehat{\pi},\sigma} \rangle > \varepsilon \big\} < \frac{1}{8} \qquad \text{and} \qquad \mathbb{P}_1\big\{ \langle \varphi, V_1^{\star,\sigma} - V_1^{\widehat{\pi},\sigma} \rangle > \varepsilon \big\} < \frac{1}{8}.$$

According to Step 1, the estimator $\widehat{\phi}$ defined in (139) must satisfy

$$\mathbb{P}_0\big(\widehat{\phi} \neq 0\big) < \frac{1}{8} \qquad \text{and} \qquad \mathbb{P}_1\big(\widehat{\phi} \neq 1\big) < \frac{1}{8}.$$

However, this cannot occur under the sample size condition (144) to avoid contradiction with (145). Thus, we have completed the proof.

### D.3 Proof of the auxiliary facts

#### D.3.1 Proof of Lemma 14

**Deriving the robust value function over different states.** For any $\mathcal{M}_\phi$ with $\phi \in \{0, 1\}$, we first characterize the robust value function of any policy $\pi$ over different states. Before proceeding, we denote the minimum of the robust value function over states as below:

$$V_{\phi,\min}^{\pi,\sigma} := \min_{s \in \mathcal{S}} V_\phi^{\pi,\sigma}(s). \tag{146}$$

Clearly, there exists at least one state $s_{\phi,\min}^\pi$ that satisfies $V_\phi^{\pi,\sigma}(s_{\phi,\min}^\pi) = V_{\phi,\min}^{\pi,\sigma}$.

With this in mind, it is easily observed that for any policy $\pi$, the robust value function at state $s = 1$ obeys

$$V_\phi^{\pi,\sigma}(1) = \mathbb{E}_{a \sim \pi(\cdot \,|\, 1)} \left[ r(1, a) + \gamma \inf_{\mathcal{P} \in \mathcal{U}^\sigma(P_{1,a}^\phi)} \mathcal{P} V_\phi^{\pi,\sigma} \right]$$

$$\overset{\text{(i)}}{=} 1 + \gamma \mathbb{E}_{a \sim \pi(\cdot \,|\, 1)} \left[ \underline{P}^\phi(1 \,|\, 1, a) V_\phi^{\pi,\sigma}(1) \right] + \gamma \sigma V_{\phi,\min}^{\pi,\sigma} \overset{\text{(ii)}}{=} 1 + \gamma(1-\sigma) V_\phi^{\pi,\sigma}(1) + \gamma \sigma V_{\phi,\min}^{\pi,\sigma}, \tag{147}$$

where (i) holds by $r(1, a) = 1$ for all $a \in \mathcal{A}'$ and (131), and (ii) follows from $P^\phi(1 \,|\, 1, a) = 1$ for all $a \in \mathcal{A}'$.

Similarly, for any $s \in \{2, 3, \cdots, S-1\}$, we have

$$V_\phi^{\pi,\sigma}(s) = 0 + \gamma \mathbb{E}_{a \sim \pi(\cdot \,|\, s)} \left[ \underline{P}^\phi(1 \,|\, s, a) V_\phi^{\pi,\sigma}(1) \right] + \gamma \sigma V_{\phi,\min}^{\pi,\sigma}$$

$$= \gamma(1-\sigma) V_\phi^{\pi,\sigma}(1) + \gamma \sigma V_{\phi,\min}^{\pi,\sigma}, \tag{148}$$

since $r(s, a) = 0$ for all $s \in \{2, 3, \cdots, S-1\}$ and the definition in (131).

Finally, we move onto compute $V_\phi^{\pi,\sigma}(0)$, the robust value function at state 0 associated with any policy $\pi$. First, it obeys

$$V_\phi^{\pi,\sigma}(0) = \mathbb{E}_{a \sim \pi(\cdot \mid 0)}\left[r(0,a) + \gamma \inf_{\mathcal{P} \in \mathcal{U}^\sigma(P_{0,a}^\phi)} \mathcal{P}V_\phi^{\pi,\sigma}\right]$$
$$= 0 + \gamma\pi(\phi \mid 0) \inf_{\mathcal{P} \in \mathcal{U}^\sigma(P_{0,\phi}^\phi)} \mathcal{P}V_\phi^{\pi,\sigma} + \gamma\pi(1-\phi \mid 0) \inf_{\mathcal{P} \in \mathcal{U}^\sigma(P_{0,1-\phi}^\phi)} \mathcal{P}V_\phi^{\pi,\sigma}. \tag{149}$$

Recall the transition kernel defined in (123) and the fact about the uncertainty set over state 0 in (132), it is easily verified that the following probability vector $P_1 \in \Delta(\mathcal{S})$ obeys $P_1 \in \mathcal{U}^\sigma(P_{0,\phi}^\phi)$, which is defined as

$$P_1(0) = 1 - p + \sigma \mathbb{1}\left(0 = s_{\phi,\min}^\pi\right), \qquad P_1(1) = \underline{p} = p - \sigma,$$
$$P_1(s) = \sigma \mathbb{1}\left(s = s_{\phi,\min}^\pi\right), \qquad \forall s \in \{2, 3, \cdots, S-1\}, \tag{150}$$

where $\underline{p} = p - \sigma$ due to (132). Similarly, the following probability vector $P_2 \in \Delta(\mathcal{S})$ also falls into the uncertainty set $\mathcal{U}^\sigma(P_{0,1-\phi}^\phi)$:

$$P_2(0) = 1 - q + \sigma \mathbb{1}\left(0 = s_{\phi,\min}^\pi\right), \qquad P_2(1) = \underline{q} = q - \sigma,$$
$$P_2(s) = \sigma \mathbb{1}\left(0 = s_{\phi,\min}^\pi\right) \qquad \forall s \in \{2, 3, \cdots, S-1\}. \tag{151}$$

It is noticed that $P_0$ and $P_1$ defined above are the worst-case perturbations, since the probability mass at state 1 will be moved to the state with the least value. Plugging the above facts about $P_1 \in \mathcal{U}^\sigma(P_{0,\phi}^\phi)$ and $P_2 \in \mathcal{U}^\sigma(P_{0,1-\phi}^\phi)$ into (149), we arrive at

$$V_\phi^{\pi,\sigma}(0) \leq \gamma\pi(\phi \mid 0)P_1 V_\phi^{\pi,\sigma} + \gamma\pi(1-\phi \mid 0)P_2 V_\phi^{\pi,\sigma}$$
$$= \gamma\pi(\phi \mid 0)\Big[(p-\sigma) V_\phi^{\pi,\sigma}(1) + (1-p) V_\phi^{\pi,\sigma}(0) + \sigma V_{\phi,\min}^{\pi,\sigma}\Big]$$
$$+ \gamma\pi(1-\phi \mid 0)\Big[(q-\sigma) V_\phi^{\pi,\sigma}(1) + (1-q) V_\phi^{\pi,\sigma}(0) + \sigma V_{\phi,\min}^{\pi,\sigma}\Big]$$
$$\overset{(i)}{=} \gamma\left(z_\phi^\pi - \sigma\right) V_\phi^{\pi,\sigma}(1) + \gamma\sigma V_{\phi,\min}^{\pi,\sigma} + \gamma(1 - z_\phi^\pi)V_\phi^{\pi,\sigma}(0), \tag{152}$$

where the last equality holds by the definition of $z_\phi^\pi$ in (134). To continue, recursively applying (152) yields

$$V_\phi^{\pi,\sigma}(0)$$
$$\leq \gamma\left(z_\phi^\pi - \sigma\right) V_\phi^{\pi,\sigma}(1) + \gamma\sigma V_{\phi,\min}^{\pi,\sigma} + \gamma(1 - z_\phi^\pi)\Big[\gamma\left(z_\phi^\pi - \sigma\right) V_\phi^{\pi,\sigma}(1)$$
$$+ \gamma\sigma V_{\phi,\min}^{\pi,\sigma} + \gamma(1-z_\phi^\pi)V_\phi^{\pi,\sigma}(0)\Big]$$
$$\overset{(i)}{\leq} \gamma\left(z_\phi^\pi - \sigma\right) V_\phi^{\pi,\sigma}(1) + \gamma\sigma V_{\phi,\min}^{\pi,\sigma} + \gamma(1 - z_\phi^\pi)\Big[\gamma z_\phi^\pi V_\phi^{\pi,\sigma}(1) + \gamma(1-z_\phi^\pi)V_\phi^{\pi,\sigma}(0)\Big]$$
$$\leq \dots$$
$$\leq \gamma\left(z_\phi^\pi - \sigma\right) V_\phi^{\pi,\sigma}(1) + \gamma\sigma V_{\phi,\min}^{\pi,\sigma} + \gamma z_\phi^\pi \sum_{t=1}^\infty \gamma^t(1-z_\phi^\pi)^t V_\phi^{\pi,\sigma}(1) + \lim_{t \to \infty} \gamma^t(1-z_\phi^\pi)^t V_\phi^{\pi,\sigma}(0)$$
$$\overset{(ii)}{\leq} \gamma\left(z_\phi^\pi - \sigma\right) V_\phi^{\pi,\sigma}(1) + \gamma\sigma V_{\phi,\min}^{\pi,\sigma} + \gamma(1-z_\phi^\pi)\frac{\gamma z_\phi^\pi}{1 - \gamma(1-z_\phi^\pi)}V_\phi^{\pi,\sigma}(1) + 0$$
$$< \gamma\left(z_\phi^\pi - \sigma\right) V_\phi^{\pi,\sigma}(1) + \gamma\sigma V_{\phi,\min}^{\pi,\sigma} + \gamma(1-z_\phi^\pi)V_\phi^{\pi,\sigma}(1)$$
$$= \gamma\left(1 - \sigma\right) V_\phi^{\pi,\sigma}(1) + \gamma\sigma V_{\phi,\min}^{\pi,\sigma}, \tag{153}$$

where (i) uses $V_{\phi,\min}^{\pi,\sigma} \leq V_\phi^{\pi,\sigma}(1)$, (ii) follows from $\gamma(1 - z_\phi^\pi) < 1$, and the penultimate line follows from the trivial fact that $\frac{\gamma z_\phi^\pi}{1-\gamma(1-z_\phi^\pi)} < 1$.

Combining (147), (148), and (153), we have that for any policy $\pi$,

$$V_\phi^{\pi,\sigma}(0) = V_{\phi,\min}^{\pi,\sigma}, \tag{154}$$

which directly leads to

$$V_\phi^{\pi,\sigma}(1) = 1 + \gamma\left(1 - \sigma\right) V_\phi^{\pi,\sigma}(1) + \gamma\sigma V_{\phi,\min}^{\pi,\sigma} = \frac{1 + \gamma\sigma V_\phi^{\pi,\sigma}(0)}{1 - \gamma\left(1 - \sigma\right)}. \tag{155}$$

Let's now return to the characterization of $V_\phi^{\pi,\sigma}(0)$. In view of (154), the equality in (152) holds, and we have

$$\begin{aligned}
V_\phi^{\pi,\sigma}(0) &= \gamma\left(z_\phi^\pi - \sigma\right) V_\phi^{\pi,\sigma}(1) + \gamma\left(1 - z_\phi^\pi + \sigma\right) V_\phi^{\pi,\sigma}(0) \\
&\stackrel{(i)}{=} \gamma\left(z_\phi^\pi - \sigma\right) \frac{1 + \gamma\sigma V_\phi^{\pi,\sigma}(0)}{1 - \gamma\left(1 - \sigma\right)} + \gamma\left(1 - z_\phi^\pi + \sigma\right) V_\phi^{\pi,\sigma}(0) \\
&= \frac{\gamma\left(z_\phi^\pi - \sigma\right)}{1 - \gamma\left(1 - \sigma\right)} + \gamma\left(1 + \left(z_\phi^\pi - \sigma\right) \frac{\gamma\sigma - (1 - \gamma(1 - \sigma))}{1 - \gamma\left(1 - \sigma\right)}\right) V_\phi^{\pi,\sigma}(0) \\
&= \frac{\gamma\left(z_\phi^\pi - \sigma\right)}{1 - \gamma\left(1 - \sigma\right)} + \gamma\left(1 - \frac{(1 - \gamma)\left(z_\phi^\pi - \sigma\right)}{1 - \gamma\left(1 - \sigma\right)}\right) V_\phi^{\pi,\sigma}(0),
\end{aligned}$$

where (i) arises from (155). Solving this relation gives

$$V_\phi^{\pi,\sigma}(0) = \frac{\frac{\gamma\left(z_\phi^\pi - \sigma\right)}{1 - \gamma(1 - \sigma)}}{(1 - \gamma)\left(1 + \frac{\gamma\left(z_\phi^\pi - \sigma\right)}{1 - \gamma(1 - \sigma)}\right)}. \tag{156}$$

**The optimal robust policy and optimal robust value function.** We move on to characterize the robust optimal policy and its corresponding robust value function. To begin with, denoting

$$z := \frac{\gamma\left(z_\phi^\pi - \sigma\right)}{1 - \gamma\left(1 - \sigma\right)}, \tag{157}$$

we rewrite (156) as

$$V_\phi^{\pi,\sigma}(0) = \frac{z}{(1 - \gamma)(1 + z)} =: f(z).$$

Plugging in the fact that $z_\phi^\pi \geq q \geq \sigma > 0$ in (130), it follows that $z > 0$. So for any $z > 0$, the derivative of $f(z)$ w.r.t. $z$ obeys

$$\frac{(1 - \gamma)(1 + z) - (1 - \gamma)z}{(1 - \gamma)^2(1 + z)^2} = \frac{1}{(1 - \gamma)(1 + z)^2} > 0. \tag{158}$$

Observing that $f(z)$ is increasing in $z$, $z$ is increasing in $z_\phi^\pi$, and $z_\phi^\pi$ is also increasing in $\pi(\phi\,|\,0)$ (see the fact $p \geq q$ in (130)), the optimal policy in state 0 thus obeys

$$\pi_\phi^\star(\phi\,|\,0) = 1. \tag{159}$$

Considering that the action does not influence the state transition for all states $s > 0$, without loss of generality, we choose the robust optimal policy to obey

$$\forall s > 0: \quad \pi_\phi^\star(\phi\,|\,s) = 1. \tag{160}$$

Taking $\pi = \pi_\phi^\star$, we complete the proof by showing that the corresponding robust optimal robust value function at state 0 as follows:

$$V_\phi^{\star,\sigma}(0) = \frac{\frac{\gamma\left(z_\phi^{\pi^\star} - \sigma\right)}{1 - \gamma(1 - \sigma)}}{(1 - \gamma)\left(1 + \frac{\gamma\left(z_\phi^{\pi^\star} - \sigma\right)}{1 - \gamma(1 - \sigma)}\right)} = \frac{\frac{\gamma(p - \sigma)}{1 - \gamma(1 - \sigma)}}{(1 - \gamma)\left(1 + \frac{\gamma(p - \sigma)}{1 - \gamma(1 - \sigma)}\right)}. \tag{161}$$

 **D.3.2  Proof of the claim** (137)

Plugging in the definition of $\varphi$, we arrive at that for any policy $\pi$,

$$\left\langle \varphi, V_\phi^{\star,\sigma} - V_\phi^{\pi,\sigma} \right\rangle = V_\phi^{\star,\sigma}(0) - V_\phi^{\pi,\sigma}(0) = \frac{\frac{\gamma\left(p - z_\phi^\pi\right)}{1 - \gamma(1-\sigma)}}{(1-\gamma)\left(1 + \frac{\gamma(p-\sigma)}{1-\gamma(1-\sigma)}\right)\left(1 + \frac{\gamma\left(z_\phi^\pi - \sigma\right)}{1-\gamma(1-\sigma)}\right)}, \quad (162)$$

which follows from applying (133) and basic calculus. Then, we proceed to control the above term in two cases separately in terms of the uncertainty level $\sigma$.

- When $\sigma \in (0, 1-\gamma]$. Then regarding the important terms in (162), we observe that

$$1 - \gamma < 1 - \gamma(1-\sigma) \leq 1 - \gamma(1 - (1-\gamma)) = (1-\gamma)(1+\gamma) \leq 2(1-\gamma), \quad (163)$$

  which directly leads to

$$\frac{\gamma\left(z_\phi^\pi - \sigma\right)}{1 - \gamma(1-\sigma)} \overset{(i)}{\leq} \frac{\gamma(p-\sigma)}{1-\gamma(1-\sigma)} \leq \frac{\gamma c_1(1-\gamma)}{1-\gamma(1-\sigma)} \overset{(ii)}{<} c_1\gamma, \quad (164)$$

  where (i) holds by $z_\phi^\pi < p$, and (ii) is due to (163). Inserting (163) and (164) back into (162), we arrive at

$$\left\langle \varphi, V_\phi^{\star,\sigma} - V_\phi^{\pi,\sigma} \right\rangle \geq \frac{\frac{\gamma\left(p - z_\phi^\pi\right)}{2(1-\gamma)}}{(1-\gamma)(1 + c_1\gamma)^2} \geq \frac{\gamma\left(p - z_\phi^\pi\right)}{8(1-\gamma)^2}$$
$$= \frac{\gamma(p-q)\left(1 - \pi(\phi \,|\, 0)\right)}{8(1-\gamma)^2} = \frac{\gamma\Delta\left(1 - \pi(\phi \,|\, 0)\right)}{8(1-\gamma)^2} \geq 2\varepsilon\left(1 - \pi(\phi \,|\, 0)\right), \quad (165)$$

  where the last inequality holds by setting ($\gamma \geq 1/2$)

$$\Delta = 32(1-\gamma)^2\varepsilon. \quad (166)$$

  Finally, it is easily verified that

$$\varepsilon \leq \frac{c_1}{32(1-\gamma)} \quad \Longrightarrow \quad \Delta \leq c_1(1-\gamma).$$

- When $\sigma \in (1-\gamma, 1-c_1]$. Regarding (162), we observe that

$$\gamma\sigma < 1 - \gamma(1-\sigma) = 1 - \gamma + \gamma\sigma \leq (1+\gamma)\sigma \leq 2\sigma, \quad (167)$$

  which directly leads to

$$\frac{\gamma\left(z_\phi^\pi - \sigma\right)}{1 - \gamma(1-\sigma)} \leq \frac{\gamma(p-\sigma)}{1-\gamma(1-\sigma)} \leq \frac{\gamma c_1\sigma}{1-\gamma(1-\sigma)} \overset{(i)}{<} c_1, \quad (168)$$

  where (i) holds by (167). Inserting (167) and (168) back into (162), we arrive at

$$\left\langle \varphi, V_\phi^{\star,\sigma} - V_\phi^{\pi,\sigma} \right\rangle \geq \frac{\frac{\gamma\left(p - z_\phi^\pi\right)}{2\sigma}}{(1-\gamma)(1 + c_1)^2} \geq \frac{\gamma\left(p - z_\phi^\pi\right)}{8(1-\gamma)\sigma} = \frac{\gamma(p-q)\left(1 - \pi(\phi \,|\, 0)\right)}{8(1-\gamma)\sigma}$$
$$= \frac{\gamma\Delta\left(1 - \pi(\phi \,|\, 0)\right)}{8(1-\gamma)\sigma} \geq 2\varepsilon\left(1 - \pi(\phi \,|\, 0)\right), \quad (169)$$

  where the last inequality holds by letting ($\gamma \geq 1/2$)

$$\Delta = 32(1-\gamma)\sigma\varepsilon. \quad (170)$$

  Finally, it is easily verified that

$$\varepsilon \leq \frac{c_1}{32(1-\gamma)} \quad \Longrightarrow \quad \Delta \leq c_1\sigma. \quad (171)$$

 # E Proof of the upper bound with $\chi^2$ divergence: Theorem 3

The proof of Theorem 3 mainly follows the structure of the proof of Theorem 1 in Appendix C. Throughout this section, for any nominal transition kernel $P$, the uncertainty set is taken as (see (8))

$$\mathcal{U}^\sigma(P) = \mathcal{U}^\sigma_{\chi^2}(P) := \otimes \, \mathcal{U}^\sigma_{\chi^2}(P_{s,a}),$$

$$\mathcal{U}^\sigma_{\chi^2}(P_{s,a}) := \left\{ P'_{s,a} \in \Delta(\mathcal{S}) : \sum_{s'\in\mathcal{S}} \frac{(P'(s'\,|\,s,a) - P(s'\,|\,s,a))^2}{P(s'\,|\,s,a)} \leq \sigma \right\}. \qquad (172)$$

## E.1 Proof of Theorem 3

In order to control the performance gap $\big\| V^{\star,\sigma} - V^{\widehat{\pi},\sigma} \big\|_\infty$, recall the error decomposition in (49):

$$V^{\star,\sigma} - V^{\widehat{\pi},\sigma} \leq \left( V^{\pi^\star,\sigma} - \widehat{V}^{\pi^\star,\sigma} \right) + \frac{2\gamma\varepsilon_{\mathsf{opt}}}{1-\gamma} 1 + \left( \widehat{V}^{\widehat{\pi},\sigma} - V^{\widehat{\pi},\sigma} \right), \qquad (173)$$

where $\varepsilon_{\mathsf{opt}}$ (cf. (48)) shall be specified later (which justifies Remark **??**). To further control (173), we bound the remaining two terms separately.

**Step 1: controlling** $\big\| \widehat{V}^{\pi^\star,\sigma} - V^{\pi^\star,\sigma} \big\|_\infty$. Towards this, recall the bound in (54) which holds for any uncertainty set:

$$\left\| \widehat{V}^{\pi^\star,\sigma} - V^{\pi^\star,\sigma} \right\|_\infty \leq \gamma \max \left\{ \left\| \left( I - \gamma\underline{\widehat{P}}^{\pi^\star,\widehat{V}} \right)^{-1} \left( \underline{\widehat{P}}^{\pi^\star,V} V^{\pi^\star,\sigma} - \underline{P}^{\pi^\star,V} V^{\pi^\star,\sigma} \right) \right\|_\infty, \right.$$

$$\left. \left\| \left( I - \gamma\underline{\widehat{P}}^{\pi^\star,V} \right)^{-1} \left( \underline{\widehat{P}}^{\pi^\star,V} V^{\pi^\star,\sigma} - \underline{P}^{\pi^\star,V} V^{\pi^\star,\sigma} \right) \right\|_\infty \right\}. \qquad (174)$$

To control the main term $\underline{\widehat{P}}^{\pi^\star,V} V^{\pi^\star,\sigma} - \underline{P}^{\pi^\star,V} V^{\pi^\star,\sigma}$ in (174), we first introduce an important lemma whose proof is postponed to Appendix E.2.1.

**Lemma 15.** *Consider any $\sigma > 0$ and the uncertainty set $\mathcal{U}^\sigma(\cdot) := \mathcal{U}^\sigma_{\chi^2}(\cdot)$. For any $\delta \in (0,1)$ and any fixed policy $\pi$, one has with probability at least $1 - \delta$,*

$$\left\| \underline{\widehat{P}}^{\pi,V} V^{\pi,\sigma} - \underline{P}^{\pi,V} V^{\pi,\sigma} \right\|_\infty \leq 4\sqrt{\frac{2(1+\sigma)\log(\frac{24SAN}{\delta})}{(1-\gamma)^2 N}}.$$

Applying Lemma 15 by taking $\pi = \pi^\star$ gives

$$\left\| \underline{\widehat{P}}^{\pi^\star,V} V^{\pi^\star,\sigma} - \underline{P}^{\pi^\star,V} V^{\pi^\star,\sigma} \right\|_\infty \leq 4\sqrt{\frac{2(1+\sigma)\log(\frac{24SAN}{\delta})}{(1-\gamma)^2 N}}, \qquad (175)$$

which directly leads to

$$\left\| \left( I - \gamma\underline{\widehat{P}}^{\pi^\star,\widehat{V}} \right)^{-1} \left( \underline{\widehat{P}}^{\pi^\star,V} V^{\pi^\star,\sigma} - \underline{P}^{\pi^\star,V} V^{\pi^\star,\sigma} \right) \right\|_\infty$$

$$\leq \left\| \underline{\widehat{P}}^{\pi^\star,V} V^{\pi^\star,\sigma} - \underline{P}^{\pi^\star,V} V^{\pi^\star,\sigma} \right\|_\infty \cdot \left\| \left( I - \gamma\underline{\widehat{P}}^{\pi^\star,\widehat{V}} \right)^{-1} 1 \right\|_\infty \leq 4\sqrt{\frac{2(1+\sigma)\log(\frac{24SAN}{\delta})}{(1-\gamma)^4 N}}. \qquad (176)$$

Similarly, we have

$$\left\| \left( I - \gamma\underline{\widehat{P}}^{\pi^\star,V} \right)^{-1} \left( \underline{\widehat{P}}^{\pi^\star,V} V^{\pi^\star,\sigma} - \underline{P}^{\pi^\star,V} V^{\pi^\star,\sigma} \right) \right\|_\infty \leq 4\sqrt{\frac{2(1+\sigma)\log(\frac{24SAN}{\delta})}{(1-\gamma)^4 N}}. \qquad (177)$$

Inserting (176) and (177) back to (174) yields

$$\left\| \widehat{V}^{\pi^\star,\sigma} - V^{\pi^\star,\sigma} \right\|_\infty \leq 4\sqrt{\frac{2(1+\sigma)\log(\frac{24SAN}{\delta})}{(1-\gamma)^4 N}}. \qquad (178)$$

**Step 2: controlling** $\left\|\widehat{V}^{\widehat{\pi},\sigma} - V^{\widehat{\pi},\sigma}\right\|_\infty$. Recall the bound in (55) which holds for any uncertainty set:

$$\left\|\widehat{V}^{\widehat{\pi},\sigma} - V^{\widehat{\pi},\sigma}\right\|_\infty \leq \gamma \max\left\{ \left\|\left(I - \gamma \underline{P}^{\widehat{\pi},V}\right)^{-1}\left(\underline{\widehat{P}}^{\widehat{\pi},\widehat{V}}\widehat{V}^{\widehat{\pi},\sigma} - \underline{P}^{\widehat{\pi},\widehat{V}}\widehat{V}^{\widehat{\pi},\sigma}\right)\right\|_\infty, \right.$$
$$\left. \left\|\left(I - \gamma \underline{P}^{\widehat{\pi},\widehat{V}}\right)^{-1}\left(\underline{\widehat{P}}^{\widehat{\pi},\widehat{V}}\widehat{V}^{\widehat{\pi},\sigma} - \underline{P}^{\widehat{\pi},\widehat{V}}\widehat{V}^{\widehat{\pi},\sigma}\right)\right\|_\infty\right\}. \quad (179)$$

We introduce the following lemma which controls $\underline{\widehat{P}}^{\widehat{\pi},\widehat{V}}\widehat{V}^{\widehat{\pi},\sigma} - \underline{P}^{\widehat{\pi},\widehat{V}}\widehat{V}^{\widehat{\pi},\sigma}$ in (179); the proof is deferred to Appendix E.2.2.

**Lemma 16.** *Consider the uncertainty set* $\mathcal{U}^\sigma(\cdot) := \mathcal{U}^\sigma_{\chi^2}(\cdot)$ *and any* $\delta \in (0,1)$. *With probability at least* $1 - \delta$, *one has*

$$\left\|\underline{\widehat{P}}^{\widehat{\pi},\widehat{V}}\widehat{V}^{\widehat{\pi},\sigma} - \underline{P}^{\widehat{\pi},\widehat{V}}\widehat{V}^{\widehat{\pi},\sigma}\right\|_\infty \leq 12\sqrt{\frac{2(1+\sigma)\log(\frac{36SAN^2}{\delta})}{(1-\gamma)^2 N}} + \frac{2\gamma\varepsilon_{\mathsf{opt}}}{1-\gamma} + 4\sqrt{\frac{\sigma\varepsilon_{\mathsf{opt}}}{(1-\gamma)^2}}. \quad (180)$$

Repeating the arguments from (175) to (178) yields

$$\left\|\widehat{V}^{\widehat{\pi},\sigma} - V^{\widehat{\pi},\sigma}\right\|_\infty \leq 12\sqrt{\frac{2(1+\sigma)\log(\frac{36SAN^2}{\delta})}{(1-\gamma)^4 N}} + \frac{2\gamma\varepsilon_{\mathsf{opt}}}{(1-\gamma)^2} + 4\sqrt{\frac{\sigma\varepsilon_{\mathsf{opt}}}{(1-\gamma)^4}}. \quad (181)$$

Finally, inserting (178) and (181) back to (173) complete the proof

$$\|V^{\star,\sigma} - V^{\widehat{\pi},\sigma}\|_\infty \leq \left\|V^{\pi^\star,\sigma} - \widehat{V}^{\pi^\star,\sigma}\right\|_\infty + \frac{2\gamma\varepsilon_{\mathsf{opt}}}{1-\gamma} + \left\|\widehat{V}^{\widehat{\pi},\sigma} - V^{\widehat{\pi},\sigma}\right\|_\infty$$
$$\leq 4\sqrt{\frac{2(1+\sigma)\log(\frac{24SAN}{\delta})}{(1-\gamma)^4 N}} + \frac{2\gamma\varepsilon_{\mathsf{opt}}}{1-\gamma} + 12\sqrt{\frac{2(1+\sigma)\log(\frac{36SAN^2}{\delta})}{(1-\gamma)^4 N}}$$
$$+ \frac{2\gamma\varepsilon_{\mathsf{opt}}}{(1-\gamma)^2} + 4\sqrt{\frac{\sigma\varepsilon_{\mathsf{opt}}}{(1-\gamma)^4}}$$
$$\leq 24\sqrt{\frac{2(1+\sigma)\log(\frac{36SAN^2}{\delta})}{(1-\gamma)^4 N}}, \quad (182)$$

where the last line holds by taking $\varepsilon_{\mathsf{opt}} \leq \min\left\{\sqrt{\frac{32(1+\sigma)\log(\frac{36SAN^2}{\delta})}{N}}, \frac{4\log(\frac{36SAN^2}{\delta})}{N}\right\}$.

## E.2 Proof of the auxiliary lemmas

### E.2.1 Proof of Lemma 15

**Step 1: controlling the point-wise concentration.** Consider any fixed policy $\pi$ and the corresponding robust value vector $V := V^{\pi,\sigma}$ (independent from $\widehat{P}^0$). Invoking Lemma 5 leads to that for any $(s,a) \in \mathcal{S} \times \mathcal{A}$,

$$\left|\widehat{P}^{\pi,V}_{s,a}V^{\pi,\sigma} - P^{\pi,V}_{s,a}V^{\pi,\sigma}\right| = \left| \max_{\alpha\in[\min_s V(s),\max_s V(s)]}\left\{P^0_{s,a}[V]_\alpha - \sqrt{\sigma\mathsf{Var}_{P^0_{s,a}}([V]_\alpha)}\right\} \right.$$
$$\left. - \max_{\alpha\in[\min_s V(s),\max_s V(s)]}\left\{\widehat{P}^0_{s,a}[V]_\alpha - \sqrt{\sigma\mathsf{Var}_{\widehat{P}^0_{s,a}}([V]_\alpha)}\right\} \right|$$
$$\leq \max_{\alpha\in[\min_s V(s),\max_s V(s)]}\left|\left(P^0_{s,a} - \widehat{P}^0_{s,a}\right)[V]_\alpha + \sqrt{\sigma\mathsf{Var}_{\widehat{P}^0_{s,a}}([V]_\alpha)} - \sqrt{\sigma\mathsf{Var}_{P^0_{s,a}}([V]_\alpha)}\right|$$
$$\leq \max_{\alpha\in[\min_s V(s),\max_s V(s)]}\left|\left(P^0_{s,a} - \widehat{P}^0_{s,a}\right)[V]_\alpha\right| +$$

$$+ \max_{\alpha \in [\min_s V(s), \max_s V(s)]} \sqrt{\sigma} \left| \sqrt{\mathsf{Var}_{\widehat{P}^0_{s,a}}([V]_\alpha)} - \sqrt{\mathsf{Var}_{P^0_{s,a}}([V]_\alpha)} \right|, \tag{183}$$

where the first inequality follows by that the maximum operator is 1-Lipschitz, and the second inequality follows from the triangle inequality. Observing that the first term in (183) is exactly the same as (89), recalling the fact in (94) directly leads to: with probability at least $1 - \delta$,

$$\max_{\alpha \in [\min_s V(s), \max_s V(s)]} \left| \left( P^0_{s,a} - \widehat{P}^0_{s,a} \right) [V]_\alpha \right| \leq 2 \sqrt{\frac{\log(\frac{2SAN}{\delta})}{(1-\gamma)^2 N}} \tag{184}$$

holds for all $(s, a) \in \mathcal{S} \times \mathcal{A}$. Then the remainder of the proof focuses on controlling the second term in (183).

**Step 2: controlling the second term in** (183). For any given $(s, a) \in \mathcal{S} \times \mathcal{A}$ and fixed $\alpha \in [0, \frac{1}{1-\gamma}]$, applying the concentration inequality (Panaganti and Kalathil, 2022, Lemma 6) with $\|[V]_\alpha\|_\infty \leq \frac{1}{1-\gamma}$, we arrive at

$$\left| \sqrt{\mathsf{Var}_{\widehat{P}^0_{s,a}}([V]_\alpha)} - \sqrt{\mathsf{Var}_{P^0_{s,a}}([V]_\alpha)} \right| \leq \sqrt{\frac{2\log(\frac{2}{\delta})}{(1-\gamma)^2 N}} \tag{185}$$

holds with probability at least $1 - \delta$. To obtain a uniform bound, we first observe the follow lemma proven in Appendix E.2.3.

**Lemma 17.** *For any $V$ obeying $\|V\|_\infty \leq \frac{1}{1-\gamma}$, the function $J_{s,a}(\alpha, V) := \left| \sqrt{\mathsf{Var}_{\widehat{P}^0_{s,a}}([V]_\alpha)} - \sqrt{\mathsf{Var}_{P^0_{s,a}}([V]_\alpha)} \right|$ w.r.t. $\alpha$ obeys*

$$|J_{s,a}(\alpha_1, V) - J_{s,a}(\alpha_2, V)| \leq 4 \sqrt{\frac{|\alpha_1 - \alpha_2|}{1 - \gamma}}.$$

In addition, we can construct an $\varepsilon_3$-net $N_{\varepsilon_3}$ over $[0, \frac{1}{1-\gamma}]$ whose size is $|N_{\varepsilon_3}| \leq \frac{3}{\varepsilon_3(1-\gamma)}$ (Vershynin, 2018). Armed with the above, we can derive the uniform bound over $\alpha \in [\min_s V(s), \max_s V(s)] \subset [0, 1/(1-\gamma)]$: with probability at least $1 - \frac{\delta}{SA}$, it holds that for any $(s, a) \in \mathcal{S} \times \mathcal{A}$,

$$\max_{\alpha \in [\min_s V(s), \max_s V(s)]} \left| \sqrt{\mathsf{Var}_{\widehat{P}^0_{s,a}}([V]_\alpha)} - \sqrt{\mathsf{Var}_{P^0_{s,a}}([V]_\alpha)} \right|$$

$$\leq \max_{\alpha \in [0, 1/(1-\gamma)]} \left| \sqrt{\mathsf{Var}_{\widehat{P}^0_{s,a}}([V]_\alpha)} - \sqrt{\mathsf{Var}_{P^0_{s,a}}([V]_\alpha)} \right|$$

$$\overset{(i)}{\leq} 4 \sqrt{\frac{\varepsilon_3}{1-\gamma}} + \sup_{\alpha \in N_{\varepsilon_3}} \left| \sqrt{\mathsf{Var}_{\widehat{P}^0_{s,a}}([V]_\alpha)} - \sqrt{\mathsf{Var}_{P^0_{s,a}}([V]_\alpha)} \right|$$

$$\overset{(ii)}{\leq} 4 \sqrt{\frac{\varepsilon_3}{1-\gamma}} + \sqrt{\frac{2\log(\frac{2SA|N_{\varepsilon_3}|}{\delta})}{(1-\gamma)^2 N}}$$

$$\overset{(iii)}{\leq} 2 \sqrt{\frac{2\log(\frac{2SA|N_{\varepsilon_3}|}{\delta})}{(1-\gamma)^2 N}} \leq 2 \sqrt{\frac{2\log(\frac{24SAN}{\delta})}{(1-\gamma)^2 N}}, \tag{186}$$

where (i) holds by the property of $N_{\varepsilon_3}$, (ii) follows from (185), (iii) arises from taking $\varepsilon_3 = \frac{\log(\frac{2SA|N_{\varepsilon_3}|}{\delta})}{8N(1-\gamma)}$, and the last inequality is verified by $|N_{\varepsilon_3}| \leq \frac{3}{\varepsilon_3(1-\gamma)} \leq 24N$.

Inserting (184) and (186) back to (183) and taking the union bound over $(s, a) \in \mathcal{S} \times \mathcal{A}$, we arrive at that for all $(s, a) \in \mathcal{S} \times \mathcal{A}$, with probability at least $1 - \delta$,

$$\left| \widehat{P}^{\pi,V}_{s,a} V - P^{\pi,V}_{s,a} V \right| \leq \max_{\alpha \in [\min_s V(s), \max_s V(s)]} \left| \left( P^0_{s,a} - \widehat{P}^0_{s,a} \right) [V]_\alpha \right| +$$

$$+ \max_{\alpha \in [\min_s V(s), \max_s V(s)]} \left| \sqrt{\sigma \mathsf{Var}_{\widehat{P}^0_{s,a}}([V]_\alpha)} - \sqrt{\sigma \mathsf{Var}_{P^0_{s,a}}([V]_\alpha)} \right|$$

$$\leq \sqrt{\frac{2\log(\frac{2SAN}{\delta})}{(1-\gamma)^2 N}} + 2\sqrt{\frac{2\sigma\log(\frac{24SAN}{\delta})}{(1-\gamma)^2 N}} \leq 4\sqrt{\frac{2(1+\sigma)\log(\frac{24SAN}{\delta})}{(1-\gamma)^2 N}}.$$

1124 Finally, we complete the proof by recalling the matrix form as below:

$$\left\| \widehat{\underline{P}}^{\pi,V} V^{\pi,\sigma} - \underline{P}^{\pi,V} V^{\pi,\sigma} \right\|_\infty \leq \max_{(s,a)\in\mathcal{S}\times\mathcal{A}} \left| \widehat{P}_{s,a}^{\pi,V} V - P_{s,a}^{\pi,V} V \right| \leq 4\sqrt{\frac{2(1+\sigma)\log(\frac{24SAN}{\delta})}{(1-\gamma)^2 N}}.$$

### E.2.2 Proof of Lemma 16

1126 **Step 1: decomposing the term of interest.** The proof follows the routine of the proof of Lemma 12
1127 in Appendix C.3.5. To begin with, for any $(s,a) \in \mathcal{S}\times\mathcal{A}$, following the same arguments of (183)
1128 yields

$$\left| \widehat{P}_{s,a}^{\widehat{\pi},\widehat{V}} \widehat{V}^{\widehat{\pi},\sigma} - P_{s,a}^{\widehat{\pi},\widehat{V}} \widehat{V}^{\widehat{\pi},\sigma} \right| \leq \max_{\alpha\in\left[\min_s \widehat{V}^{\widehat{\pi},\sigma}(s),\max_s \widehat{V}^{\widehat{\pi},\sigma}(s)\right]} \left| \left( P_{s,a}^0 - \widehat{P}_{s,a}^0 \right) \left[ \widehat{V}^{\widehat{\pi},\sigma} \right]_\alpha \right| +$$

$$+ \max_{\alpha\in\left[\min_s \widehat{V}^{\widehat{\pi},\sigma}(s),\max_s \widehat{V}^{\widehat{\pi},\sigma}(s)\right]} \sqrt{\sigma} \left| \sqrt{\mathsf{Var}_{\widehat{P}_{s,a}^0}\left(\left[\widehat{V}^{\widehat{\pi},\sigma}\right]_\alpha\right)} - \sqrt{\mathsf{Var}_{P_{s,a}^0}\left(\left[\widehat{V}^{\widehat{\pi},\sigma}\right]_\alpha\right)} \right|. \quad (187)$$

1129 Invoking the fact in (118) (for proving Lemma 12), the first term in (187) obeys

$$\max_{\alpha\in\left[\min_s \widehat{V}^{\widehat{\pi},\sigma}(s),\max_s \widehat{V}^{\widehat{\pi},\sigma}(s)\right]} \left| \left( P_{s,a}^0 - \widehat{P}_{s,a}^0 \right) \left[ \widehat{V}^{\widehat{\pi},\sigma} \right]_\alpha \right| \leq \max_{\alpha\in[0,1/(1-\gamma)]} \left| \left( P_{s,a}^0 - \widehat{P}_{s,a}^0 \right) \left[ \widehat{V}^{\widehat{\pi},\sigma} \right]_\alpha \right|$$

$$\leq 4\sqrt{\frac{\log(\frac{3SAN^{3/2}}{(1-\gamma)\delta})}{(1-\gamma)^2 N}} + \frac{2\gamma\varepsilon_{\mathsf{opt}}}{1-\gamma}. \quad (188)$$

1130 The remainder of the proof will focus on controlling the second term of (187).

1131 **Step 2: controlling the second term of (187).** Towards this, we recall the auxiliary robust MDP
1132 $\widetilde{\mathcal{M}}_{\mathsf{rob}}^{s,u}$ defined in Appendix C.3.5. Taking the uncertainty set $\mathcal{U}^\sigma(\cdot) := \mathcal{U}_{\chi^2}^\sigma(\cdot)$ for both $\widehat{\mathcal{M}}_{\mathsf{rob}}^{s,u}$ and
1133 $\widehat{\mathcal{M}}_{\mathsf{rob}}$, we recall the corresponding robust Bellman operator $\widehat{\mathcal{T}}_{s,u}^\sigma(\cdot)$ in (107) and the following
1134 definition in (108)

$$u^\star := \widehat{V}^{\star,\sigma}(s) - \gamma \inf_{\mathcal{P}\in\mathcal{U}^\sigma(e_s)} \mathcal{P}\widehat{V}^{\star,\sigma}. \quad (189)$$

1135 Following the arguments in Appendix C.3.5, it can be verified that there exists a unique fixed point
1136 $\widehat{Q}_{s,u}^{\star,\sigma}$ of the operator $\widehat{\mathcal{T}}_{s,u}^\sigma(\cdot)$, which satisfies $0 \leq \widehat{Q}_{s,u}^{\star,\sigma} \leq \frac{1}{1-\gamma}\mathbf{1}$. In addition, the corresponding robust
1137 value function coincides with that of the operator $\widehat{\mathcal{T}}^\sigma(\cdot)$, i.e., $\widehat{V}_{s,u}^{\star,\sigma} = \widehat{V}^{\star,\sigma}$.

1138 We recall the $N_{\varepsilon_2}$-net over $\left[0,\frac{1}{1-\gamma}\right]$ whose size obeying $|N_{\varepsilon_2}| \leq \frac{3}{\varepsilon_2(1-\gamma)}$ (Vershynin, 2018). Then
1139 for all $u \in N_{\varepsilon_2}$ and a fixed $\alpha$, $\widehat{\mathcal{M}}_{\mathsf{rob}}^{s,u}$ is statistically independent from $\widehat{P}_{s,a}^0$, which indicates the
1140 independence between $[\widehat{V}_{s,u}^{\star,\sigma}]_\alpha$ and $\widehat{P}_{s,a}^0$. With this in mind, invoking the fact in (186) and taking the
1141 union bound over all $(s,a) \in \mathcal{S}\times\mathcal{A}$ and $u \in N_{\varepsilon_2}$ yields that, with probability at least $1 - \delta$,

$$\max_{\alpha\in[0,1/(1-\gamma)]} \left| \sqrt{\mathsf{Var}_{\widehat{P}_{s,a}^0}\left([\widehat{V}_{s,u}^{\star,\sigma}]_\alpha\right)} - \sqrt{\mathsf{Var}_{P_{s,a}^0}\left([\widehat{V}_{s,u}^{\star,\sigma}]_\alpha\right)} \right| \leq 2\sqrt{\frac{2\log(\frac{24SAN|N_{\varepsilon_2}|}{\delta})}{(1-\gamma)^2 N}} \quad (190)$$

1142 holds for all $(s,a,u) \in \mathcal{S}\times\mathcal{A}\times N_{\varepsilon_2}$.

1143 To continue, we decompose the term of interest in (187) as follows:

$$\max_{\alpha\in\left[\min_s \widehat{V}^{\widehat{\pi},\sigma}(s),\max_s \widehat{V}^{\widehat{\pi},\sigma}(s)\right]} \left| \sqrt{\mathsf{Var}_{\widehat{P}_{s,a}^0}\left(\left[\widehat{V}^{\widehat{\pi},\sigma}\right]_\alpha\right)} - \sqrt{\mathsf{Var}_{P_{s,a}^0}\left(\left[\widehat{V}^{\widehat{\pi},\sigma}\right]_\alpha\right)} \right|$$

$$\leq \max_{\alpha\in[0,1/(1-\gamma)]}\left|\sqrt{\mathsf{Var}_{\widehat{P}^0_{s,a}}\left(\left[\widehat{V}^{\widehat{\pi},\sigma}\right]_\alpha\right)}-\sqrt{\mathsf{Var}_{P^0_{s,a}}\left(\left[\widehat{V}^{\widehat{\pi},\sigma}\right]_\alpha\right)}\right|$$

$$\overset{(i)}{\leq} \max_{\alpha\in[0,1/(1-\gamma)]}\left|\sqrt{\mathsf{Var}_{\widehat{P}^0_{s,a}}\left(\left[\widehat{V}^{\star,\sigma}\right]_\alpha\right)}-\sqrt{\mathsf{Var}_{P^0_{s,a}}\left(\left[\widehat{V}^{\star,\sigma}\right]_\alpha\right)}\right|$$

$$+\max_{\alpha\in[0,1/(1-\gamma)]}\left[\sqrt{\left|\mathsf{Var}_{\widehat{P}^0_{s,a}}\left(\left[\widehat{V}^{\widehat{\pi},\sigma}\right]_\alpha\right)-\mathsf{Var}_{\widehat{P}^0_{s,a}}\left(\left[\widehat{V}^{\star,\sigma}\right]_\alpha\right)\right|}\right.$$

$$\left.+\sqrt{\left|\mathsf{Var}_{P^0_{s,a}}\left(\left[\widehat{V}^{\widehat{\pi},\sigma}\right]_\alpha\right)-\mathsf{Var}_{P^0_{s,a}}\left(\left[\widehat{V}^{\star,\sigma}\right]_\alpha\right)\right|}\right]$$

$$\overset{(ii)}{\leq} \max_{\alpha\in[0,1/(1-\gamma)]}\left|\sqrt{\mathsf{Var}_{\widehat{P}^0_{s,a}}\left(\left[\widehat{V}^{\star,\sigma}\right]_\alpha\right)}-\sqrt{\mathsf{Var}_{P^0_{s,a}}\left(\left[\widehat{V}^{\star,\sigma}\right]_\alpha\right)}\right|$$

$$+\max_{\alpha\in[0,1/(1-\gamma)]}2\sqrt{\frac{2}{(1-\gamma)}\left\|\left[\widehat{V}^{\widehat{\pi},\sigma}\right]_\alpha-\left[\widehat{V}^{\star,\sigma}\right]_\alpha\right\|_\infty}$$

$$\leq \max_{\alpha\in[0,1/(1-\gamma)]}\left|\sqrt{\mathsf{Var}_{\widehat{P}^0_{s,a}}\left(\left[\widehat{V}^{\star,\sigma}\right]_\alpha\right)}-\sqrt{\mathsf{Var}_{P^0_{s,a}}\left(\left[\widehat{V}^{\star,\sigma}\right]_\alpha\right)}\right|+4\sqrt{\frac{\varepsilon_{\mathsf{opt}}}{(1-\gamma)^2}}, \qquad (191)$$

where (i) holds by the triangle inequality, (ii) arises from applying Lemma 2, and the last inequality holds by (48).

Armed with the above facts, invoking the identity $\widehat{V}^{\star,\sigma}=\widehat{V}^{\star,\sigma}_{s,u^\star}$ leads to that for all $(s,a)\in\mathcal{S}\times\mathcal{A}$, with probability at least $1-\delta$,

$$\max_{\alpha\in[0,1/(1-\gamma)]}\left|\sqrt{\mathsf{Var}_{\widehat{P}^0_{s,a}}\left(\left[\widehat{V}^{\star,\sigma}\right]_\alpha\right)}-\sqrt{\mathsf{Var}_{P^0_{s,a}}\left(\left[\widehat{V}^{\star,\sigma}\right]_\alpha\right)}\right|$$

$$=\max_{\alpha\in[0,1/(1-\gamma)]}\left|\sqrt{\mathsf{Var}_{\widehat{P}^0_{s,a}}\left(\left[\widehat{V}^{\star,\sigma}_{s,u^\star}\right]_\alpha\right)}-\sqrt{\mathsf{Var}_{P^0_{s,a}}\left(\left[\widehat{V}^{\star,\sigma}_{s,u^\star}\right]_\alpha\right)}\right|$$

$$\overset{(i)}{\leq}\max_{\alpha\in[0,1/(1-\gamma)]}\left|\sqrt{\mathsf{Var}_{\widehat{P}^0_{s,a}}\left(\left[\widehat{V}^{\star,\sigma}_{s,\overline{u}}\right]_\alpha\right)}-\sqrt{\mathsf{Var}_{P^0_{s,a}}\left(\left[\widehat{V}^{\star,\sigma}_{s,\overline{u}}\right]_\alpha\right)}\right|$$

$$+\max_{\alpha\in[0,1/(1-\gamma)]}\left[\sqrt{\left|\mathsf{Var}_{\widehat{P}^0_{s,a}}\left(\left[\widehat{V}^{\star,\sigma}_{s,u^\star}\right]_\alpha\right)-\mathsf{Var}_{\widehat{P}^0_{s,a}}\left(\left[\widehat{V}^{\star,\sigma}_{s,\overline{u}}\right]_\alpha\right)\right|}\right.$$

$$\left.+\sqrt{\left|\mathsf{Var}_{P^0_{s,a}}\left(\left[\widehat{V}^{\star,\sigma}_{s,u^\star}\right]_\alpha\right)-\mathsf{Var}_{P^0_{s,a}}\left(\left[\widehat{V}^{\star,\sigma}_{s,\overline{u}}\right]_\alpha\right)\right|}\right]$$

$$\overset{(ii)}{\leq}\max_{\alpha\in[0,1/(1-\gamma)]}\left|\sqrt{\mathsf{Var}_{\widehat{P}^0_{s,a}}\left(\left[\widehat{V}^{\star,\sigma}_{s,\overline{u}}\right]_\alpha\right)}-\sqrt{\mathsf{Var}_{P^0_{s,a}}\left(\left[\widehat{V}^{\star,\sigma}_{s,\overline{u}}\right]_\alpha\right)}\right|+4\sqrt{\frac{\varepsilon_2}{(1-\gamma)}}$$

$$\overset{(iii)}{\leq}2\sqrt{\frac{2\log(\frac{24SAN|N_{\varepsilon_2}|}{\delta})}{(1-\gamma)^2 N}}+4\sqrt{\frac{\varepsilon_2}{(1-\gamma)}}$$

$$\leq 6\sqrt{\frac{2\log(\frac{36SAN^2|N_{\varepsilon_2}|}{\delta})}{(1-\gamma)^2 N}}, \qquad (192)$$

where (i) holds by the triangle inequality, (ii) arises from applying Lemma 2 and the fact $\left\|\widehat{V}^{\star,\sigma}_{s,\overline{u}}-\widehat{V}^{\star,\sigma}_{s,u^\star}\right\|_\infty\leq\frac{\varepsilon_2}{(1-\gamma)}$ (see (114)), (iii) follows from (190), and the last inequality holds by letting $\varepsilon_2=\frac{2\log(\frac{24SAN|N_{\varepsilon_2}|}{\delta})}{(1-\gamma)N}$, which leads to $|N_{\varepsilon_2}|\leq\frac{3}{\varepsilon_2(1-\gamma)}\leq\frac{3N}{2}$.

In summary, inserting (192) back to (191) and (191) leads to with probability at least $1-\delta$,

$$\max_{\alpha\in\left[\min_s\widehat{V}^{\widehat{\pi},\sigma}(s),\max_s\widehat{V}^{\widehat{\pi},\sigma}(s)\right]}\left|\sqrt{\mathsf{Var}_{\widehat{P}^0_{s,a}}\left(\left[\widehat{V}^{\widehat{\pi},\sigma}\right]_\alpha\right)}-\sqrt{\mathsf{Var}_{P^0_{s,a}}\left(\left[\widehat{V}^{\widehat{\pi},\sigma}\right]_\alpha\right)}\right|$$

$$\leq 6\sqrt{\frac{2\sigma\log(\frac{36SAN^2|N_{\varepsilon_2}|}{\delta})}{(1-\gamma)^2 N}} + 4\sqrt{\frac{\sigma\varepsilon_{\mathsf{opt}}}{(1-\gamma)^2}} \tag{193}$$

holds for all $(s,a) \in \mathcal{S} \times \mathcal{A}$.

**Step 4: finishing up.** Inserting (193) and (188) back to (187), we complete the proof: with probability at least $1 - \delta$,

$$\left\| \underline{\widehat{P}}^{\widehat{\pi},\widehat{V}} \widehat{V}^{\widehat{\pi},\sigma} - \underline{P}^{\widehat{\pi},\widehat{V}} \widehat{V}^{\widehat{\pi},\sigma} \right\|_\infty$$

$$\leq 4\sqrt{\frac{\log(\frac{3SAN^{3/2}}{(1-\gamma)\delta})}{(1-\gamma)^2 N}} + \frac{2\gamma\varepsilon_{\mathsf{opt}}}{1-\gamma} + 6\sqrt{\frac{2\sigma\log(\frac{36SAN^2|N_{\varepsilon_2}|}{\delta})}{(1-\gamma)^2 N}} + 4\sqrt{\frac{\sigma\varepsilon_{\mathsf{opt}}}{(1-\gamma)^2}}$$

$$\leq 12\sqrt{\frac{2(1+\sigma)\log(\frac{36SAN^2}{\delta})}{(1-\gamma)^2 N}} + \frac{2\gamma\varepsilon_{\mathsf{opt}}}{1-\gamma} + 4\sqrt{\frac{\sigma\varepsilon_{\mathsf{opt}}}{(1-\gamma)^2}}. \tag{194}$$

### E.2.3 Proof of Lemma 17

For any $0 \leq \alpha_1, \alpha_2 \leq 1/(1-\gamma)$, one has

$$|J_{s,a}(\alpha_1, V) - J_{s,a}(\alpha_2, V)|$$

$$= \left| \left| \sqrt{\mathsf{Var}_{\widehat{P}_{s,a}^0}([V]_{\alpha_1})} - \sqrt{\mathsf{Var}_{P_{s,a}^0}([V]_{\alpha_1})} \right| - \left| \sqrt{\mathsf{Var}_{\widehat{P}_{s,a}^0}([V]_{\alpha_2})} - \sqrt{\mathsf{Var}_{P_{s,a}^0}([V]_{\alpha_2})} \right| \right|$$

$$\overset{(i)}{\leq} \left| \sqrt{\mathsf{Var}_{\widehat{P}_{s,a}^0}([V]_{\alpha_1})} - \sqrt{\mathsf{Var}_{P_{s,a}^0}([V]_{\alpha_1})} - \sqrt{\mathsf{Var}_{\widehat{P}_{s,a}^0}([V]_{\alpha_2})} + \sqrt{\mathsf{Var}_{P_{s,a}^0}([V]_{\alpha_2})} \right|$$

$$\leq \left| \sqrt{\mathsf{Var}_{\widehat{P}_{s,a}^0}([V]_{\alpha_1})} - \sqrt{\mathsf{Var}_{\widehat{P}_{s,a}^0}([V]_{\alpha_2})} \right| + \left| \sqrt{\mathsf{Var}_{P_{s,a}^0}([V]_{\alpha_1})} - \sqrt{\mathsf{Var}_{P_{s,a}^0}([V]_{\alpha_2})} \right|$$

$$\overset{(ii)}{\leq} \sqrt{\mathsf{Var}_{\widehat{P}_{s,a}^0}([V]_{\alpha_2}) - \mathsf{Var}_{\widehat{P}_{s,a}^0}([V]_{\alpha_1})} + \sqrt{\mathsf{Var}_{P_{s,a}^0}([V]_{\alpha_2}) - \mathsf{Var}_{P_{s,a}^0}([V]_{\alpha_1})}$$

$$\overset{(iii)}{\leq} \sqrt{\left| \widehat{P}_{s,a}^0 \left[ ([V]_{\alpha_1}) \circ ([V]_{\alpha_1}) - ([V]_{\alpha_2}) \circ ([V]_{\alpha_2}) \right] \right| + \left| \widehat{P}_{s,a}^0 ([V]_{\alpha_1} + [V]_{\alpha_2}) \cdot \widehat{P}_{s,a}^0 ([V]_{\alpha_1} - [V]_{\alpha_2}) \right|}$$

$$+ \sqrt{\left| P_{s,a}^0 \left[ ([V]_{\alpha_1}) \circ ([V]_{\alpha_1}) - ([V]_{\alpha_2}) \circ ([V]_{\alpha_2}) \right] \right| + \left| P_{s,a}^0 ([V]_{\alpha_1} + [V]_{\alpha_2}) \cdot P_{s,a}^0 ([V]_{\alpha_1} - [V]_{\alpha_2}) \right|}$$

$$\leq 2\sqrt{2(\alpha_1 + \alpha_2)|\alpha_1 - \alpha_2|} \leq 4\sqrt{\frac{|\alpha_1 - \alpha_2|}{1-\gamma}}. \tag{195}$$

where (i) holds by the fact $||x| - |y|| \leq |x - y|$ for all $x, y \in \mathbb{R}$, (ii) follows from the fact that $\sqrt{x} - \sqrt{y} \leq \sqrt{x - y}$ for any $x \geq y \geq 0$ and $\mathsf{Var}_P([V]_{\alpha_2}) \geq \mathsf{Var}_P([V]_{\alpha_1})$ for any transition kernel $P \in \Delta(\mathcal{S})$, (iii) holds by the definition of $\mathsf{Var}_P(\cdot)$ defined in (24), and the last inequality arises from $0 \leq \alpha_1, \alpha_2 \leq 1/(1-\gamma)$.

## F  Proof of the lower bound with $\chi^2$ divergence: Theorem 4

To prove Theorem 4, we shall first construct some hard instances and then characterize the sample complexity requirements over these instances. The structure of the hard instances are the same as the ones used in the proof of Theorem 2.

### F.1  Construction of the hard problem instances

First, note that we shall use the same MDPs defined in Appendix D.1 as follows

$$\left\{ \mathcal{M}_\phi = \left( \mathcal{S}, \mathcal{A}, P^\phi, r, \gamma \right) \mid \phi = \{0, 1\} \right\}.$$

In particular, we shall keep the structure of the transition kernel in (123), reward function in (125) and initial state distribution in (126), while $p$ and $\Delta$ shall be specified differently later.

**Uncertainty set of the transition kernels.** Recalling the uncertainty set associated with $\chi^2$ divergence in (172), for any uncertainty level $\sigma$, the uncertainty set throughout this section is defined as $\mathcal{U}^\sigma(P^\phi)$:

$$\mathcal{U}^\sigma(P^\phi) := \otimes\, \mathcal{U}^\sigma_{\chi^2}(P^\phi_{s,a}),$$

$$\mathcal{U}^\sigma_{\chi^2}(P^\phi_{s,a}) := \left\{ P_{s,a} \in \Delta(\mathcal{S}) : \sum_{s' \in \mathcal{S}} \frac{\left(P(s'\,|\,s,a) - P^\phi(s'\,|\,s,a)\right)^2}{P^\phi(s'\,|\,s,a)} \le \sigma \right\}. \tag{196}$$

Clearly, $\mathcal{U}^\sigma(P^\phi_{s,a}) = P^\phi_{s,a}$ whenever the state transition is deterministic for $\chi^2$ divergence. Here, $q$ and $\Delta$ (whose choice will be specified later in more detail) which determine the instances are specified as

$$0 \le q = \begin{cases} 1 - \gamma & \text{if } \sigma \in \left(0, \frac{1-\gamma}{4}\right) \\ \frac{\sigma}{1+\sigma} & \text{if } \sigma \in \left[\frac{1-\gamma}{4}, \infty\right) \end{cases}, \qquad p = q + \Delta, \tag{197}$$

and

$$0 < \Delta \le \begin{cases} \frac{1}{4}(1-\gamma) & \text{if } \sigma \in \left(0, \frac{1-\gamma}{4}\right) \\ \min\left\{\frac{1}{4}(1-\gamma), \frac{1}{2(1+\sigma)}\right\} & \text{if } \sigma \in \left[\frac{1-\gamma}{4}, \infty\right) \end{cases}. \tag{198}$$

This directly ensures that

$$p = \Delta + q \le \max\left\{ \frac{\frac{1}{2} + \sigma}{1+\sigma}, \frac{5}{4}(1-\gamma) \right\} \le 1$$

since $\gamma \in \left[\frac{3}{4}, 1\right)$.

To continue, for any $(s, a, s') \in \mathcal{S} \times \mathcal{A} \times \mathcal{S}$, we denote the infimum probability of moving to the next state $s'$ associated with any perturbed transition kernel $P_{s,a} \in \mathcal{U}^\sigma(P^\phi_{s,a})$ as

$$\underline{P}^\phi(s'\,|\,s,a) := \inf_{P_{s,a} \in \mathcal{U}^\sigma(P^\phi_{s,a})} P(s'\,|\,s,a). \tag{199}$$

In addition, we denote the transition from state 0 to state 1 as follows, which plays an important role in the analysis,

$$\underline{p} := \underline{P}^\phi(1\,|\,0,\phi), \qquad \underline{q} := \underline{P}^\phi(1\,|\,0,1-\phi). \tag{200}$$

Before continuing, we introduce some facts about $\underline{p}$ and $\underline{q}$ which are summarized as the following lemma; the proof is postponed to Appendix F.3.1.

**Lemma 18.** *Consider any* $\sigma \in (0, \infty)$ *and any* $p, q, \Delta$ *obeying* (197) *and* (198)*, the following properties hold*

$$\begin{cases} \frac{1-\gamma}{2} < \underline{q} < 1-\gamma, \quad \underline{q} + \frac{3}{4}\Delta \le \underline{p} \le \underline{q} + \Delta \le \frac{5(1-\gamma)}{4} & \text{if } \sigma \in \left(0, \frac{1-\gamma}{4}\right), \\ \underline{q} = 0, \quad \frac{\sigma+1}{2}\Delta \le \underline{p} \le (3+\sigma)\Delta & \text{if } \sigma \in \left[\frac{1-\gamma}{4}, \infty\right). \end{cases} \tag{201}$$

**Value functions and optimal policies.** Armed with above facts, we are positioned to derive the corresponding robust value functions, the optimal policies, and its corresponding optimal robust value functions. For any RMDP $\mathcal{M}_\phi$ with the uncertainty set defined in (196), we denote the robust optimal policy as $\pi^\star_\phi$, the robust value function of any policy $\pi$ (resp. the optimal policy $\pi^\star_\phi$) as $V^{\pi,\sigma}_\phi$ (resp. $V^{\star,\sigma}_\phi$). The following lemma describes some key properties of the robust (optimal) value functions and optimal policies whose proof is postponed to Appendix F.3.2.

**Lemma 19.** *For any* $\phi = \{0, 1\}$ *and any policy* $\pi$*, one has*

$$V^{\pi,\sigma}_\phi(0) = \frac{\gamma z^\pi_\phi}{(1-\gamma)\left(1 - \gamma\left(1 - z^\pi_\phi\right)\right)}, \tag{202}$$

*where* $z^\pi_\phi$ *is defined as*

$$z^\pi_\phi := \underline{p}\,\pi(\phi\,|\,0) + \underline{q}\,\pi(1-\phi\,|\,0). \tag{203}$$

*In addition, the optimal value functions and the optimal policies obey*

$$V^{\star,\sigma}_\phi(0) = \frac{\gamma \underline{p}}{(1-\gamma)\left(1 - \gamma\left(1 - \underline{p}\right)\right)}, \tag{204a}$$

$$\pi^\star_\phi(\phi\,|\,s) = 1, \qquad \text{for } s \in \mathcal{S}. \tag{204b}$$

 **F.2   Establishing the minimax lower bound**

Our goal is to control the performance gap w.r.t. any policy estimator $\widehat{\pi}$ based on the generated
dataset and the chosen initial distribution $\varphi$ in (126), which gives

$$\left\langle \varphi, V_\phi^{\star,\sigma} - V_\phi^{\widehat{\pi},\sigma} \right\rangle = V_\phi^{\star,\sigma}(0) - V_\phi^{\widehat{\pi},\sigma}(0). \tag{205}$$

**Step 1: converting the goal to estimate $\phi$.**    To achieve the goal, we first introduce the following
fact which shall be verified in Appendix F.3.3: given

$$\varepsilon \leq \begin{cases} \frac{1}{72(1-\gamma)} & \text{if } \sigma \in \left(0, \frac{1-\gamma}{4}\right), \\ \frac{1}{256(1+\sigma)(1-\gamma)} & \text{if } \sigma \in \left[\frac{1-\gamma}{4}, \frac{1}{3(1-\gamma)}\right), \\ \frac{3}{32} & \text{if } \sigma > \frac{1}{3(1-\gamma)}. \end{cases} \tag{206}$$

choosing

$$\Delta = \begin{cases} 18(1-\gamma)^2\varepsilon & \text{if } \sigma \in \left(0, \frac{1-\gamma}{4}\right), \\ 64(1+\sigma)(1-\gamma)^2\varepsilon & \text{if } \sigma \in \left[\frac{1-\gamma}{4}, \frac{1}{3(1-\gamma)}\right), \\ \frac{16}{3(1+\sigma)}\varepsilon & \text{if } \sigma > \frac{1}{3(1-\gamma)}. \end{cases} \tag{207}$$

which satisfies the requirement of $\Delta$ in (197), it holds that for any policy $\widehat{\pi}$,

$$\left\langle \varphi, V_\phi^{\star,\sigma} - V_\phi^{\widehat{\pi},\sigma} \right\rangle \geq 2\varepsilon\left(1 - \widehat{\pi}(\phi \,|\, 0)\right). \tag{208}$$

**Step 2: arriving at the final results.**    To continue, following the same definitions and argument in
Appendix D.2, we recall the minimax probability of the error and its property as follows:

$$p_{\mathrm{e}} \geq \frac{1}{4}\exp\left\{ - N\Big(\mathsf{KL}\big(P^0(\cdot\,|\,0,0) \,\|\, P^1(\cdot\,|\,0,0)\big) + \mathsf{KL}\big(P^0(\cdot\,|\,0,1) \,\|\, P^1(\cdot\,|\,0,1)\big)\Big)\right\}, \tag{209}$$

then we can complete the proof by showing $p_{\mathrm{e}} \geq \frac{1}{8}$ given the bound for the sample size $N$. In the
following, we shall control the KL divergence terms in (209) in three different cases.

- Case 1: $\sigma \in \left(0, \frac{1-\gamma}{4}\right)$. In this case, applying $\gamma \in [\frac{3}{4}, 1)$ yields

$$1 - q > 1 - p = 1 - q - \Delta > \gamma - \frac{1-\gamma}{4} > \frac{3}{4} - \frac{1}{16} > \frac{1}{2},$$
$$p \geq q = 1 - \gamma. \tag{210}$$

  Armed with the above facts, applying Lemma 1 (cf. (23)) yields

$$\mathsf{KL}\big(P^0(\cdot\,|\,0,1) \,\|\, P^1(\cdot\,|\,0,1)\big) = \mathsf{KL}\left(p \,\|\, q\right) \leq \frac{(p-q)^2}{(1-p)p} \overset{\text{(i)}}{=} \frac{\Delta^2}{p(1-p)}$$
$$\overset{\text{(ii)}}{=} \frac{324(1-\gamma)^4\varepsilon^2}{p(1-p)}$$
$$\overset{\text{(iii)}}{\leq} 648(1-\gamma)^3\varepsilon^2, \tag{211}$$

  where (i) follows from the definition in (197), (ii) holds by plugging in the expression of $\Delta$ in
  (207), and (iii) arises from (210). The same bound can be established for $\mathsf{KL}\big(P_1^0(\cdot\,|\,0,0) \,\|\,$
  $P_1^1(\cdot\,|\,0,0)\big)$. Substituting (211) back into (209) demonstrates that: if the sample size is
  chosen as

$$N \leq \frac{\log 2}{1296(1-\gamma)^3\varepsilon^2}, \tag{212}$$

  then one necessarily has

$$p_{\mathrm{e}} \geq \frac{1}{4}\exp\left\{ - N \cdot 1296(1-\gamma)^3\varepsilon^2 \right\} \geq \frac{1}{8}. \tag{213}$$

- Case 2: $\sigma \in \left[\frac{1-\gamma}{4}, \frac{1}{3(1-\gamma)}\right)$. Applying the facts of $\Delta$ in (198), one has

$$1 - q > 1 - p = 1 - q - \Delta \geq \frac{1}{1+\sigma} - \frac{1}{2(1+\sigma)} = \frac{1}{2(1+\sigma)},$$
$$p \geq q = \frac{\sigma}{1+\sigma}. \tag{214}$$

Given (214), applying Lemma 1 (cf. (23)) yields

$$\mathsf{KL}\big(P^0(\cdot\,|\,0,1) \,\|\, P^1(\cdot\,|\,0,1)\big) = \mathsf{KL}\left(p \,\|\, q\right) \leq \frac{(p-q)^2}{(1-p)p} \overset{\text{(i)}}{=} \frac{\Delta^2}{p(1-p)}$$
$$\overset{\text{(ii)}}{=} \frac{4096(1+\sigma)^2(1-\gamma)^4\varepsilon^2}{p(1-p)}$$
$$\overset{\text{(iii)}}{\leq} \frac{4096(1+\sigma)^2(1-\gamma)^4\varepsilon^2}{\frac{\sigma}{2(1+\sigma)^2}} \leq \frac{8192(1-\gamma)^4(1+\sigma)^4\varepsilon^2}{\sigma}, \tag{215}$$

where (i) follows from the definition in (197), (ii) holds by plugging in the expression of $\Delta$ in (207), and (iii) arises from (214). The same bound can be established for $\mathsf{KL}\big(P_1^0(\cdot\,|\,0,0) \,\| \, P_1^1(\cdot\,|\,0,0)\big)$.

Substituting (215) back into (142) demonstrates that: if the sample size is chosen as

$$N \leq \frac{\sigma \log 2}{16384(1-\gamma)^4(1+\sigma)^4\varepsilon^2}, \tag{216}$$

then one necessarily has

$$p_{\mathrm{e}} \geq \frac{1}{4} \exp\left\{ -N \frac{16384(1-\gamma)^4(1+\sigma)^4\varepsilon^2}{\sigma} \right\} \geq \frac{1}{8}. \tag{217}$$

- Case 3: $\sigma > \frac{1}{3(1-\gamma)} \geq \frac{1}{3}$. Regarding this, one gives

$$1 - q > 1 - p = 1 - q - \Delta \geq \frac{1}{1+\sigma} - \frac{1}{4(1+\sigma)} \geq \frac{1}{2(1+\sigma)},$$
$$p \geq q \geq \frac{1}{4}. \tag{218}$$

Given $p \geq q \geq 1/2$ and (218), applying Lemma 1 (cf. (23)) yields

$$\mathsf{KL}\big(P^0(\cdot\,|\,0,1) \,\|\, P^1(\cdot\,|\,0,1)\big) = \mathsf{KL}\left(p \,\|\, q\right) \leq \frac{(p-q)^2}{(1-p)p} \overset{\text{(i)}}{=} \frac{\Delta^2}{p(1-p)}$$
$$\overset{\text{(ii)}}{\leq} \frac{\frac{64}{(1+\sigma)^2}\varepsilon^2}{p(1-p)}$$
$$\overset{\text{(iii)}}{\leq} \frac{492\varepsilon^2}{\sigma}, \tag{219}$$

where (i) follows from the definition in (197), (ii) holds by plugging in the expression of $\Delta$ in (207), and (iii) arises from (218). The same bound can be established for $\mathsf{KL}\big(P_1^0(\cdot\,|\,0,0) \,\|\, P_1^1(\cdot\,|\,0,0)\big)$. Substituting (219) back into (142) demonstrates that: if the sample size is chosen as

$$N \leq \frac{\sigma \log 2}{984\varepsilon^2}, \tag{220}$$

then one necessarily has

$$p_{\mathrm{e}} \geq \frac{1}{4} \exp\left\{ -N \frac{984\varepsilon^2}{\sigma} \right\} \geq \frac{1}{8}. \tag{221}$$

**Step 3: putting things together.** Finally, summing up the results in (212), (216), and (220), combined with the requirement in (206), one has when

$$\varepsilon \leq c_1 \begin{cases} \frac{1}{1-\gamma} & \text{if } \sigma \in \left(0, \frac{1-\gamma}{4}\right) \\ \max\left\{\frac{1}{(1+\sigma)(1-\gamma)}, 1\right\} & \text{if } \sigma \in \left[\frac{1-\gamma}{4}, \infty\right) \end{cases}, \tag{222}$$

taking

$$N \leq c_2 \begin{cases} \frac{1}{(1-\gamma)^3 \varepsilon^2} & \text{if } \sigma \in \left(0, \frac{1-\gamma}{4}\right) \\ \frac{\sigma}{\min\{1, (1-\gamma)^4 (1+\sigma)^4\} \varepsilon^2} & \text{if } \sigma \in \left[\frac{1-\gamma}{4}, \infty\right) \end{cases} \tag{223}$$

leads to $p_e \geq \frac{1}{8}$, for some universal constants $c_1, c_2 > 0$.

### F.3 Proof of the auxiliary facts

We begin with some basic facts about the $\chi^2$ divergence defined in (22) for any two Bernoulli distributions $\mathsf{Ber}(w)$ and $\mathsf{Ber}(x)$, denoted as

$$f(w, x) := \chi^2(x \parallel w) = \frac{(w-x)^2}{w} + \frac{(1-w-(1-x))^2}{1-w} = \frac{(w-x)^2}{w(1-w)}. \tag{224}$$

For $x \in [0, w)$, it is easily verified that the partial derivative w.r.t. $x$ obeys $\frac{\partial f(w,x)}{\partial x} = \frac{2(x-w)}{w(1-w)} < 0$, implying that

$$\forall\, x_1 < x_2 \in [0, w), \qquad f(w, x_1) > f(w, x_2). \tag{225}$$

In other words, the $\chi^2$ divergence $f(w, x)$ increases as $x$ decreases from $w$ to 0.

Next, we introduce the following function for any fixed $\sigma \in (0, \infty)$ and any $x \in \left[\frac{\sigma}{1+\sigma}, 1\right)$:

$$f_\sigma(x) := \inf_{\{y : \chi^2(y \parallel x) \leq \sigma, y \in [0, x]\}} y \stackrel{\text{(i)}}{=} \max\left\{0, x - \sqrt{\sigma x(1-x)}\right\} = x - \sqrt{\sigma x(1-x)}, \tag{226}$$

where (i) has been verified in Yang et al. (2022, Corollary B.2), and the last equality holds since $x \geq \frac{\sigma}{1+\sigma}$. The next lemma summarizes some useful facts about $f_\sigma(\cdot)$, which again has been verified in Yang et al. (2022, Lemma B.12 and Corollary B.2).

**Lemma 20.** *Consider any $\sigma \in (0, \infty)$. For $x \in \left[\frac{\sigma}{1+\sigma}, 1\right)$, $f_\sigma(x)$ is convex and differentiable, which obeys*

$$f_\sigma'(x) = 1 + \frac{\sqrt{\sigma}(2x-1)}{2\sqrt{x(1-x)}}.$$

#### F.3.1 Proof of Lemma 18

Let us control $\underline{q}$ and $\underline{p}$ respectively.

**Step 1: controlling $\underline{q}$.** We shall control $\underline{q}$ in different cases w.r.t. the uncertainty level $\sigma$.

- Case 1: $\sigma \in \left(0, \frac{1-\gamma}{4}\right)$. In this case, recall that $q = 1 - \gamma$ defined in (197), applying (226) with $x = q$ leads to

$$1 - \gamma = q > \underline{q} = f_\sigma(q) = 1 - \gamma - \sqrt{\sigma \gamma(1-\gamma)} \geq 1 - \gamma - \sqrt{\frac{1-\gamma}{4}\gamma(1-\gamma)} > \frac{1-\gamma}{2}. \tag{227}$$

- Case 2: $\sigma \in \left[\frac{1-\gamma}{4}, \infty\right)$. Note that it suffices to treat $P_{0,1-\phi}^\phi$ as a Bernoulli distribution $\mathsf{Ber}(q)$ over states 1 and 0, since we do not allow transition to other states. Recalling $q = \frac{\sigma}{1+\sigma}$ in (197) and noticing the fact that

$$f(q, 0) = \frac{q^2}{q} + \frac{(1-(1-q))^2}{1-q} = \frac{q}{(1-q)} = \sigma, \tag{228}$$

one has the probability $\mathsf{Ber}(0)$ falls into the uncertainty set of $\mathsf{Ber}(q))$ of size $\sigma$. As a result, recalling the definition (200) leads to

$$\underline{q} = \underline{P}^\phi(1 \mid 0, 1-\phi) = 0, \tag{229}$$

since $\underline{q} \geq 0$.

**Step 2: controlling $\underline{p}$.** To characterize the value of $\underline{p}$, we also divide into several cases separately.

- Case 1: $\sigma \in \left(0, \frac{1-\gamma}{4}\right)$. In this case, note that $p > q = 1 - \gamma \geq \frac{\sigma}{1+\sigma}$. Therefore, applying that $f_\sigma(\cdot)$ is convex and the form of its derivative in Lemma 20, one has

$$\underline{p} = f_\sigma(p) \geq f_\sigma(q) + f'_\sigma(q)(p - q)$$

$$= \underline{q} + \left(1 + \frac{\sqrt{\sigma}(2q - 1)}{2\sqrt{q(1-q)}}\right)\Delta \geq \underline{q} + \left(1 - \frac{\sqrt{\frac{1-\gamma}{4}}(1 - 2(1-\gamma))}{2\sqrt{(1-\gamma)\gamma}}\right)\Delta \geq \underline{q} + \frac{3\Delta}{4}. \tag{230}$$

Similarly, applying Lemma 20 leads to

$$\underline{p} = f_\sigma(p) \leq f_\sigma(q) + f'_\sigma(p)(p - q)$$

$$= \underline{q} + \left(1 - \frac{\sqrt{\sigma}(1 - 2p)}{2\sqrt{p(1-p)}}\right)\Delta \leq \underline{q} + \Delta, \tag{231}$$

where the last inequality holds by $1 - 2p > 0$ due to the fact $p = q + \Delta \leq \frac{5}{4}(1 - \gamma) \leq \frac{5}{16} < \frac{1}{2}$ (cf. (198) and $\gamma \in [\frac{3}{4}, 1)$). To sum up, given $\sigma \in \left(0, \frac{1-\gamma}{4}\right)$, combined with (227), we arrive at

$$\underline{q} + \frac{3}{4}\Delta \leq \underline{p} \leq \underline{q} + \Delta \leq \frac{5(1-\gamma)}{4}, \tag{232}$$

where the last inequality holds by $\Delta \leq \frac{1}{4}(1 - \gamma)$ (see (197)).

- Case 2: $\sigma \in \left[\frac{1-\gamma}{4}, \infty\right)$. We recall that $p = q + \Delta > q = \frac{\sigma}{1+\sigma}$ in (197). To derive the lower bound for $\underline{p}$ in (200), similar to (230), one has

$$\underline{p} = f_\sigma(p) \geq f_\sigma(q) + f'_\sigma(q)(p - q)$$

$$= \underline{q} + \left(1 + \frac{\sqrt{\sigma}(2q - 1)}{2\sqrt{q(1-q)}}\right)\Delta$$

$$\overset{\text{(i)}}{=} 0 + \left(1 + \frac{\sqrt{\sigma}\frac{\sigma-1}{1+\sigma}}{2\sqrt{\frac{\sigma}{1+\sigma}\frac{1}{1+\sigma}}}\right)\Delta = \left(1 + \frac{\sigma-1}{2}\right)\Delta = \left(\frac{\sigma+1}{2}\right)\Delta, \tag{233}$$

where (i) follows from $q = \frac{\sigma}{1+\sigma}$ and $\underline{q} = 0$ (see (229)). For the other direction, similar to (231), we have

$$\underline{p} = f_\sigma(p) \leq f_\sigma(q) + f'_\sigma(p)(p - q) = \underline{q} + \left(1 + \frac{\sqrt{\sigma}(2p - 1)}{2\sqrt{p(1-p)}}\right)\Delta$$

$$\overset{\text{(i)}}{=} \left(1 + \frac{\sqrt{\sigma}(2p - 1)}{2\sqrt{p(1-p)}}\right)\Delta \overset{\text{(ii)}}{=} \left(1 + \frac{\sqrt{\sigma}\left(\frac{\sigma-1}{1+\sigma} + 2\Delta\right)}{2\sqrt{\left(\frac{\sigma}{1+\sigma} + \Delta\right)\left(\frac{1}{1+\sigma} - \Delta\right)}}\right)\Delta$$

$$\overset{\text{(iii)}}{\leq} \left(1 + \frac{\sqrt{\sigma}(1 + 2\Delta)}{2\sqrt{\frac{\sigma}{1+\sigma} \cdot \frac{1}{2(1+\sigma)}}}\right)\Delta \overset{\text{(iv)}}{\leq} \left(1 + (1 + \sigma)\left(1 + \frac{1}{1+\sigma}\right)\right)\Delta = (3 + \sigma)\Delta, \tag{234}$$

where (i) holds by $\underline{q} = 0$ (see (229)), (ii) follows from plugging in $p = q + \Delta = \frac{\sigma}{1+\sigma} + \Delta$, and (iii) and (iv) arises from $\Delta = \min\left\{\frac{1}{4}(1 - \gamma), \frac{1}{2(1+\sigma)}\right\} \leq 1$ in (198). Combining (233) and (234) yields

$$\frac{\sigma + 1}{2}\Delta \leq \underline{p} \leq (3 + \sigma)\Delta. \tag{235}$$

**Step 3: combining all the results.** Finally, summing up the results for both $\underline{q}$ (in (227) and (229)) and $\underline{p}$ (in (232) and (235)), we arrive at the advertised bound.

### F.3.2 Proof of Lemma 19

**The robust value function for any policy $\pi$.** For any $\mathcal{M}_\phi$ with $\phi \in \{0, 1\}$, we first characterize the robust value function of any policy $\pi$ over different states.

Towards this, it is easily observed that for any policy $\pi$, the robust value functions at state $s = 1$ or any $s \in \{2, 3, \cdots, S-1\}$ obey

$$V_\phi^{\pi,\sigma}(1) \overset{\text{(i)}}{=} 1 + \gamma V_\phi^{\pi,\sigma}(1) = \frac{1}{1-\gamma} \tag{236a}$$

and

$$\forall s \in \{2, 3, \cdots, S\}: \qquad V_\phi^{\pi,\sigma}(s) \overset{\text{(ii)}}{=} 0 + \gamma V_\phi^{\pi,\sigma}(1) = \frac{\gamma}{1-\gamma}, \tag{236b}$$

where (i) and (ii) is according to the facts that the transitions defined over states $s \geq 1$ in (123) give only one possible next state 1, leading to a non-random transition in the uncertainty set associated with $\chi^2$ divergence, and $r(1, a) = 1$ for all $a \in \mathcal{A}'$ and $r(s, a) = 0$ holds all $(s, a) \in \{2, 3, \cdots, S-1\} \times \mathcal{A}$. To continue, the robust value function at state 0 with policy $\pi$ satisfies

$$
\begin{aligned}
V_\phi^{\pi,\sigma}(0) &= \mathbb{E}_{a \sim \pi(\cdot \,|\, 0)} \left[ r(0, a) + \gamma \inf_{\mathcal{P} \in \mathcal{U}^\sigma(P_{0,a}^\phi)} \mathcal{P} V_\phi^{\pi,\sigma} \right] \\
&= 0 + \gamma \pi(\phi \,|\, 0) \inf_{\mathcal{P} \in \mathcal{U}^\sigma(P_{0,\phi}^\phi)} \mathcal{P} V_\phi^{\pi,\sigma} + \gamma \pi(1-\phi \,|\, 0) \inf_{\mathcal{P} \in \mathcal{U}^\sigma(P_{0,1-\phi}^\phi)} \mathcal{P} V_\phi^{\pi,\sigma} \tag{237} \\
&\overset{\text{(i)}}{\leq} \frac{\gamma}{1-\gamma}, \tag{238}
\end{aligned}
$$

where (i) holds by that $\|V_\phi^{\pi,\sigma}\|_\infty \leq \frac{1}{1-\gamma}$. Summing up the results in (236b) and (238) leads to

$$\forall s \in \{2, 3, \cdots, S\}, \qquad V_\phi^{\pi,\sigma}(1) > V_\phi^{\pi,\sigma}(s) \geq V_\phi^{\pi,\sigma}(0). \tag{239}$$

With the transition kernel in (123) over state 0 and the fact in (239), (237) can be rewritten as

$$
\begin{aligned}
V_\phi^{\pi,\sigma}(0) &= \gamma \pi(\phi \,|\, 0) \inf_{\mathcal{P} \in \mathcal{U}^\sigma(P_{0,\phi}^\phi)} \mathcal{P} V_\phi^{\pi,\sigma} + \gamma \pi(1-\phi \,|\, 0) \inf_{\mathcal{P} \in \mathcal{U}^\sigma(P_{0,1-\phi}^\phi)} \mathcal{P} V_\phi^{\pi,\sigma} \\
&\overset{\text{(i)}}{=} \gamma \pi(\phi \,|\, 0) \Big[ \underline{p} V_\phi^{\pi,\sigma}(1) + \left(1 - \underline{p}\right) V_\phi^{\pi,\sigma}(0) \Big] + \gamma \pi(1-\phi \,|\, 0) \Big[ \underline{q} V_\phi^{\pi,\sigma}(1) + \left(1 - \underline{q}\right) V_\phi^{\pi,\sigma}(0) \Big] \\
&\overset{\text{(ii)}}{=} \gamma z_\phi^\pi V_\phi^{\pi,\sigma}(1) + \gamma \left(1 - z_\phi^\pi\right) V_\phi^{\pi,\sigma}(0) \\
&= \frac{\gamma z_\phi^\pi}{(1-\gamma)\left(1 - \gamma(1 - z_\phi^\pi)\right)}, \tag{240}
\end{aligned}
$$

where (i) holds by the definition of $\underline{p}$ and $\underline{q}$ in (200), (ii) follows from the definition of $z_\phi^\pi$ in (203), and the last line holds by applying (236a) and solving the resulting linear equation for $V_\phi^{\pi,\sigma}(0)$.

**Optimal policy and its optimal value function.** To continue, observing that $V_\phi^{\pi,\sigma}(0) =: f(z_\phi^\pi)$ is increasing in $z_\phi^\pi$ since the derivative of $f(z_\phi^\pi)$ w.r.t. $z_\phi^\pi$ obeys

$$f'(z_\phi^\pi) = \frac{\gamma(1-\gamma)\left(1 - \gamma(1 - z_\phi^\pi)\right) - \gamma^2 z_\phi^\pi(1-\gamma)}{(1-\gamma)^2 \left(1 - \gamma(1 - z_\phi^\pi)\right)^2} = \frac{\gamma}{\left(1 - \gamma(1 - z_\phi^\pi)\right)^2} > 0,$$

where the last inequality holds by $0 \leq z_\phi^\pi \leq 1$. Further, $z_\phi^\pi$ is also increasing in $\pi(\phi \,|\, 0)$ (see the fact $\underline{p} \geq \underline{q}$ in (200)), the optimal robust policy in state 0 thus obeys

$$\pi_\phi^\star(\phi \,|\, 0) = 1. \tag{241}$$

Considering that the action does not influence the state transition for all states $s > 0$, without loss of generality, we choose the optimal robust policy to obey

$$\forall s > 0 : \quad \pi_\phi^\star(\phi \,|\, s) = 1. \tag{242}$$

Taking $\pi = \pi_\phi^\star$ and $z_\phi^{\pi_\phi^\star} = \underline{p}$ in (240), we complete the proof by showing the corresponding optimal robust value function at state 0 as follows:

$$V_\phi^{\star,\sigma}(0) = \frac{\gamma z_\phi^{\pi_\phi^\star}}{(1-\gamma)\left(1 - \gamma\left(1 - z_\phi^{\pi_\phi^\star}\right)\right)} = \frac{\gamma \underline{p}}{(1-\gamma)\left(1 - \gamma\left(1 - \underline{p}\right)\right)}.$$

### F.3.3 Proof of the claim (208)

Plugging in the definition of $\varphi$, we arrive at that for any policy $\pi$,

$$
\begin{aligned}
\left\langle \varphi, V_\phi^{\star,\sigma} - V_\phi^{\pi,\sigma} \right\rangle &= V_\phi^{\star,\sigma}(0) - V_\phi^{\pi,\sigma}(0) \\
&\overset{\text{(i)}}{=} \frac{\gamma \underline{p}}{(1-\gamma)\left(1 - \gamma(1 - \underline{p})\right)} - \frac{\gamma z_\phi^\pi}{(1-\gamma)\left(1 - \gamma(1 - z_\phi^\pi)\right)} \\
&= \frac{\gamma\left(\underline{p} - z_\phi^\pi\right)}{\left(1 - \gamma(1 - \underline{p})\right)\left(1 - \gamma(1 - z_\phi^\pi)\right)} \overset{\text{(ii)}}{\geq} \frac{\gamma\left(\underline{p} - z_\phi^\pi\right)}{\left(1 - \gamma\left(1 - \underline{p}\right)\right)^2} \\
&\overset{\text{(iii)}}{=} \frac{\gamma(\underline{p} - \underline{q})\left(1 - \pi(\phi \,|\, 0)\right)}{\left(1 - \gamma\left(1 - \underline{p}\right)\right)^2},
\end{aligned}
\tag{243}
$$

where (i) holds by applying Lemma 19, (ii) arises from $z_\phi^\pi \leq \underline{p}$ (see the definition of $z_\phi^\pi$ in (203) and the fact $\underline{p} \geq \underline{q} + \frac{3\Delta}{4}$ in (200)), and (iii) follows from the definition of $z_\phi^\pi$ in (203).

To further control (243), we consider it in two cases separately:

- Case 1: $\sigma \in \left(0, \frac{1-\gamma}{4}\right)$. In this case, applying Lemma 18 to (243) yields

$$
\begin{aligned}
\left\langle \varphi, V_\phi^{\star,\sigma} - V_\phi^{\pi,\sigma} \right\rangle &\geq \frac{\gamma(\underline{p} - \underline{q})\left(1 - \pi(\phi \,|\, 0)\right)}{\left(1 - \gamma\left(1 - \underline{p}\right)\right)^2} \geq \frac{\gamma \frac{3\Delta}{4}\left(1 - \pi(\phi \,|\, 0)\right)}{\left(1 - \gamma\left(1 - \frac{5(1-\gamma)}{4}\right)\right)^2} \\
&\geq \frac{\Delta\left(1 - \pi(\phi \,|\, 0)\right)}{9(1-\gamma)^2} = 2\varepsilon\left(1 - \pi(\phi \,|\, 0)\right),
\end{aligned}
\tag{244}
$$

where the penultimate inequality follows from $\gamma \geq 3/4$, and the last inequality holds by taking the specification of $\Delta$ in (207) as follows:

$$\Delta = 18(1-\gamma)^2 \varepsilon. \tag{245}$$

It is easily verified that taking $\varepsilon \leq \frac{1}{72(1-\gamma)}$ as in (206) directly leads to meeting the requirement in (198), i.e., $\Delta \leq \frac{1}{4}(1-\gamma)$.

- Case 2: $\sigma \in \left[\frac{1-\gamma}{4}, \infty\right)$. Similarly, applying Lemma 18 to (243) gives

$$
\left\langle \varphi, V_\phi^{\star,\sigma} - V_\phi^{\pi,\sigma} \right\rangle \geq \frac{\gamma(\underline{p} - \underline{q})\left(1 - \pi(\phi \,|\, 0)\right)}{\left(1 - \gamma\left(1 - \underline{p}\right)\right)^2} \geq \frac{\gamma \frac{\sigma+1}{2}\Delta\left(1 - \pi(\phi \,|\, 0)\right)}{\min\left\{1, (1 - \gamma(1 - (3+\sigma)\Delta))^2\right\}}
\tag{246}
$$

Before continuing, it can be verified that

$$
\begin{aligned}
1 - \gamma\left(1 - (3+\sigma)\Delta\right) &= 1 - \gamma + \gamma(3+\sigma)\Delta \\
&\overset{\text{(i)}}{\leq} 1 - \gamma + (3+\sigma)\min\left\{\frac{1}{4}(1-\gamma), \frac{1}{2(\sigma+1)}\right\}
\end{aligned}
$$

$$\leq \min\left\{2(1+\sigma)(1-\gamma), \frac{3}{2}\right\}, \tag{247}$$

where (i) is obtained by $\Delta \leq \min\left\{\frac{1}{4}(1-\gamma), \frac{1}{2(1+\sigma)}\right\}$ (see (197)). Applying the above fact to (246) gives

$$\left\langle \varphi, V_\phi^{\star,\sigma} - V_\phi^{\pi,\sigma}\right\rangle \geq \frac{\gamma^{\frac{\sigma+1}{2}}\Delta\big(1-\pi(\phi\,|\,0)\big)}{\min\left\{1, (1-\gamma(1-(3+\sigma)\Delta))^2\right\}} \overset{(i)}{\geq} \frac{3(\sigma+1)\Delta\big(1-\pi(\phi\,|\,0)\big)}{8\min\left\{4(1+\sigma)^2(1-\gamma)^2, 1\right\}}$$

$$\geq \frac{\Delta\big(1-\pi(\phi\,|\,0)\big)}{\min\left\{32(1+\sigma)(1-\gamma)^2, \frac{8}{3(1+\sigma)}\right\}} = 2\varepsilon\big(1-\pi(\phi\,|\,0)\big), \tag{248}$$

where (i) holds by $\gamma \geq \frac{3}{4}$ and (246), and the last equality holds by the specification in (207):

$$\Delta = \begin{cases} 64(1+\sigma)(1-\gamma)^2\varepsilon & \text{if } \sigma \in \left[\frac{1-\gamma}{4}, \frac{1}{3(1-\gamma)}\right), \\ \frac{16}{3(1+\sigma)}\varepsilon & \text{if } \sigma > \frac{1}{3(1-\gamma)}. \end{cases} \tag{249}$$

As a result, it is easily verified that the requirement in (198)

$$\Delta \leq \min\left\{\frac{1}{4}(1-\gamma), \frac{1}{2(1+\sigma)}\right\} \tag{250}$$

is met if we let

$$\varepsilon \leq \begin{cases} \frac{1}{256(1+\sigma)(1-\gamma)} & \text{if } \sigma \in \left[\frac{1-\gamma}{4}, \frac{1}{3(1-\gamma)}\right), \\ \frac{3}{32} & \text{if } \sigma > \frac{1}{3(1-\gamma)}, \end{cases} \tag{251}$$

as in (206).

The proof is then completed by summing up the results in the above two cases.