# OpenReview forum: "The Curious Price of Distributional Robustness in Reinforcement Learning with a Generative Model"
_NeurIPS.cc/2023/Conference — NeurIPS 2023 poster_

### Official Review · Reviewer_Pv5G · 2023-07-04

**Soundness:** 4 excellent
**Presentation:** 4 excellent
**Contribution:** 4 excellent
**Rating:** 8
**Confidence:** 4

**Summary:**

This paper studies the statistical nature of distributionally robust reinforcement learning under the generative model. Specifically, it studies two divergences, total variation or chi-square divergence, over the full range of uncertainty levels. The paper improves the upper and lower bounds, especially when the uncertainty levels are small. Based on these results, the paper partially answers the question of whether distributionally robust RL is harder to learn compared to the non-robust counterpart.

**Strengths:**

The paper's theoretical results are both interesting and important in understanding the statistical limits of distributionally robust RL. The paper gives a complete story in the TV divergence and chi-square divergence.

**Weaknesses:**

A brief explanation of the reason for the improvements in both the upper and lower bounds, compared to the previous technique, would significantly help readers better understand the technical contributions of this paper.

**Questions:**

Can the authors briefly explain the current challenges in giving parallel results under the KL divergence?

**Limitations:**

See weakness.

---

> ### Author Rebuttal · Authors · 2023-08-09
>
> We extend our thanks to the reviewer for their meticulous review and perceptive insights regarding future directions. It is gratifying to know that the reviewer found our results both interesting and important. In what follows, we provide our response to the reviewer's comments.
>
> ### 1. Adding discussion on the technical contribution for improving both the upper and lower bounds.
> Thanks for this valuable suggestion. We shall highlight the following discussion in the introduction.
> * **Technical contribution to improving upper bound.** Achieving the tighter upper bound for TV and $\chi^2$ uncertainty set requires different technical tools. For TV uncertainty set, the key part that pins down the sample complexity to be better than standard RL is a carefully derived tighter bound for the range of the robust value function (i.e., Lemma 7). For $\chi^2$ uncertainty set, we improve the existing sample complexity with a quadratic dependency on the state space $S$ to a linear one through 1) the property of the dual equivalence in Lemma 5; 2) the leave-one-out technique to decouple statistical dependency inspired by [3][4].
> * **Technical contribution to improving lower bound.** To achieve a much tighter lower bound compared to prior work [1], we construct new hard instances (RMDPs) illustrated in Figure 2(a) of the uploaded **PDF**, which are different from the ones that are usually used in standard RL [2] and prior art [1] of robust RL. The new instances are inspired by the asymmetric structure of RMDPs induced by the additional infimum operator in the robust value function, which are tailored based on the uncertainty level $\sigma$. Please refer to Appendix D and F for more details.
>
> ### 2. The challenges of extending to the KL divergence case.
> Thanks for raising this insightful question. Extending the current results to KL divergence is definitely of great interest. For the lower bound, it is promising to use the hard instance construction approach in this work to improve the lower bound of the existing work [5] about KL divergence, while the main challenge is that the calculation of KL case is much more complicated since it is more non-linear than both TV and $\chi^2$ and thus require careful design. To improve the upper bound in KL divergence case, we may need to carefully control the bound for the dual parameter --- which is now the bottleneck to achieve minimax-optimal sample complexity with respect to the effective horizon length $\frac{1}{1-\gamma}$. In summary, extending the results to KL divergence is promising but may need additional specific techniques.
>
>     [1] Yang, Wenhao, Liangyu Zhang, and Zhihua Zhang. "Toward theoretical understandings of robust Markov decision processes: Sample complexity and asymptotics." The Annals of Statistics 50.6 (2022): 3223-3248.
>     [2] Gheshlaghi Azar, Mohammad, Rémi Munos, and Hilbert J. Kappen. "Minimax PAC bounds on the sample complexity of reinforcement learning with a generative model." Machine learning 91 (2013): 325-349.
>     [3] Agarwal, Alekh, Sham Kakade, and Lin F. Yang. "Model-based reinforcement learning with a generative model is minimax optimal." Conference on Learning Theory. PMLR, 2020.
>     [4] Li, Gen, et al. "Breaking the sample size barrier in model-based reinforcement learning with a generative model." Advances in neural information processing systems 33 (2020): 12861-12872.
>     [5] Shi, Laixi, and Yuejie Chi. "Distributionally robust model-based offline reinforcement learning with near-optimal sample complexity." arXiv preprint arXiv:2208.05767 (2022).

---

> > ### Author Response · Authors · 2023-08-17
> > **Thanks for your insightful suggestions!**
> >
> > Dear reviewer,
> >
> > Thank you once again for investing your valuable time in providing feedback on our paper. Your insightful suggestions have led to significant improvements in our work, and we look forward to possibly receiving more feedback from you. Since the discussion period between the author and reviewer is rapidly approaching its end, we kindly request you to review our responses to ensure that we have addressed all of your concerns. Also, we remain eager to engage in further discussion about any additional questions you may have.
> >
> > Best,
> >
> > Authors

---

> > > ### Comment · Reviewer_Pv5G · 2023-08-19
> > >
> > > Thanks for your reply. I will keep my scores. Great job!

---

> > > > ### Author Response · Authors · 2023-08-19
> > > > **Thank you for your response!**
> > > >
> > > > Dear reviewer,
> > > >
> > > > Thank you again for supporting this work and valuable feedback!
> > > >
> > > > Best,
> > > >
> > > > Authors

---

### Official Review · Reviewer_wpEG · 2023-07-07

**Soundness:** 4 excellent
**Presentation:** 4 excellent
**Contribution:** 4 excellent
**Rating:** 7
**Confidence:** 3

**Summary:**

1. This paper studies the model robustness in RL via the framework of distributionally robust MDP.
2. The authors derive the sample complexity of RMDPs using a model-based algorithm called distributionally robust value iteration when the uncertainty set is measured via either total variation or chi square divergence.
3. The lower bounds are developed to benchmark its tightness
4. A interesting insight from this paper is that the required sample complexity of RMDPs can be higher or lower than the standard MDP based on the choice of uncertainty measurement.

**Strengths:**

1. Nolvety: the improvement of the upper and lower bounds over the existing work is not easy, new techniques are developed.
2. Presentation is very clear, especially the summarizations in Tables 1-2 and the illustrations in Figure 1
3. The insights from theoretical results are interesting: RMDPs can be harder than standard MDPs under the chi square distance but can be easier than standard MDPs under the TV distance.


**Weaknesses:**

1. It is better to add experiments to empirically demonstrate the theoretical results in the paper.
2. It is better to add a discussion on how the theoretical results in this paper could help the practitioners.

**Questions:**

What is the intuition / theoretical explanation that whether RMDPs are not harder nor easier than standard MDPs highly depends on both the size and shape of the uncertainty set?

---

> ### Author Rebuttal · Authors · 2023-08-09
>
> We appreciate the reviewer for careful review and insightful feedback. It is rewarding to know that the reviewer recognizes the significance of our contributions --- the new techniques and interesting takeaway message. In what follows, we provide our response to the reviewer's comments.
>
> ### 1. Experimentally demonstrate the theoretical results
> Thanks for the insightful suggestion. We have conducted new numerical experiments to demonstrate and verify the theoretical findings as the reviewer suggested. Please refer to **General response** and uploaded **PDF** for details.
>
>
>
> ### 2. Adding discussion of how the results can help practitioners
> Thanks for the valuable suggestion. We shall add the discussion in the main text:
> The findings motivate the practitioners to:
> * **Design robust RL algorithms since robustness may be a free lunch.** The results show that promoting additional robustness in RL algorithms does not necessarily harder than standard RL in terms of sample requirement. So we can take this free lunch to have robustness property for RL algorithms.
> * **Design the uncertainty set carefully.** The results show that the statistical difficulty of robust RL heavily depends on the shape and size of the prescribed uncertainty set. So we need to carefully design the uncertainty set: 1) shape: the results for TV uncertainty show that using simple or linear divergence function such as $\ell_p$ norm (including TV) may be easier than other divergence functions and leads to less sample requirement; 2) size: the results for $\chi^2$ uncertainty indicate that we should not choose too large uncertainty set, which both induce over-conservative algorithm and sample inefficiency (seen from the exploded sample requirement as $\sigma$ increasing).
>
>
> ### 3. Explanation of why RMDPs are not necessarily harder nor easier than standard MDPs --- depends on both the shape and size of the uncertainty set.
> Thanks for raising this insightful question. We would be happy to provide more explanation. The difficulty of solving standard RL or robust RL is mainly determined by the following error terms. Given the same number of samples (same $\widehat{P}^0$), a smaller error term means the task is easier.
>
> $\text{Standard RL:} \quad  \quad \delta\_{\text{RL}} = \underset{ {\color{blue}{\bf \text{  linear}}} \text{ w.r.t. } P^0 - \widehat{P}^0 }{\underbrace{\Big| P^0\widehat{V} -\widehat{P}^0\widehat{V} \Big|}}$
>
> $\text{Robust RL:} \quad  \delta\_{\text{robust RL}}= \underset{ {\color{red}{\bf \text{complex form} }}  \text{ w.r.t. } P^0 - \widehat{P}^0 \text{ due to inner problem over uncertainty set } \mathcal{U}^\sigma\_\rho(\cdot)}{\underbrace{ \Big|\inf\_{\mathcal{P}\in \mathcal{U}^\sigma\_\rho\left(P^0 \right)} \mathcal{P}\widehat{V}\_{\text{rob}}- \inf\_{\mathcal{P} \in \mathcal{U}^\sigma\_\rho\left(\widehat{P}^0 \right)} \mathcal{P}\widehat{V}\_{\text{rob}} \Big|}}$
>
> The error terms mainly depend on two factors: 1) the relationship w.r.t. the model estimate error $P^0 - \widehat{P}^0$; 2) the range of the value function $\widehat{V}$ or $\widehat{V}_{\text{rob}}$.
>
> **Briefly, both the shape and size of the uncertainty set influence the error term and thus determine the difficulty of robust RL problems with certain sample size:** 1) using different uncertainty shapes will lead to either simple relationship (TV) or complex relationship ($\chi^2$) w.r.t $P^0 - \widehat{P}^0$； 2）different size determined by $\sigma$ will lead to different range of robust value function $\widehat{V}_{\text{rob}}$.
> More specifically, we show how the shape and size determine the statistical difficulty of robust RL for TV or $\chi^2$ uncertainty set, respectively:
> * **Using TV uncertainty set: easier than standard RL.** In this case, the error term is shown to be $\delta\_{\text{robust RL}} = \Big| P^0\widehat{V}\_{\text{rob}} -\widehat{P}^0\widehat{V}\_{\text{rob}} \Big|$ (determined by the TV uncertainty shape) that is also linear w.r.t. $P^0 - \widehat{P}^0$ --- the same as standard RL. While the range of robust value function $\widehat{V}\_{\text{rob}}$ in robust RL can reduce rapidly (as the size $\sigma$ increases) and becomes smaller than the range of $\widehat{V}$ in standard RL, since the values in all states are pushed toward the minimum one and become close to each other. As a result, the error term of robust RL $\delta\_{\text{robust RL}}$ is equal to or smaller than standard RL $\delta_{\text{RL}}$ when trained with the same size of samples, i.e., robust RL becomes easier than standard RL.
> * **Using $\chi^2$ uncertainty set: can be harder than standard RL.** In this case, the error term of robust RL $\delta_{\text{robust RL}}$ is non-linear w.r.t. and sensitive to model estimate error $P^0 - \widehat{P}^0$ (determined by the $\chi^2$ uncertainty shape), which leads to a large error term even if $P^0 - \widehat{P}^0$ is small, especially when $\sigma$ is large (e.g., $\sigma > O(\frac{1}{1-\gamma})$). So when using the same size of samples --- same $P^0 - \widehat{P}^0$, standard RL has an error term that is linear to $P^0 - \widehat{P}^0$, while the error term of robust RL may explode, namely, robust RL becomes much harder than standard RL.

---

> > ### Author Response · Authors · 2023-08-17
> > **Thanks for your insightful suggestions!**
> >
> > Dear reviewer,
> >
> > Thank you once again for investing your valuable time in providing feedback on our paper. We add new experiments inspired by your insightful suggestions, and we look forward to possibly receiving more feedback from you. Since the discussion period between the author and reviewer is rapidly approaching its end, we kindly request you to review our responses to ensure that we have addressed all of your concerns. Also, we remain eager to engage in further discussion about any additional questions you may have.
> >
> > Best,
> >
> > Authors

---

> > ### Author Response · Authors · 2023-08-21
> > **Kindly reminder of the discussion ending**
> >
> > Dear reviewer,
> >
> > Thank you once again for supporting this work and providing feedback on our paper. As the discussion period will end within the next day, we kindly ask that you review our responses to ensure that we have addressed all of your concerns.
> >
> > Best,
> >
> > Authors

---

### Official Review · Reviewer_fPn4 · 2023-07-09

**Soundness:** 3 good
**Presentation:** 3 good
**Contribution:** 3 good
**Rating:** 5
**Confidence:** 4

**Summary:**

The research primarily focuses on the robust Markov Decision Process setting, where the objective is to learn a robust policy with the uncertainty set being measured by f-divergences using a model-based algorithm. One of the key idea of the paper lies in deriving precise sample complexity bounds for the same with access to the generative model and shows that RMDPs is not necessarily more difficult than standard MDPs, which is an important finding in general.

**Strengths:**

One of the primary challenges in the field of Robust MDPs in general lies in the selection of the uncertainty set and shape and size of the uncertainty set, and the past literature has assumed certain structures around the same specifically (s,a) rectangularity to solve the robust MDP problem.  The paper provides a detailed study of the effect of uncertainty set on the robust MDP and the sample complexity analysis. A primary contribution of the paper lies in reducing the significant gap between the upper and lower bounds of past literature with different f-divergences. A standard formulation in robust MDP deals with defining the uncertainty-set $U_{\rho}^{\sigma}(P^0)$ with a divergence $\rho$ and radius $\sigma$ around the nominal transition kernel following the (s,a) rectangularity condition and solving the robust Value iteration. Although the robust value iteration is hard to solve in general, the dual of the problem is reasonably simple and can be applied due to the strong duality condition. With this, the authors develop distributionally robust value iteration and improved the upper bounds for RMDPs with various f-divergences. The Theorems also highlight the sample complexity over the entire range of the uncertainty level and how the geometry of the uncertainty set can affect the sample complexity analysis of RMDPs. Overall the paper is clearly written, with detailed analysis, and also provides the theory and proofs of earlier results as required, which I found very helpful.





**Weaknesses:**

1. In the statement of Theorem 1, where the relation of RMDP's sample complexity with the standard MDP is discussed, it states when $\sigma \leq 1 -\gamma$, the sample complexity is almost similar to MDP which makes sense since for smaller $\sigma$ means eventually we are optimizing just over the nominal $P^0$. However, the other result shows that when $\sigma > 1- \gamma$, the sample complexity of RMDPs becomes smaller than MDP, which is not very intuitive. When the uncertainty set over which the robust optimization is performed, when we increase the size of the set ideally the complexity should increase. Hence, it is not very clear what is the intuition of this result and is not explicitly stated here. Why the shape of the uncertainty set with TV can produce a better complexity analysis is not absolutely clear and intuitive.
2. The authors propose a distributionally robust value iteration algorithm (Algorithm 1) which is resulting in an improved analysis. However, the novelty of the proposed algorithm is not very clear as robust value iteration and solving the robust problem with Wasserstein divergence have been there long before. It will be critical to specifically highlight the novel aspects of the algorithm and which component is causing the improvement in the analysis.
3. The papers lack any experimental study to verify the theoretical claims for the proposed algorithm. It will be helpful if the authors can provide simple experimental evaluations or ablations on some vanilla RL environments, and how the proposed algorithm is achieving the improvements over prior in terms of sample complexity. That will make the intuition of the improvement in the algorithm very clear.
4. It is mentioned the distribution shift issue for RMDPs when the transition kernel drawn from the uncertainty set can be different from the nominal kernel. However, it's not clear whether it is addressed by the current algorithm and how? Also, won't this issue be increased when the size of the uncertainty set is large requiring more samples, which seems opposite to what it is shown in Theorem 1? It also needs further clarity.

**Questions:**

1. In Theorem 1, when $\sigma > 1- \gamma$, the sample complexity of RMDPs becomes smaller than MDP. However, when we increase the size of the uncertainty set, indicates we are more unsure of the transition and can come from a larger radius around the nominal. Shouldn't we need more samples in this case, why is considering the total variation causing the reverse? This is critical and will be helpful to have a detailed discussion.
2. The author compares the sample-complexity results of RMDPs with standard MDP which is interesting and important. However, the result(Th 1) holds for $\epsilon \in [0, \frac{k_0}{1 - \gamma}$. However, in Agarwal et.al, they also show the sample complexity is $O(S^2 A)$ when $\epsilon^2 \geq \frac{1}{k. (1-\gamma)^3}$ i.e. in regimes where no meaningfully accurate approximation to the actual transition probabilities can be constructed. It will be in
3. The authors have mentioned the geometry of the uncertainty set in connection to $\sigma$, but how is the geometry defined and what the geometry of the set refers to is not explicitly mentioned. What exactly does the geometry of the set refer to? Does it just refer to the shape owing to the type of divergence and size corresponding to the radius? Or there are further detailed connections drawn to the geometry?
4. Although the research closes the gap b/w the upper and lower bounds with more precise bounds, the generative model assumption, in general, is restrictive, and one of the major hardness for model-based RL is to approximate the transition model which is hard.

#Minor Comment
In several places, the citation and references were not coming properly, especially in the Appendix (Lines 663, 689, 692, 701, 800, 801, 1079), which makes it a little unsmooth to follow.

**Limitations:**

Some of the future and potential research directions are mentioned in the discussion, which I feel is sufficient.

---

> ### Author Rebuttal · Authors · 2023-08-09
>
> We appreciate the reviewer for the comprehensive feedback and recognize the significance of our contributions.
>
> ### 1. The intuition of why robust RL with TV uncertainty is easier than standard RL.
> The intuition is kind of the opposite --- as the uncertainty level $\sigma$ increases, fewer samples are needed to achieve a desired policy with a certain accuracy. This intuition is not limited to TV, but also any $\ell_p$ norm, even other uncertainty divergence functions. Specifically, recall the goal of robust RL is to find a policy $\pi$ whose performance is close to the robust optimal policy, i.e., $V^{\star,\sigma}(s) - V^{\pi,\sigma}(s) \leq \varepsilon$. As the uncertainty level increases, the performance gap between the optimal policy and any policy decreases rapidly, given that no policy can significantly outperform others when the environment may vary substantially. Consequently, only a small number of samples are required to reach the desired accuracy since the potential for improvement is minimal.
>
> ### 2. Clarification of the algorithm novelty in Section 3.
> * **We focus on statistical understanding of existing methods.** We didn't propose a new algorithm. Algorithm 1 in Section 3 is a well-known algorithm [1] for robust RL that we show here to keep this paper self-consistent. We highlight that this work focuses on studying the inherent difficulty of robust MDPs in terms of sample cost to provide a solid foundation for algorithm design. So we take Algorithm 1 as an example. Our statistical results can work for any model-based algorithms obeying certain conditions (can achieve a policy $\widehat{\pi}$ obeying $\|\widehat{V}^{\star, \sigma} - \widehat{V}^{\widehat{\pi}, \sigma}\|_\infty \leq \varepsilon_1$ for small enough $\varepsilon_1$) but not limited to Algorithm 1.
> * **Wasserstein uncertainty set.** We concur that Wasserstein distance is also an option for the uncertainty set, but it's relatively new in robust RL [2]. Investigating Wasserstein sets based on our findings presents an interesting direction.
>
> ### 3. New experiments to verify the theoretical claims for the proposed algorithm.
> We add new numerical experiments to verify the theoretical findings of the proposed algorithm as the reviewer suggested. Please refer to **General response** and uploaded **PDF** for details.
>
>
> ### 4. How Algorithm 1 addresses the uncertainty issue.
> To address the uncertainty that the transition kernel $\mathcal{P}$ may perturbs from the (estimated) nominal kernel $\widehat{P}^{0}$ inside the set $\mathcal{U}^\sigma(\widehat{P}^{0})$, each value iteration step in Algorithm 1 involves an additional infimum operator over $\mathcal{U}^\sigma(\widehat{P}^{0})$ as below (line 5 of Algorithm 1):
> $$\widehat{Q}\_t = r + \gamma \inf\_{ \mathcal{P} \in \mathcal{U}^\sigma(\widehat{P}^{0})}  \mathcal{P} \widehat{V}\_{t-1}.$$ In words, Algorithm 1 addresses the uncertainty by considering the worst case performance when the transition kernel is arbitrarily drawn in the uncertainty set $\mathcal{U}^\sigma(\widehat{P}^{0})$. And in addition, there won't be an issue when the uncertainty set size increases with $\sigma$ since Algorithm 1 knows the size ($\sigma$) and will solve the corresponding problem based on the size.
>
> ### 5. Discussion about the range of $\varepsilon$ in Theorem 1
> Theorem 1 in this work holds for a large range of accuracy level when $\varepsilon \in \left(0, \sqrt{1/\max\{1-\gamma, \sigma\}} \right]$. We leave the extension to full range as future work. Note that such extension is non-trivial that may require additional technical tools, such as [4] extend the range $\varepsilon \in \left(0, \sqrt{1/1-\gamma} \right]$ in [3] (the one reviewer mentioned) to full range in standard RL case.
>
> ### 6. Improving the specification.
> * **The uncertainty set geometry.** The formal definition of the uncertainty set is shown in Equation (3):
> $$\mathcal{U}\_\rho^{\sigma}(P^0) := \otimes \; \mathcal{U}\_\rho^{\sigma}(P^0\_{s,a})\qquad \text{with}\quad
> 	\mathcal{U}\_\rho^{\sigma}(P^0\_{s,a}) := \left\\{ P\_{s,a} \in \Delta (\mathcal{S}): \rho \left(P\_{s,a}, P^0\_{s,a}\right) \leq \sigma \right\\}.$$So the geometry of the uncertainty set (can be seen as a ball) is determined by two factor: 1) the divergence function $\rho(\cdot)$ which determine the 'distance' between two distribution point; 2) uncertainty level $\sigma$ that represents the radius of the uncertainty ball.
> * **Polishing the appendix.** We revise all the typos that the reviewer mentioned and polish the paper again.
>
> ### 7. Discussing the limitation of the generative model setting and model-based algorithms
> * **Generative model setting is a good starting point.** We believe understanding robust RL in the fundamental generative model setting is highly nontrivial and plays an essential role in shaping the theoretical foundation of robust RL, where all prior analyses still fail to give a clear message. Theoretical underpinnings in this setting are critically needed before addressing more complex cases, e.g., online/offline robust RL.
> * **Limitation of model-based RL.** We believe our work lays a solid foundation to design and understand the counterpart of model-based RL --- model-free RL which does not require estimating the model explicitly.
>
> > [1] Iyengar, G. N. (2005). Robust dynamic programming. Mathematics of Operations Research, 368 30(2):257–280.
> [2] Xu, Zaiyan, Kishan Panaganti, and Dileep Kalathil. "Improved sample complexity bounds for distributionally robust reinforcement learning." International Conference on Artificial Intelligence and Statistics. PMLR, 2023.\
>     [3] Agarwal, Alekh, Sham Kakade, and Lin F. Yang. "Model-based reinforcement learning with a generative model is minimax optimal." Conference on Learning Theory. PMLR, 2020.\
>     [4] Li, Gen, et al. "Breaking the sample size barrier in model-based reinforcement learning with a generative model." Advances in neural information processing systems 33 (2020): 12861-12872.

---

> > ### Comment · Reviewer_fPn4 · 2023-08-16
> > **Response to Rebuttal by Authors**
> >
> > Thanks for providing detailed and concrete comments on my concerns. I agree with most of the comments and justifications provided by the authors.
> >
> > I agree that indeed Generative model setting is a good starting point and potentially can extend to setting removing this assumption and this concern most originated regarding the claims and Point 2. Although, its reasonable to start with Generative model setting.
> > The insights are interesting and also want to thank the authors for providing the additional experiment which supports the hypothesis.
> > I am still not clear regarding the description on Point 1 especially the point on the second part " ....the sample complexity of RMDPs becomes smaller than MDP, which is not very intuitive". Can the authors clarify on this further for my understanding.
> > Also, the Wasserstein set in the context of robust MDP is not very new [1,2] and would request the authors to include a discussion on the same,
> >
> > 1. Esther Derman and Shie Mannor. (2020). Distributional Robustness and Regularization in Reinforcement Learning. Retrieved from https://arxiv.org/pdf/2003.02894.pdf
> >
> > 2. Mohammed Amin Abdullah, Hang Ren, Haitham Bou Ammar, Vladimir Milenkovic, Rui Luo, Mingtian Zhang, & Jun Wang. (2019). Wasserstein Robust Reinforcement Learning. Retrieved from https://arxiv.org/pdf/1907.13196.pdf

---

> > > ### Author Response · Authors · 2023-08-16
> > > **Response to reviewer fPn4**
> > >
> > > Dear Reviewer:
> > >
> > > Thank you so much for engaging in discussion with us and provide insightful feedback! We are grateful that the reviewer found our experiments helpful! We shall answer your questions and as following.
> > >
> > > ### More intuition about solving RMDPs with TV uncertainty is easier than standard MDPs.
> > > Thank you so much for raising this question since it is a key finding in this work --- promoting additional robustness in RL algorithms sometimes can be a free lunch in terms of sample cost. We shall briefly show the key technical intuition and hope this will be helpful.
> > >
> > > The difficulty of solving standard RL or robust RL is mainly determined by the following error terms. Given the same number of samples (same model estimate $\widehat{P}^0$), a smaller error term means the task is easier.
> > > $\text{Standard RL:} \quad  \quad \delta\_{\text{RL}} = \underset{ {\color{blue}{\bf \text{  linear}}} \text{ w.r.t. } P^0 - \widehat{P}^0 }{\underbrace{\Big| P^0\widehat{V} -\widehat{P}^0\widehat{V} \Big|}}$
> > >
> > > $\text{Robust RL:} \quad  \delta\_{\text{robust RL}}= \underset{ {\color{red}{\bf \text{complex form} } } \text{ w.r.t. } P^0 - \widehat{P}^0 \text{ due to inner problem over uncertainty set } \mathcal{U}^\sigma\_\rho(\cdot)}{\underbrace{ \Big|\inf\_{\mathcal{P}\in \mathcal{U}^\sigma\_\rho\left(P^0 \right)} \mathcal{P}\widehat{V}^{\sigma}\_{\text{rob}}- \inf\_{\mathcal{P} \in \mathcal{U}^\sigma_\rho\left(\widehat{P}^0 \right)} \mathcal{P}\widehat{V}^{\sigma}_{\text{rob}} \Big|}}$
> > >
> > > The error terms mainly depend on two factors: 1) the relationship w.r.t. the model estimate error $P^0 - \widehat{P}^0$; 2) the range of the value function $\widehat{V}$ or $\widehat{V}^{\sigma}\_{\text{rob}}$.
> > > * **Using TV uncertainty set: easier than standard RL.** In this case, the error term is shown to be $\delta\_{\text{robust RL}} = \Big| P^0\widehat{V}^{\sigma}\_{\text{rob}} -\widehat{P}^0\widehat{V}^{\sigma}\_{\text{rob}} \Big|$ that is also linear w.r.t. $P^0 - \widehat{P}^0$ --- the same as standard RL. While the range of robust value function $\widehat{V}^{\sigma}\_{\text{rob}}$ in robust RL decrease rapidly as level $\sigma$ increasing and becomes smaller than the range of $\widehat{V}$ in standard RL, since the values in all states are pushed toward the minimum one and become close to each other. As a result, the error term of robust RL $\delta\_{\text{robust RL}}$ becomes smaller than standard RL $\delta_{\text{RL}}$ as $\sigma$ grows, i.e., robust RL becomes easier than standard RL.
> > >
> > >
> > > ### Using Wasserstein distance in Robust RL.
> > > Thanks for raising this question and providing important related works [1][2]. We will definitely add these references and provide more discussions in the related work section. We agree with the reviewer that the use of Wasserstein distance for robustness in RL has been explored by previous works [1-4]. However, these prior investigations using Wasserstein in RL either concentrate on empirical algorithms [2-3] or robust formulations distinct from robust MDPs (as considered in our work), until the recent study [4]. Specifically, [1] addresses a different robust formulation through regularization using Wasserstein distance and elucidates the theoretical connection between regularization and robust MDPs.
> > >
> > > We believe that the exploration of the Wasserstein uncertainty set in robust MDPs is far from mature [4] and is a interesting future work. Although our current work focuses on $f$-divergence (TV or $\chi^2$), the technical tools and discoveries in this study have substantial potential for Wasserstein distance, offering further insights.
> > >
> > > **We thank again for the reviewer's time and would be glad to discuss further if there are additional concerns.**
> > >
> > >     [1]Esther Derman and Shie Mannor. (2020). Distributional Robustness and Regularization in Reinforcement Learning. Retrieved from https://arxiv.org/pdf/2003.02894.pdf
> > >     [2]Mohammed Amin Abdullah, Hang Ren, Haitham Bou Ammar, Vladimir Milenkovic, Rui Luo, Mingtian Zhang, & Jun Wang. (2019). Wasserstein Robust Reinforcement Learning. Retrieved from https://arxiv.org/pdf/1907.13196.pdf
> > >     [3]Hou, Linfang, et al. "Robust reinforcement learning with Wasserstein constraint." arXiv preprint arXiv:2006.00945 (2020).
> > >     [4]Xu, Zaiyan, Kishan Panaganti, and Dileep Kalathil. "Improved sample complexity bounds for distributionally robust reinforcement learning." International Conference on Artificial Intelligence and Statistics. PMLR, 2023.

---

> > > ### Author Response · Authors · 2023-08-21
> > > **Kindly reminder of the discussion ending**
> > >
> > > Dear reviewer,
> > >
> > > Thank you once again for engaging in the discussion with us and providing insightful feedback. As the discussion period will end within the next day, we kindly ask that you review our responses to ensure that we have addressed your concerns. If we have met your expectations, we would greatly appreciate your consideration in raising your support for this paper!
> > >
> > > Best,
> > >
> > > Authors

---

### Official Review · Reviewer_ZjPC · 2023-07-11

**Soundness:** 3 good
**Presentation:** 3 good
**Contribution:** 3 good
**Rating:** 5
**Confidence:** 4

**Summary:**

This paper presents new upper and lower bounds for distributionally robust MDPs where the uncertainty set for the transition kernel is specified as a ball with $\chi^{2}$ divergence or total variation as the distance measure. The bounds significantly improve previous results.


**Strengths:**

The paper is generally well-written, and is easy and interesting to read. The results significantly improve existing bounds and are surprising. However, I didn't read the proofs.


**Weaknesses:**

* The notations $\lesssim$ and $\asymp$ should be defined. While $\lesssim$ is often used in many places, it seems it should be just $\le$ or $<$ according to the theorems.
* The distributionally robust value iteration algorithm is taken from (Yang et al., 2022; Iyengar, 2005), but Section 3 describes the algorithm as if this is a contribution from this paper.
* The bounds usually do not hold for all $\gamma \in (0, 1)$, thus there are still some gaps in the results, while the paper did not make this clear.  An intuitive discussion on why the results do not hold for $\gamma$ values outside the ranges given in the theorems are helpful.
* I find the statements of Theorem 2 and Theorem 4 unclear. A sample complexity lower bound result should show that no algorithm can solve a problem given fewer than a certain number of samples. However, these theorems do not mention algorithms, but instead mention two robust MDPs and take infimum with respect to the random policy $\hat{\pi}$. Did I miss something?
* I find the lower bound in Theorem 4 very surprising, as the lower bound is discontinuous and non-monotonic. I note that the paper claim the bounds are at least nearly tight, but is it possible that the lower bounds are not sufficiently tight? Can an intuitive explanation for the discontinuity and non-monotonicity be given?
* There are some discussions on whether uncertainty of the transition kernel makes learning a $\epsilon$-optimal value function harder, and the discussion is based on comparison of sample complexity lower bounds for robust MDPs and standard MDPs. However, this may be misleading as the lower bounds can be underestimating the complexity of a problem.
* The appendix has some broken references (??).


**Questions:**

I would appreciate responses to the points regarding the bounds in Weaknesses.

**Limitations:**

The paper didn't discuss the limitations. I believe the paper should at least point out that the results do not hold for all $\gamma$ values and provide a discussion on this.

---

> ### Author Rebuttal · Authors · 2023-08-09
>
> We would like to express our gratitude to the reviewer for their insightful feedback and valuable comments.
>
>
>
>
>
> ### 1. The range of the discounted factor $\gamma$ that is considered in our theorems.
> Recall that the discounted factor $\gamma$ determines the effective horizon length $\frac{1}{1-\gamma}$ of RL tasks, i.e., larger $\gamma$ leads to tasks with longer horizons.
> * **We consider the entire reasonable range.** We would like to highlight that the main feature and challenge of RL tasks is the **sequential** structure. So when $\gamma$ is small (e.g., $\gamma \in (0, \frac{1}{2}]$ leads to the effective horizon length is at most $2$), the sequential structure almost disappears and is of much less interest for RL community. So people usually focus on reasonable range $\gamma \in (c, 1)$ for some small positive constant $c$ [1][2], such as $\gamma \in [\frac{1}{2},1)$. The situations when $\gamma\in (0, c]$ are generally not paid attention by the RL community.
> * **Our results can be directly extended to a broader range of $\gamma$.**  Recall that all of our four theorems at least hold for $\gamma \in (\frac{3}{4}, 1)$. The theorems can be directly extended to a broader range of $\gamma \in  (c, 1)$ along with $c$ as small as desired so that almost cover the full range $(0,1)$.
>
>
> ### 2. Improving the statement of Theorem 2 and 4.
> The reviewer is correct that the lower bound means that no algorithm (estimator) can achieve the desired accuracy, given the sample size that is fewer than a certain number. So $\hat{\pi}$ in Theorem 2 and 4 does not represent a random policy, but represents the output of any algorithm (estimator). We will definitely improve the statement and make the definition of $\hat{\pi}$ more clear.
>
> ### 3. More explanation of the lower bound in Theorem 4.
> Thanks for being interested in our results.
> * **A nearly tight lower bound.** The lower bound is nearly tight regarding that: 1) it is tight at least for a certain range of uncertainty levels (when $\sigma \in O(1)$), since it matches with the upper bound in Theorem 3 which verifies that the lower bound in this range can't be improved anymore. In addition, the lower bound nearly matches with the upper bound up to the term about $\gamma$ when $\sigma \gtrsim O(\frac{1}{1-\gamma})$. The reviewer can refer to Figure 1(b) for illustration.
> While in other cases of $\sigma$, the reviewer is right that the lower bound may not be tight enough, which we leave to future work.
> * **Intuition of the non-monotonicity shape.** We highlight that as the uncertainty level $\sigma$ varies, the sample requirement may have different behaviors due to uncertainty set changing--- which is an important message from this work. However, we agree with the reviewer that the non-monotonicity is counterintuitive and may can be improved. Although we significantly improve the prior lower bound, we believe there may exist a gap towards the optimal one. The gap may be due to we construct the same set of hard instances (RMDPs) when $\sigma$ varies. While for different $\sigma$, distinct hard instances may be required to achieve tighter lower bound --- need specific designs.
>
> ### 4. Comparing the difficulty of robust RL and standard RL in terms of sample requirement --- only based on lower bound?
> We compare the difficulty of robust and standard RL by comprehensively summarizing the information of both lower bounds and upper bounds. Specifically, the sample complexity of standard RL was settled as $\widetilde{O}\left(\frac{SA}{(1-\gamma)^3\varepsilon^2} \right)$ by the matched lower bound and upper bound [3].
>
> The messages in this work are: 1) robust MDPs are easier to learn than standard MDPs under the TV distance. We arrive at this conclusion by settling down the sample complexity of robust MDPs with as $\widetilde{O} \left( \frac{SA}{(1-\gamma)^2\varepsilon^2} \min \left\\{ \frac{1}{1-\gamma}, \frac{1}{\sigma} \right\\} \right)$ through showing the matched lower bound (Theorem 2) and upper bound (Theorem 1); 2) robust MDPs can be harder to learn than standard MDPs under the $\chi^2$ divergence. This claim is reasonable since the derived sample complexity lower bound for robust MDPs (Theorem 4) already exceeds the sample complexity of standard MDPs, at least for a certain range of uncertainty levels (see Figure 1(b)). Although the lower bound in Theorem 4 may not be sufficiently tight, it already shows that robust MDPs can be harder than standard MDPs.
>
> ### 5. Improving clarity and writing.
> Thanks for the careful review and valuable suggestions.
> * **Confusion of Section 3.** The reviewer is correct that we didn't propose a new algorithm but concentrated on understanding existing methods. We shall definitely highlight that Algorithm 1 in Section 3 was proposed by [4] and is just an example. While indeed, our sample complexity upper bound can work for any model-based algorithms obeying certain conditions (can learn a policy $\widehat{\pi}$ obeying $\|\widehat{V}^{\star, \sigma} - \widehat{V}^{\widehat{\pi}, \sigma}\|_\infty \leq \varepsilon_1$ for small enough $\varepsilon_1$) but not limited to Algorithm 1.
> * **Notation and definitions of $\lesssim, \asymp$.** We add clarification for these notations in the main text.
> * **Typos in the appendix.** We fixed the typos mentioned by the reviewers and have polished the main text and appendix again.
>
>
> > [1]  Li, Gen, et al. "Settling the sample complexity of model-based offline reinforcement learning." arXiv preprint arXiv:2204.05275 (2022). \
> [2]    Yan, Yuling, et al. "The efficacy of pessimism in asynchronous Q-learning." IEEE Transactions on Information Theory (2023). \
> [3] Li, Gen, et al. "Breaking the sample size barrier in model-based reinforcement learning with a generative model." Advances in neural information processing systems 33 (2020): 12861-12872. \
> [4] Iyengar, G. N. (2005). Robust dynamic programming. Mathematics of Operations Research, 368 30(2):257–280.

---

> > ### Author Response · Authors · 2023-08-17
> > **Thanks for your insightful suggestions!**
> >
> > Dear reviewer,
> >
> > Thank you once again for investing your valuable time in providing feedback on our paper. Your insightful suggestions have led to significant improvements in our work, and we look forward to possibly receiving more feedback from you. Since the discussion period between the author and reviewer is rapidly approaching its end, we kindly request you to review our responses to ensure that we have addressed all of your concerns. Also, we remain eager to engage in further discussion about any additional questions you may have.
> >
> > Best,
> >
> > Authors

---

> > > ### Author Response · Authors · 2023-08-21
> > > **Kindly reminder of the discussion ending**
> > >
> > > Dear reviewer,
> > >
> > > Thank you once again for dedicating your valuable time to provide feedback on our paper. As the discussion period will end within the next day, we kindly ask that you review our responses to ensure that we have fully addressed all of your concerns. If we have met your expectations, we would greatly appreciate your consideration in raising your support for this paper!
> > >
> > > Best,
> > >
> > > Authors

---

> > ### Comment · Reviewer_ZjPC · 2023-08-21
> > **discussion**
> >
> > Thanks for your response. Re the range of $\gamma$, it's not true that RL papers focus on "reasonable ranges": quite a few previous works only assume $\gamma \in (0, 1)$. For example, see Agarwal et al. 2020 (which is cited in your paper) and the references given there. Is there a technical reason that the range $(0, 1)$ can't be used? While your response says the results can be extended to a broader range, the way it is worded seems to suggest that simply replacing the restricted intervals by $(0, 1)$ doesn't work?
> >
> > Since the main text isn't updated, can you also clarify what $\lesssim$ and $\asymp$ mean in this paper?

---

> > > ### Author Response · Authors · 2023-08-21
> > > **Thank you engaging in the discussion and providing insightful feedback!**
> > >
> > > Dear reviewer,
> > >
> > > ### The range of $\gamma$.
> > >
> > > Thank you for raising this question! We totally agree with the reviewer that many works deal with the full range $\gamma\in(0,1)$. We are sorry about the confusion: we do not mean no one considers $\gamma\in(0,1)$, but just saying researchers usually do not consider two results as too different if the only difference is that one works for $\gamma\in(0,1)$ and the other works for $\gamma\in(\frac{1}{2}, 1)$. The reason is that tasks with $\gamma$ being very small will make the important sequential structure of RL disappear and is of much less interest.
> > >
> > > In addition, our Theorems **can work for the full range $\gamma\in(0,1)$** by adapting some numerical numbers in the proof. Since the assumption for $\gamma\in [c, 1)$ is just for calculation convenience such as in Equation (70) of Appendix C.2. As long as $\gamma$ is a constant obeying $\gamma \in (0,1)$, we can have the same conclusion.
> > >
> > >
> > >
> > > ### Notation and definitions of $\lesssim, \asymp$
> > > Thank you for raising this question. We are sorry about this since we hope to include this in the first response but delete it due to the space limit. We will add this in the introduction for clarification.
> > >
> > > > Here and throughout, we use the standard notation $f(n)=O(g(n))$ to indicate that $f(n)/g(n)$ is bounded above by a constant as $n$ grows. The notation $f(\mathcal{X}) = O(g(\mathcal{X}))$ or $f(\mathcal{X}) \lesssim g(\mathcal{X})$ indicates that there exists a universal constant $C_1>0$ such that $f\leq C_1 g$, the notation $f(\mathcal{X}) \gtrsim g(\mathcal{X})$ indicates that $g(\mathcal{X}) = O(f(\mathcal{X}))$, and the notation $f(\mathcal{X})\asymp g(\mathcal{X})$ indicates that  $f(\mathcal{X}) \lesssim g(\mathcal{X})$ and  $f(\mathcal{X}) \gtrsim g(\mathcal{X})$ hold simultaneously.
> > >
> > > Thank you once again for engaging in discussion with us. Hope that our answers are helpful and we look forward to further discussion if you have additional concerns.
> > >
> > > Best,
> > >
> > > Authors

---

### Official Review · Reviewer_aUcN · 2023-07-21

**Soundness:** 2 fair
**Presentation:** 2 fair
**Contribution:** 3 good
**Rating:** 6
**Confidence:** 3

**Summary:**

This paper focuses on the sample complexity of learning robust MDPs under generative model access, with uncertainty sets measured with  TV and chi-squared divergences. The paper proposes tight upper and lower bounds for sample complexity under TV robustness and chi-squared robustness (under some range of the uncertainty radius). Curiously, the minimax bound is smaller than standard MDPs. The upper bound is attained by a model-based distributionally robust value iteration algorithm.

(changed score from 4 -> 6 during discussion.)

**Strengths:**

1. The paper provides tight bounds for the TV case, and somewhat tight bounds for the chi-squared case, and illustrates the interesting phenomenon where the minimax optimal bounds are sometimes easier/harder than standard RL.

**Weaknesses:**

1. The presentation can be improved. For example, the algorithm box 1 is very bare and does not contain all the necessary details for implementation. Also, computational efficiency is a key issue in robust MDPs and the current version brushed it aside (in Line 221) into the appendix, which is 46 pages. It would be better to present it clearly in the main paper. Also, what makes this algorithm different from Yang et al, 2022? Is it just an improved analysis or something fundamentally changed in the algorithm to obtain minimax-optimal bounds?
2. The robust VI algorithm seems not novel, as it is a common method for robust MDPs, eg. Iyengar 2005.
3. While minimax-optimal theoretical results are nice, there is limited discussion on the practicality of the algorithm, besides its worst-case sample complexities.

**Questions:**

1. Which parts of the current analysis breaks when trying to extend to more general MDPs, like linear MDPs?
2. I really like Figure 1, but I do not understand the intuition on why robust MDP is easier/harder in certain cases. I can certainly follow the algebra based on minimax optimal bounds, but can you describe at an intuitive level why this "curious price" appears?

**Limitations:**

See above.

---

> ### Author Rebuttal · Authors · 2023-08-09
>
> We thank gratefully the reviewer for various valuable suggestions and the praise of our interesting findings!
>
> ### 1. Questions about Algorithm 1 in Section 3.
> Thanks for raising questions about algorithm 1. We shall address them as below:
> * **Improving the specification of Algorithm 1 and its computation efficiency.**
>  The introduced Algorithm 1 in Section 3 --- robust value iteration (VI) is a well-known algorithm proposed by prior art [1], which is efficient with computation cost $O(S \log(S))$ per iteration --- with only a modest increase compared to standard VI ($O(S)$). We shall introduce more details of the update rule and computation complexity in the main text to keep this paper self-consistent.
> * **Does this work propose a new algorithm?** The brief answer is no. We highlight that this work focuses on investigating the inherent difficulty of solving robust MDPs in terms of sample cost to provide a solid foundation for algorithm design. So we didn't propose new algorithms but concentrated on understanding existing ones, where we choose robust VI (Algorithm 1) as an example to develop sample complexity upper bound.
> While indeed, our statistical results can work for any model-based algorithms obeying certain conditions (can learn a policy $\widehat{\pi}$ obeying $\|\widehat{V}^{\star, \sigma} - \widehat{V}^{\widehat{\pi}, \sigma}\|_\infty \leq \varepsilon_1$ for small enough $\varepsilon_1$) but not limited to Algorithm 1.
> * **Practicity of Algorithm 1.** We add new experiments to evaluate Algorithm 1 and demonstrate our theoretical findings (please check the **General response** for details). In addition, Algorithm 1 with TV uncertainty or $\chi^2$ uncertainty set can be applied to more complex tasks and achieve robust performance when the testing environment deviates from the training one, such as Gambler’s problem and Frozen Lake environment in OpenAI Gym [3].
>
> ### 2. Challenges of extending to general MDP like linear MDPs.
> We believe the findings of current results in tabular cases lays a solid foundation to carry out general MDP cases with function approximation, e.g., the finding of using a simple divergence function such as TV may lead to less sample size requirement. While the entire pipeline in this work will need to adapt since general MDP cases (e.g., linear MDPs) will require distinct problem formulations --- still an open problem with few studies [2], algorithm design, and theoretical analysis framework --- requires more assumptions such as linearization for linear MDPs [4] and realizability/low-rank structure for general function approximation.
>
> ### 3. Intuitions of why robust MDPs are easier or harder than standard MDPs in certain cases.
> Thanks for liking our illustration in Figure 1. The difficulty of solving standard RL or robust RL is mainly determined by the following error terms. Given the same number of samples (same $\widehat{P}^0$), a smaller error term means the task is easier.
>
> $\text{Standard RL:} \quad  \quad \delta\_{\text{RL}} = \underset{ {\color{blue}{\bf \text{  linear}}} \text{ w.r.t. } P^0 - \widehat{P}^0 }{\underbrace{\Big| P^0\widehat{V} -\widehat{P}^0\widehat{V} \Big|}}$
>
> $\text{Robust RL:} \quad  \delta\_{\text{robust RL}}= \underset{ {\color{red}{\bf \text{complex form} }}  \text{ w.r.t. } P^0 - \widehat{P}^0 \text{ due to inner problem over uncertainty set } \mathcal{U}^\sigma\_\rho(\cdot)}{\underbrace{ \Big|\inf\_{\mathcal{P}\in \mathcal{U}^\sigma\_\rho\left(P^0 \right)} \mathcal{P}\widehat{V}\_{\text{rob}}- \inf\_{\mathcal{P} \in \mathcal{U}^\sigma\_\rho\left(\widehat{P}^0 \right)} \mathcal{P}\widehat{V}\_{\text{rob}} \Big|}}$
>
> The error terms mainly depend on two factors: 1) the relationship w.r.t. the model estimate error $P^0 - \widehat{P}^0$; 2) the range of the value function $\widehat{V}$ or $\widehat{V}_{\text{rob}}$.
> * **Using TV uncertainty set: easier than standard RL.** In this case, the error term is shown to be $\delta\_{\text{robust RL}} = \Big| P^0\widehat{V}\_{\text{rob}} -\widehat{P}^0\widehat{V}\_{\text{rob}} \Big|$ that is also linear w.r.t. $P^0 - \widehat{P}^0$ --- the same as standard RL. While the range of robust value function $\widehat{V}\_{\text{rob}}$ in robust RL decrease rapidly and becomes smaller than the range of $\widehat{V}$ in standard RL, since the values in all states are pushed toward the minimum one and become close to each other. As a result, the error term of robust RL $\delta_{\text{robust RL}}$ becomes smaller than standard RL $\delta_{\text{RL}}$ as $\sigma$ grows, i.e., robust RL becomes easier than standard RL.
> * **Using $\chi^2$ uncertainty set: can be harder than standard RL.** In this case, the error term of robust RL $\delta_{\text{robust RL}}$ is non-linear w.r.t. and sensitive to model estimate error $P^0 - \widehat{P}^0$, which can induce large error term even if $P^0 - \widehat{P}^0$ is small, especially when $\sigma$ is large (e.g., $\sigma > O(\frac{1}{1-\gamma})$). Standard RL has an error term that is linear to $P^0 - \widehat{P}^0$, while the error term of robust RL may explode even $P^0 - \widehat{P}^0$ is small, namely, robust RL becomes much harder than standard RL.
>
>
>
>
>
>
> > [1] Iyengar, G. N. (2005). Robust dynamic programming. Mathematics of Operations Research, 368 30(2):257–280. \
> [2] Ma, Xiaoteng, et al. "Distributionally robust offline reinforcement learning with linear function approximation." arXiv preprint arXiv:2209.06620 (2022). \
> [3] Panaganti, Kishan, and Dileep Kalathil. "Sample complexity of robust reinforcement learning with a generative model." International Conference on Artificial Intelligence and Statistics. PMLR, 2022. \
> [4] Jin, Chi, et al. "Provably efficient reinforcement learning with linear function approximation." Conference on Learning Theory. PMLR, 2020.

---

> ### Author Response · Authors · 2023-08-17
> **Thanks for your insightful suggestions!**
>
> Dear reviewer,
>
> Thank you once again for investing your valuable time in providing feedback on our paper. Your insightful suggestions have led to significant improvements in our work, and we look forward to possibly receiving more feedback from you. Since the discussion period between the author and reviewer is rapidly approaching its end, we kindly request you to review our responses to ensure that we have addressed all of your concerns. Also, we remain eager to engage in further discussion about any additional questions you may have.
>
> Best,
>
> Authors

---

> > ### Author Response · Authors · 2023-08-19
> > **Thank you raising the score and supporting this work!**
> >
> > Dear reviewer,
> >
> > Thank you so much for raising the score and increasing the support for this work!
> >
> > We truly value your feedback and hope that we have addressed your insightful concerns and questions adequately.
> >
> > Best,
> >
> > Authors

---

### Author Rebuttal · Authors · 2023-08-09

We thank the reviewers for their careful reading of the paper and their insightful and valuable feedback. Below we provide some new numerical results to corroborate the theoretical findings in this work.

### New numerical results
As reviewers suggested, we add new experiments to corroborate and demonstrate the theoretical findings in this paper:



* **Experimental settings.** We demonstrate the sample size requirements in robust RL when the uncertainty level $\sigma$ varies. Specifically, we evaluate robust value iteration (Algorithm 1) in the following robust MDP $\mathcal{M}_\phi= \left(\mathcal{S}, \mathcal{A}, P^0, r, \gamma \right)$ illustrated in the uploaded **PDF Figure 2(a)**, where $\gamma$ is the discount parameter, $\mathcal{S} = \{0, 1\}$, $\mathcal{A} = \{0, 1\}$, nominal transition kernel $P^0$ obeys $P^0(1|1,0) = P^0(1|1,1) =1, P^0(1|0,0) =p, P^0(1|0,1) =q$, reward $r(0,0) = r(0,1) =0, r(1,0) = r(1,1) = 1$. Denote N as the sample size per state-action pair. For each point $(N, \gamma, \sigma)$, we conduct 100 Monte Carlo simulations and claim $N$ successfully attain $\varepsilon$ accuracy if the accuracy is achieved at least $95$ times.
* **Results with TV uncertainty set.** For TV uncertainty set, we focus on the effect of the discount complexity $\frac{1}{1-\gamma}$, which dominates the difference of robust RL compared to standard RL (see Figure 1(a)). We set $p = 2 \max(1-\gamma, \sigma)$, and $q = p - 16(1-\gamma)\max(1-\gamma, \sigma)\varepsilon$ inspired by our lower bound, and fix $\varepsilon= 0.13$ (a randomly chosen small value). The results in **PDF Figure 2(b)** show that the numerical sample complexity per state-action pair $N$ scales on the order of $\frac{1}{(1-\gamma)^3}$ when the uncertainty level is small ($\sigma = 0.005$), while on the order of  $\frac{1}{(1-\gamma)^2}$ when the uncertainty level is large ($\sigma = 0.3$). The results match the derived sample requirements illustrated in Figure 1.
* **Results with $\chi^2$ uncertainty set.** For $\chi^2$ uncertainty set, we focus on the effect of $\sigma$ especially when $\sigma$ is very large, since the exploded sample requirement as $\sigma$ increases is a key finding. We set $q = \frac{\sigma}{1+\sigma}$, $p = q + \frac{8}{3(1+\sigma)}\varepsilon$, and fix $\varepsilon= 0.13$ and $\gamma = 0.9$. **PDF Figure 2(c)** demonstrates that the required sample complexity increases linearly w.r.t. the uncertainty level $\sigma$ when $\sigma$ is large (in the range $(\frac{1}{1-\gamma},\infty)$), which matches our theoretical findings (see Figure 1).


We would be grateful if the reviewers could take a look at the responses and consider raising support for this work if we have addressed your concerns adequately. We would be glad to discuss further if there are additional concerns.

---

### Decision · Program_Chairs · 2023-09-21

**Decision:**

Accept (poster)

**Comment:**

After discussion, reviewers all agree that that paper's new result on robust MDP is a good contribution, and the paper should be accepted.